# Learning the *essential* in less than 2k additional weights - a simple approach to improve image classification stability under corruptions

**Kai Bäuerle**                                           *kaibauerle@gmail.com*
*University of Mannheim, Germany*

**Patrick Müller**                              *patrick.mueller@uni-mannheim.de*
*University of Mannheim, Germany*

**Syed Muhammad Kazim**                              *syed.kazim@uni-siegen.de*
*University of Siegen and Center for Sensor Systems (ZESS), Siegen*

**Ivo Ihrke**                                       *ivo.ihrke@uni-siegen.de*
*University of Siegen and Center for Sensor Systems (ZESS), Siegen*

**Margret Keuper**                                 *keuper@uni-mannheim.de*
*University of Mannheim and Max Planck Institute for Informatics, Saarland Informatics Campus, Germany*

**Reviewed on OpenReview:** *https://openreview.net/forum?id=i2SuGWtIIm*

## Abstract

The performance of image classification on well-known benchmarks such as ImageNet is remarkable, but in safety-critical situations, the accuracy often drops significantly under adverse conditions. To counteract these performance drops, we propose a very simple modification to the models: we pre-pend a single, dimension preserving convolutional layer with a large linear kernel whose purpose it is to extract the information that is essential for image classification. We show that our simple modification can increase the robustness against common corruptions significantly, especially for corruptions of high severity. We demonstrate the impact of our channel-specific layers on ImageNet-100 and ImageNette classification tasks and show an increase of up to 30% accuracy on corrupted data in the top1 accuracy. Further, we conduct a set of designed experiments to qualify the conditions for our findings. Our main result is that a data- and network-dependent linear subspace carries the most important classification information (the *essential*), which our proposed pre-processing layer approximately identifies for most corruptions, and at very low cost.

## 1 Introduction

Intensive research into DNN architectures (He et al., 2016; Szegedy et al., 2015; Tan & Le, 2019; Liu et al., 2022d), improved for example by Neural Architecture Search (NAS) (Dosovitskiy et al., 2021; Tan & Le, 2019) and advanced training schemes (Touvron et al., 2021; Chen et al., 2023b), has produced impressive classification results (Dosovitskiy et al., 2021; Foret et al., 2021). The performance of models on well-known benchmarks such as ImageNet (Russakovsky et al., 2015), Cifar-100 (Krizhevsky & Hinton, 2009) and others has improved significantly over the last decade. However, a persistent challenge arises when these systems are exposed to adverse conditions, such as changes in lighting, weather and other optical corruptions (Hendrycks & Dietterich, 2019; Müller et al., 2023). Despite achieving high accuracy on in-domain data, DNNs often experience a significant drop in performance under these real-world challenges (Müller et al., 2023; Hendrycks

et al., 2020; Sakaridis et al., 2021). Intensive research is therefore being carried out to increase model robustness to various disturbances.

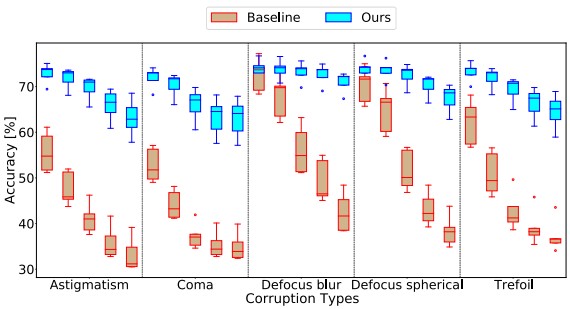

Figure 1: A pre-pended large kernel convolution layer (ours) can increase the robustness of classification networks against unknown corruptions *without additional data augmentation*. Here, we show ResNet50 improved with a trainable pre-pended filter evaluated on ImageNette (Howard, 2023) blur and corruptions from OpticsBench (Müller et al., 2023). For each corruption type, five levels of severity are shown from left to right. The variation, visualized via the box plots, results from five different seeds per model.

A common strategy to increase robustness is to appropriately augment the original training dataset with relevant diversification. Such data augmentation techniques include geometric transformations, cutouts and mixing of images (Yun et al., 2019), color jitter or the simulation of out-of-distribution data by introducing common corruptions (Hendrycks & Dietterich, 2019), and optical corruptions (Müller et al., 2023). Other methods involve adversarial training, whose objective is yet at odds with robustness to some real-world corruption types (Yin et al., 2019).

In this paper, we propose a very simple yet effective trick to improve model generalization, which consists of pre-pending to the model a single large kernel depth-wise convolution operation without strides. The proposed layer is trained with the model and can, in principle, learn a complete representation of the image (*e.g.* with an identity mapping), but no over-complete one. This is in contrast to the usual first model layers that create over-complete representations to facilitate sparse coding. Surprisingly, we find that our simple input layer trained solely on clean data without particular augmentation strategies increases classification robustness to various corruption types on multiple DNN architectures by up to 9.8% across 21 corruptions and by over 30% on specific severities with less than 2k additional parameters (*e.g.* only 0.008% for ResNet50). See Fig. 1 for an example. We thoroughly investigate this outcome with different methods to learn such large per-color-channel, *i.e.* dimension preserving kernels, and compare to the respective baseline trained without the extra layer.

**Major Findings and Contributions** Our empirical study indicates that the proposed, dimension-preserving large kernel input layer, while being able to learn a complete data representation, tends to learn a subspace projection. As such, it extracts the crucial content from the input training samples, *i.e.* the *essential*, such as to preserve a high accuracy on clean data. The dropping of non-essential parts of the signal, *i.e.* the learned subspace projection, automatically leads to an increase in the model's generalization ability without requiring for any corruption specific data augmentation. We show this on ImageNette (Howard, 2023) and ImageNet-100 (Tian et al., 2020) for the corruption types from Hendrycks & Dietterich (2019) and Müller et al. (2023) across diverse image classification models. Further, we explore in a signal theory-inspired study the properties of suitable kernels for the proposed layer, such as to gain deeper understanding and foster future progress in this very cost efficient direction of improving model generalization.

## 2 Related Work

Model robustness and stability have been discussed under various perspectives. In the following, we first give a brief overview on standard benchmarks for the evaluation of classification robustness under corruptions, then, we summarize related work on model hardening through adversarial training. The proposed method differs significantly from these approaches, as no assumptions of potential threats to the model are made during training. Instead, our approach implicitly encourages the model to learn the relevant signal content while reducing parts of the input data that are less relevant (*i.e.* noise). To contextualize this finding, we also discuss prior work on the interplay between learned frequencies and model robustness, as well as prior art on image resampling for neural networks.

**Image Corruptions and Data Augmentations** To improve classification robustness against corruptions, data augmentation can be used to mimic the diversification of real data. AugMix (Hendrycks et al., 2020) improves robustness to common corruptions, Müller et al. (2023) use optical blur kernels to additionally improve accuracy on primary optical aberrations. Others use learned augmentation policies Cubuk et al. (2018) or add more abstract augmentations with image combinations (Yun et al., 2019; Berthelot et al., 2019; 2020), feature map perturbations (Hendrycks et al., 2021), combinations of feature map perturbations and image augmentation (Erichson et al., 2024), perturbed frequency representations (Yucel et al., 2023), or by adapting/augmenting the style of training images (Li et al., 2023; Hong et al., 2021; Xue et al., 2023; Zhang & Agrawala, 2023) using SotA generative models (Ho et al., 2020). All these methods significantly increase the training time of a model. The robustness they provide is limited to corruptions that are similar to the augmented data. In comparison, our models are trained only on the clean dataset, avoiding i) the guessing of the corruptions, and ii) the computational overhead of an augmented data set, while offering improved generalization ability of the trained model in many settings.

Furthermore, Knowledge Distillation (KD) can be used to distill the robustness from a teacher model to increase adversarial robustness (Goldblum et al., 2020; Zi et al., 2021; Huang et al., 2023; Zhao et al., 2022) or out-of-distribution robustness (Zhou et al., 2023). The importance of data augmentation in KD training is discussed in (Wang et al., 2022). Such approaches are expected to reach very high ranges of robustness, yet they can only be applied when large pre-trained models are available for the considered domain.

**Adversarial Attacks and Training** Corruption benchmarks, *e.g.* Hendrycks & Dietterich (2019); Sakaridis et al. (2021), allow testing the model behavior w.r.t. predetermined corruption types. In contrast, adversarial attacks can add any (usually $\epsilon$-bounded) perturbation. They usually assume Lipschitz continuity of robust models (Goodfellow et al., 2015; Kurakin et al., 2017; Wong et al., 2020; Carlini & Wagner, 2017; Andriushchenko et al., 2020; Ilyas et al., 2018). When used during training, adversarial samples can be employed to harden a model (Goodfellow et al., 2015; Rony et al., 2019; Engstrom et al., 2019; Zhang et al., 2019; Wang et al., 2020; Wu et al., 2020; Zhang et al., 2019), where some strategies involve additional loss terms (Engstrom et al., 2019; Zhang et al., 2019) or training data (Carmon et al., 2019; Sehwag et al., 2021; Gowal et al., 2021) (*e.g.* 1M extra samples generated by Ho et al. (2020) are used in Gowal et al. (2021); Rade & Moosavi-Dezfooli (2021); Rebuffi et al. (2021) for adversarial training on CIFAR-10). While significantly improving model robustness to adversarial samples, the additional training costs are immense. The required compute resources increase *e.g.* by a factor of five to 15 even if the simple strategy of using adversarial samples during training is employed. Further, it has been discussed *e.g.* in Yin et al. (2019); Saikia et al. (2021); Gavrikov et al. (2023) that adversarial training is at odds with model generalization to some real-world corruption types, which we focus on in this work. In contrast to the above discussed methods, our approach only requires a negligible overhead of less than 2k additional parameters and our models are trained using the respective standard training parameters, *i.e.* there are negligible extra-costs, while improving model robustness to various common corruptions.

**Learned Frequencies and Robustness** Previous works have studied the effect of learned frequencies within the shallow and deep layers of neural networks on model robustness (*e.g.* Yin et al. (2019)). In Saikia et al. (2021), it is shown that regularizing a model to learn low frequencies and high frequencies separately can improve robustness to common corruptions. Grabinski et al. (2022b) demonstrated a correlation between aliasing in CNN downsampling layers and their susceptibility to adversarial attacks. Several approaches reduce or remove aliasing in the downsampling operators to improve the learned representations and their robustness (Grabinski et al., 2022a; Li et al., 2021; Zhang, 2019; Hossain et al., 2023; Zou et al., 2020). Further, Geirhos et al. (2018) showed that CNNs tend to focus on image textures rather than shapes to determine an object class and Gavrikov et al. (2023) discuss how adversarial training can shift this bias towards shapes with both positive and negative effects on model robustness to common corruptions, depending on the corruption type. In contrast to these works, our approach solely focuses on the first model layer and no specific bias towards high or low frequencies is added. Yet, by providing only a single large kernel where the number of output channels equals the number of inputs, we implicitly encourage the model to focus on the essential part of the data.

**Large Kernel CNNs in the "Era of ViTs"** Vision Transformers (Dosovitskiy et al., 2021) (ViT) trained on huge amounts of data have recently been outperforming classical small kernel CNNs (He et al., 2016)

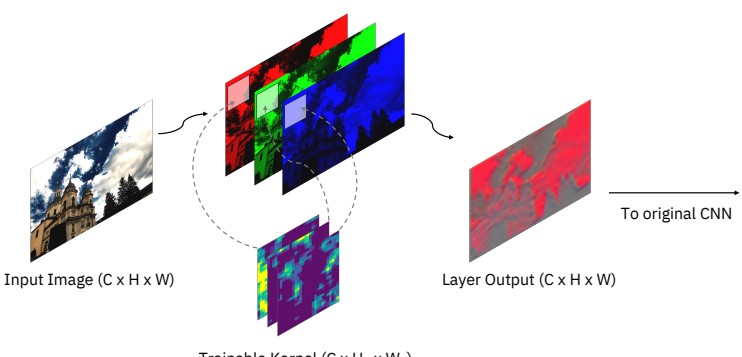

Figure 2: The architecture of our proposed trainable input layer. We learn a single depth-wise convolution to encourage the model to represent the *essential*, *i.e.* the part of the input data that is crucial for classification.

on standard benchmarks, causing the community to shift its focus towards further improving transformer based architectures (Yu et al., 2022; Chen et al., 2023a; Zhai et al., 2022). Contrarily, this has led to a trend towards increasing the filter size within deep layers of CNNs, *e.g.*, it was shown that even kernels with 7×7 convolutions in CNNs can allow them to outperform (Liu et al., 2022d) self-attention based vision transformers (Dosovitskiy et al., 2021; Liu et al., 2022b). Extending on Tolstikhin et al. (2021), Smith et al. (2023) very recently show that CNNs perform on par with vision transformers at scale. Our approach is therefore mainly evaluated on CNNs, because they reach high accuracies even when trained on rather low amounts of data, which facilitates our in-depth study. We show that our findings generalize to transformer models (Dosovitskiy et al., 2021; Liu et al., 2022b) in Sec. D.1 in the appendix.

In Ding et al. (2022); Liu et al. (2022a); Peng et al. (2017), the concept of large kernel CNNs is expanded with kernels up to sizes of 51×51 within the network, where handling the memory consumption is a challenge. Global filter networks (Rao et al., 2021) apply the filter in the frequency domain to allow for infinitely extended filters. These cases further highlight the benefit of using large kernels within the network. While we are also using a large kernel, our approach is different from the above: We use this filter only for the model input and perform an image-to-image mapping with it, encouraging the model to summarize the important information in the data, so that the entire model remains light-weight and can be trained with a low amount of data - yet improves robustness.

**Learning CNN Inputs** Several works have proposed to resample data in a non-uniform way at the model input or in deep layers to allow for precise predictions in regions of interest, *e.g.* Ziwen et al. (2023); Hesse et al. (2023); Thavamani et al. (2021); Jin et al. (2022). Such approaches aim for individual downsampling patterns for every image. In contrast, our approach treats all images equally and operates under the stationarity assumption, *i.e.* the applied convolution filter is constant over the entire image. Since downsampling can lead to aliasing, other works propose to learn to uniformly downsample so that more information is preserved (Talebi & Milanfar, 2021; Marin et al., 2019; Tu et al., 2023). They all aim to improve the model's prediction accuracy. In particular, Talebi & Milanfar (2021) propose to optimize a small deep neural network for the downsampling task, where the output of the network is restricted to be an image. Similarly, the output of our single layer is an image. Yet, our layer does not perform any resampling, avoiding potential aliasing, nor does it provide any non-linearity and can therefore be analyzed using linear techniques.

## 3   Enhancing Prediction Stability with a Trainable Convolution Input Layer

Our aim is to investigate a simple approach to improve a model's stability under corruptions without increasing the training load. Our approach follows the intuition that it is beneficial to encourage a model to learn the relevant information from the data while neglecting superfluous parts of it, *e.g.* noise. To do so, we propose to add an extra convolution layer with a large kernel in front of the input layer of the model, where the output dimensions equal the input dimensions, *i.e.* no over-complete input representation can

be learned. If needed, the model can thus learn to preserve all data. Since, however, not all parts of the training data are valuable for the classification process (*e.g.* noise), we hypothesize that less important parts will likely be dropped in the learned mapping. The architecture, shown in Fig. 2, uses one kernel for each input channel. The input to the layer is the to-be-classified image with $C$ image color channels and a size of $H \times W$. The kernel size of the convolutional input layer is $C \times H_K \times W_K$ with a stride of 1, *i.e.* the result of the depth-wise convolution has the same dimensions, as the input image. In contrast to a typical convolutional layer, this input layer does not fuse the information of the color channels. The layer's output is propagated to the first layer of the otherwise unchanged DNN model, without additional non-linearity applied. Empirically, as shown for example in Fig. 1, our simple approach leads to remarkable results.

Intuitively, the extra layer performs a specific linear transformation that can shift and/or block or emphasize the color-dependent content of an image. The data range of the kernel is not limited to positive numbers, so negative kernel values can also *sharpen* image content. This raises the question, which parts of the input data are preserved in our layer, *i.e.* whether the layer acts, for example, as an amplifier by spatially distributing the information in a better way, or whether it acts as a projection layer, where particular parts of the data are explicitly dropped. In the following, we propose a systematic approach to empirically test these options, by considering three different kernel classes.

**Study Design** The two key characteristics of a linear transformation are a) its rank and b) its conditioning, *i.e.* its noise amplification characteristics in the case of a nominally full-rank transformation. The latter is characterized by the condition number (CN) of the transformation. Let $\cdot * g$ be the linear transformation effected by a convolution with the kernel $g$, and $\cdot * g^{-1}$ its inverse, then

$$CN(\cdot * g) = \frac{|\lambda_{\max}|}{|\lambda_{\min}|} = \frac{1/|\lambda_{\min}|}{1/|\lambda_{\max}|} = CN(\cdot * g^{-1}),$$

where $\lambda_{\min}$ and $\lambda_{\max}$ are the minimum and the maximum eigenvalues of the transformation. The equation shows that the forward kernel and the inverse kernel have the same condition number, which explains our use of the term *noise amplification*. We use the condition number as a numerical indicator of the preservation of signal content[1].

While the linear behavior of our proposed pre-processing layer is well understood, the reaction of the subsequent nonlinear network architecture to this modified input, is not. Our intention is to study the effect of the above-mentioned kernel properties on classification robustness.

We therefore introduce three kernel classes for further studying the properties of the proposed convolutional pre-processing layer, with the underlying hypothesis that signal content preservation or removal is a decisive factor in the observed robustness increase.

| name | CN | rank |
|------|-----|------|
| class I (content preserving) | unity/low | full |
| class II (fully trained/static) | medium | full |
| class III (projection-type) | large/infinite | rank deficient |

Table 1: Categories of kernels with their corresponding condition number (CN) and rank.

**Class I: Content Preserving Kernels** A minimal noise amplifying kernel is one whose associated linear transformation has a determinant of one: such transformation is unitary, *i.e.* vector norms are not changed.

For a convolution kernel, the associated matrix is a circulant matrix with the convolution kernel in the rows (assuming circular boundary conditions). The eigenvectors of circulant matrices form the Fourier basis, which is the underlying reason for the convolution theorem. The associated eigenvalues are the Fourier coefficients. Since the product of the eigenvalues yields the determinant of the linear transformation, we see that all Fourier coefficients must have unit amplitude for the determinant to be unity. Since the Fourier coefficients are complex, they can still have arbitrary phases while fulfilling the unit amplitude constraint. An additional consideration, however, is that the associated kernels be real. This forces the constraint that

---

[1]We emphasize that we are not arguing in an information-theoretic, but in a numerical sense.

$G(-\omega) = G^*(\omega)$ with $G(\omega)$ being the Fourier spectrum, parameterized by angular frequency component $\omega$, of the spatial kernel function $g(x)$. The constraint implies that we have $N/2$ real degrees of freedom to construct $N$ pixel content-preserving kernels, where $N = H_K \times W_K$. We use this parameterization for our practical layer implementation. A unit condition number can only be achieved for kernels of the same size as the image ($225 \times 225$). We also experiment with smaller kernels ($25 \times 25$) of the same construction, also referring to them as content preserving even though the condition number of the associated linear transformation is a low number above unity ($10^2 - 10^3$).

**Class II: Fully Trained Kernels** are parameterized by their real value entries in the spatial domain. Positive and negative values are allowed, but not complex ones. In the absence of the aforementioned constraints of *class I* kernels, we are free to choose the size of the kernels. An ablation study of the kernel size for our proposed convolutional input layer is given in Sec. D.2 in the appendix, indicating that larger kernels can further improve model stability while trading-off accuracy on original data. We choose the kernel size to be $25 \times 25$, which provides a favorable trade-off between both. We have observed that such freely trained kernels yield condition numbers in an intermediate range of $10^4 - 10^5$. For comparison, we also include a number of static kernels in *class II*.

**Class III: Projection-type Kernels** have a CN that is (numerically) infinite, since at least one Fourier coefficient is (numerically) zero. The removed subspace dimension is equivalent to the number of zero Fourier coefficients in the kernel's spectrum. We explore two ways to generate such kernels. First, we explore whether low-value Fourier coefficients in the fully learned kernels of *class II* can be replaced by zeros (thresholding), implying that the low values found by the optimization algorithm are effectively sufficient to remove the subspace in question from the data for all purposes of the nonlinear network part. Second, we encourage zero Fourier coefficients by means of an additional $L1$-regularization on the Fourier coefficients of the fully trained kernel. The associated sparsity then encourages projection-type kernels. The interesting characterizing number for projection-type kernels is the dimensionality of the null-space of the kernel, *i.e.* the number of its zero Fourier coefficients. A larger number indicates a higher rate of signal content removal.

## 4 Experimental Evaluation

In the following, we evaluate the DNN prediction stability with our proposed, trainable image-to-image convolution input layer. The DNNs we evaluate are trained on different publicly available subsets of ImageNet (Russakovsky et al., 2015) to allow for extensive experiments. ImageNette (Howard, 2023) is a dataset consisting of 10 ImageNet classes. It has 9,469 training and 3,925 validation images (Howard, 2023). ImageNet-100 (Tian et al., 2020) uses 100 ImageNet classes with a total of 128k training and 5,000 validation images (Tian et al., 2020). In addition to ImageNette and ImageNet-100, we also incorporate the full ImageNet-1k dataset, ensuring a comprehensive assessment of the effectiveness of our approach across varying scales and complexities (Sec. D.6).

All baseline models and all models with additional convolution input layer are trained from scratch on clean data, *without* additional data augmentation, i.e. following the standard training script. To ensure a comparability between the baseline and our trained models, we only add our proposed input layer to the corresponding model and do not change any hyperparameters. The full details of chosen hy-

| Model | Version | CD | OB | CC |
|---|---|---|---|---|
| ResNet50 He et al. (2016) | Base | *0.800 | *0.592 | *0.487 |
| ResNet50 He et al. (2016) | Trainable | *0.775 | *0.685 | *0.565 |
| AlexNet Krizhevsky et al. | Base | *0.848 | *0.572 | *0.605 |
| AlexNet Krizhevsky et al. | Trainable | *0.838 | *0.670 | *0.660 |
| EfficientNet Tan & Le (2019) | Base | 0.907 | 0.604 | 0.605 |
| EfficientNet Tan & Le (2019) | Trainable | 0.903 | 0.629 | 0.633 |
| MobileNet Howard et al. | Base | 0.897 | 0.589 | 0.564 |
| MobileNet Howard et al. | Trainable | 0.893 | 0.639 | 0.611 |
| DenseNet161 Huang et al. | Base | 0.898 | 0.547 | 0.535 |
| DenseNet161 Huang et al. | Trainable | 0.885 | 0.605 | 0.597 |
| XSEResNext50 He et al. | Base | 0.936 | 0.652 | 0.607 |
| XSEResNext50 He et al. | Trainable | 0.933 | 0.677 | 0.659 |
| ConvNeXt Liu et al. (2022c) | Base | 0.824 | 0.516 | 0.489 |
| ConvNeXt Liu et al. (2022c) | Trainable | 0.796 | 0.565 | 0.539 |
| ViT Dosovitskiy et al. (2021) | Base | 0.812 | 0.525 | 0.511 |
| ViT Dosovitskiy et al. (2021) | Trainable | 0.801 | 0.641 | 0.578 |
| Swin v2 Liu et al. (2022b) | Base | *0.891 | *0.571 | *0.535 |
| Swin v2 Liu et al. (2022b) | Trainable | *0.882 | *0.698 | *0.603 |

Table 2: Top1 Accuracy results on ImageNette for conventionally trained DNNs (Base) and additional fully trainable layer (Trainable). CD = Clean Data, OB = OpticsBench (Müller et al., 2023), CC = Common corruptions (Hendrycks & Dietterich, 2019). * = average from multiple seeds. The results on the corruption benchmarks are averaged across severity and corruption. The highest accuracy per DNN is marked in bold.

perparameters are given in the supplementary material Sec. C. The implementation is based on PyTorch and the training recipes follow maintainers & contributors (2016). In order to draw a more comprehensive picture, some experiments were trained on five different seeds. Subsequently, all models are evaluated on clean data and on two corrupted datasets (Hendrycks & Dietterich, 2019; Müller et al., 2023). The performance of these trainings can be examined in Table 2, 3, and 5. Furthermore, we trained our proposed model adversarially and evaluate it against the baseline in the supplementary material Sec. D.7.

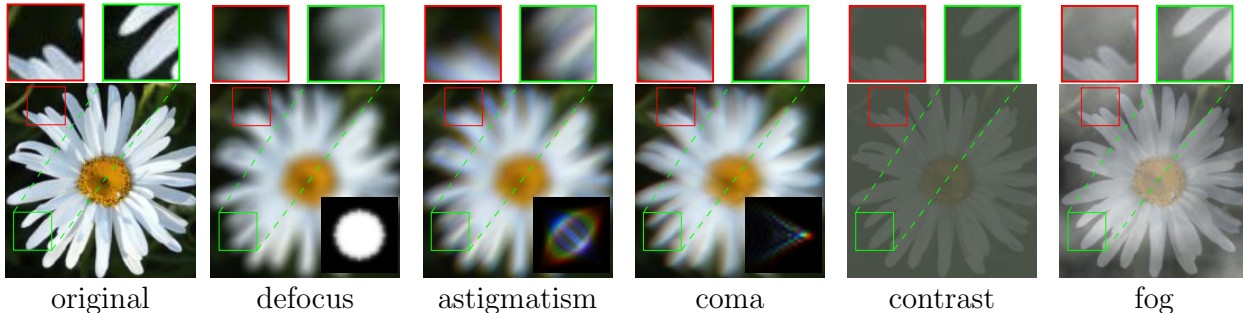

| original | defocus | astigmatism | coma | contrast | fog |

Figure 3: Overview of different corruptions from OpticsBench's primary aberrations (astigmatism, coma) (Müller et al., 2023) and rotationally symmetric defocus blur, contrast and fog from (Hendrycks & Dietterich, 2019) applied to an ImageNet sample. OpticsBench's astigmatism and coma introduce chromatic aberration (visible at the flower petals) and directional blur, which can challenge DNNs differently than rotationally symmetric luminance blur. All corruptions are shown in the supplementary material, Sec. B.

To test against corruptions, we use the common corruptions from Hendrycks & Dietterich (2019), which are each binned into five different severities and apply them to ImageNette (Howard, 2023) and ImageNet-100 (Tian et al., 2020). To test for more diverse blur types, we also include the OpticsBench from Müller et al. (2023), which covers primary optical aberrations and is similarly organized. The blur kernels are size-matched to the defocus-blur-corruption kernels from Hendrycks & Dietterich (2019). Fig. 3 gives a visual impression of some of the corruptions that we evaluate on.

For the sake of readability, we summarize all different corruptions into five super-categories (noise, blur, compression, weather, color) and take the average of all the subcategories. By categorizing the corruptions, we can more effectively highlight the overall trends and impacts observed in our experiments. We provide the full details and figures with all individual corruptions in the supplementary material in Sec. D. This section includes an in-depth breakdown of each type of corruption with the different severities.

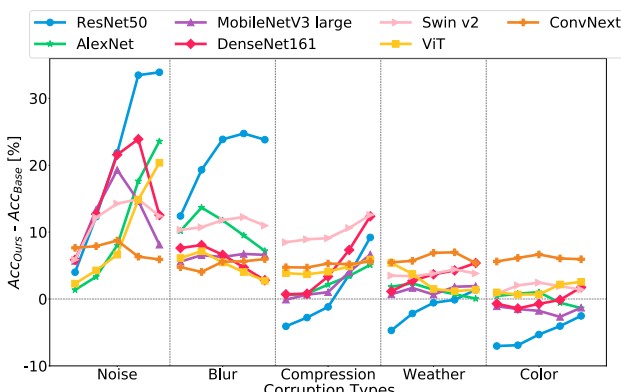

Figure 4: Relative top1 accuracy improvements on ImageNette (Howard, 2023). A fully trainable input layer (class II) can increase the robustness of classification networks against unknown corruptions *without data augmentation.* We evaluate different DNNs on blur and noise corruptions from OpticsBench (Müller et al., 2023) and Common Corruptions (Hendrycks & Dietterich, 2019). For each corruption type, five levels of severity are shown from left to right.

## 4.1 Trainable Large Kernels can Improve Prediction Stability

In the following, we compare the prediction stability under corruption of models with our proposed trainable convolutional pre-processing filter to the respective baselines. First, the *class II* kernels are evaluated on the ImageNette corruptions. In Fig. 4, we plot the performance relative to the baseline for better readability across different model families. Positive values indicate an improvement over the baseline model, and negative

values indicate worse predictions. Absolute accuracies averaged over OpticsBench and Common Corruptions and on clean data are given in Table 2.

Looking at all categories of corruption in Fig. 4, it is noticeable that blur and noise benefit significantly from the additional input filter compared to the baseline. For the other types, the accuracies are on par with the baseline, and in some cases even get slightly below. This is especially true for color corruptions, which is to be expected since our proposed depth-wise convolutions can not learn color re-combinations. Overall, yet in particular for ResNet50, the relative performance of the trainable input layer improves with increasing severity for both noise and blur. As seen in Table 2, the prediction accuracy improves by a large margin on OpticsBench (Müller et al., 2023) and Common Corruptions (Hendrycks & Dietterich, 2019) for all models, *e.g.* 9.3% on OpticsBench and 7.8% on common corruptions for ResNet50, while the accuracy on clean data is only slightly decreased. More recently published models, such as the Vision Transformer (ViT) (Dosovitskiy et al., 2021) and the Swin Transformer v2 (Liu et al., 2022b), outperform the

| Model | Version | CD | OB | CC |
|---|---|---|---|---|
| ResNet50 | Base | **\*0.801** | \*0.536 | \*0.406 |
| ResNet50 | Trainable | \*0.797 | **\*0.558** | **\*0.437** |
| AlexNet | Base | **0.698** | 0.339 | 0.307 |
| AlexNet | Trainable | 0.671 | **0.363** | **0.344** |
| EfficientNet | Base | **0.796** | 0.480 | **0.394** |
| EfficientNet | Trainable | 0.795 | **0.509** | 0.393 |
| MobileNet | Base | **0.780** | 0.470 | 0.344 |
| MobileNet | Trainable | 0.761 | **0.501** | **0.404** |
| ViT | Base | **0.684** | 0.396 | 0.296 |
| ViT | Trainable | 0.677 | **0.437** | **0.331** |
| Swin v2 (tiny) | Base | **0.779** | 0.433 | 0.323 |
| Swin v2 (tiny) | Trainable | 0.774 | **0.476** | **0.379** |

Table 3: Top1 accuracy results on ImageNet-100 for conventionally trained DNNs (Base) and additional fully trainable layer (Trainable). CD= Clean Data, OB = OpticsBench (Müller et al., 2023), CC = Common corruptions (Hendrycks & Dietterich, 2019). * = multiple seeds. The results onOB and CC are averaged across severity and corruption.

baseline on corrupted images by a higher margin when combined with our trainable input layer. Swin v2 with the additional input filter increases the performance by 12.7% on OpticsBench and 6.8% on Common Corruptions, while only slightly lacking accuracy on the clean data of about 0.9%. A more detailed evaluation of these models can be found in the supplementary material Sec. D.1. A model, which is specially designed for small ImageNet subsets, such as the XSEResNext50 (Howard, 2023), is also able to improve its performance on corrupted images despite its high baseline accuracy, suggesting that incorporating the latest training techniques to improve model performance does not undercut the benefits of the proposed method.

| Model | Version | CD | OB | CC |
|---|---|---|---|---|
| ResNet50 | Base | **0.781** | 0.482 | 0.393 |
| ResNet50 | Fully trainable | 0.774 | **0.509** | **0.425** |
| Swin v2 (tiny) | Base | **0.788** | 0.395 | 0.292 |
| Swin v2 (tiny) | Fully trainable | 0.771 | **0.426** | **0.326** |
| Swin v2 (base) | Base | **0.783** | 0.423 | 0.320 |
| Swin v2 (base) | Fully trainable | **0.783** | **0.459** | **0.355** |

Table 4: Results on ImageNet-1k for ResNet50, MobileNet v3 large, and Swin Transformer v2 (tiny and base). CD=Clean Data, OB = OpticsBench, CC=Common Corruptions. The results on OB and CC are averaged across severity and corruption.

Second, the same experiment is performed on ImageNet-100 with the same types of corruption. Table 3 lists the achieved accuracies on clean data, OpticsBench (Müller et al., 2023) and common corruptions (Hendrycks & Dietterich, 2019). The trainable layer improves again on average over the corruptions for each DNN. However, compared to the results on ImageNette, the improvements are now smaller. ResNet50 with the trainable large input kernel performs best with an increase in accuracy of +2.2% on OpticsBench and +3.1% on common corruptions. More results on ImageNet-100 are given in the supplementary material Sec. D.6.

On the ImageNet-1k dataset (Russakovsky et al., 2015), the performance differences between the baselines and our proposed input layer models are slightly lower than on ImageNet-100. However, for higher severities our models significantly outperform the baseline. Especially the transformer-based models benefit from the new input layer. These effects are visualized in the supplementary material Fig. 47 to 50. The performance on clean data is just slightly in favor of the baseline while evaluating on larger datasets as in Table 4. The larger version of the transformer-based Swin v2 is even on par with the baseline on the clean data, while outperforming the baseline on both corrupted evaluation datasets by 3.6% and 3.5%.

In the following, we investigate with help of our proposed kernel *classes I* and *III* which properties the learned kernels have. Understanding these properties is particularly intriguing because they allow the kernels to represent the input data without an increase in dimension, while the subsequent networks generalize better to many unseen corruption types.

## 4.2 Which Properties of Trainable Kernels can help? – Comparing Kernel Classes

The results in Fig. 4, Table 2 and 3 show an improvement in prediction over many corruption types. This raises the question of what kernel properties cause these results, and whether they can be improved further with different kernels. In order to deepen the considerations from Sec. 3, we analyze numerous variants of the input layer kernel for a ResNet50 and compare the different kernel *classes I-III*. The results for more models are given in the supplementary material Sec. D. Table 5 lists the absolute accuracies.

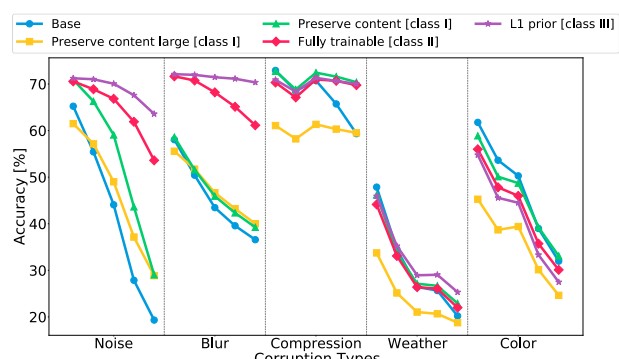

Figure 5: Comparison of different kernel types on corrupted ImageNette data for ResNet50. For additional kernels, see Table 5.

We visualize in Fig. 5 the results on ImageNette for ResNet50 in an absolute fashion to compare the impact of the different kernel classes on corruption accuracy. We plot the baseline and the fully trainable layer (*class II*) together with *class I* and *III* kernel models. The content preserving kernels represent *class I* and the *L1* prior represents *class III*. Except for color corruptions, both the *L1* prior model and the fully trainable kernel model help to stabilize the predictions of the baseline model, while the content preserving model helps only for noise and blur at high severities.

| Kernel type | class | CD | OB | CC |
|---|---|---|---|---|
| None (Base) | - | *__0.800__ | *0.592 | *0.487 |
| Preserve content | I | 0.754 | 0.476 | 0.513 |
| Preserve content large | I | 0.645 | 0.478 | 0.440 |
| Conv2D KS=25 | II | 0.655 | 0.601 | 0.501 |
| Fully trainable | II | *_0.775_ | *_0.685_ | *_0.565_ |
| Random initialization | II | 0.711 | 0.673 | 0.550 |
| L1 prior | III | 0.712 | __0.699__ | __0.567__ |
| Directional blur filter | II | 0.768 | 0.437 | 0.345 |
| Gauss blur filter | II | 0.764 | 0.668 | 0.541 |

Table 5: Top1 accuracy results on ImageNette for ResNet50 and different input layer large kernel types. CD = Clean Data, OB = OpticsBench (Müller et al., 2023), CC = Common corruptions (Hendrycks & Dietterich, 2019). * = average from multiple seeds. The results on the two corruption benchmarks are averaged across severity and corruption. Bold: best model, underline: second best.

Interestingly, the *L1* prior model produces the most favorable results for noise and blur corruptions, followed by the fully trainable kernel, and achieves an accuracy gain of more than 40% at noise severities 4 and 5. The predictions for blur with the *L1* prior model remain almost constant at 72% accuracy, while the baseline accuracy drops below 40% with increasing severity. The fully trainable kernel model largely stabilizes the predictions, but drops by around 10% at higher severities compared to the *L1* prior model. The compression corruption type is more challenging for all models compared to the baseline, while the fully trainable kernel and the *L1* prior models perform similarly and increase in accuracy from severity 3. Interestingly, the baseline also performs quite well here, which may be due to compression artifacts within the original training data.

The content preserving kernel model performs significantly worse than the baseline, while the *L1* prior and the fully trainable kernels tend to slightly increase prediction stability towards higher severities even for hard corruptions (*e.g.* weather). While the *L1* kernel yields rather high robustness, it is significantly underperforming on clean data.

In summary, the *L1* prior and the fully trainable versions seem to follow a *similar pattern at similar levels of corruption severities* in Fig. 5 and largely increase the accuracy compared to the baseline for most corruptions and severities. Yet, on clean data (*e.g.* in Table 5) the *L1* prior leads to significantly lower accuracy (-6.8%

w.r.t. the fully trainable filter). In contrast, the content preserving kernel model in Fig. 5 follows a similar trend to the baseline for noise and blur and has the worst performance for all corruptions.

This analysis indicates that for many corruption types, it is beneficial to explicitly encourage the convolutional image-to-image layer to project the input image onto a subspace in which certain spatial frequencies are not represented. The un-regularized fully trainable layer seems to do this implicitly, while preserving the essential information such as to perform well on clean data.

To further deepen our understanding, we present more experiments with other kernels from the three classes in addition to the kernel variants shown in Fig. 5. First, different sizes of *class I* kernels are compared to see the trade-off between content-preservation and feature locality. The subsequent group analyses different aspects of trainable *class II* filters. The last group compares different *class III* filters to further study the assumed subspace projection.

**Content Preserving Kernels (Class I):** To have a fully content preserving filter, the kernel needs to have the same size as the to-be-convolved images. Thus, we experiment with two sizes of *class I* kernels. The first kernel has the same size as the input images ($225 \times 225$). To also be able to compare the *class I* filters with other classes, we designed and trained a $25 \times 25$ "content preserving" kernel, *i.e.* a filter that would be content preserving for $25 \times 25$ patches. The accuracy of both kernels is given in Table 5. The larger kernel performs significantly worse on the clean data as well as on the common corruptions. Moreover, in the OpticsBench dataset and therefore also in the blur corruptions in Fig. 5, both kernel sizes perform comparably poorly. Filters that purely re-arrange content, whether they preserve locality or not, do not lead to an increase in prediction stability under corruptions.

**Fully Trainable Kernels (Class II):** We perform two additional experiments: one which replaces our trainable color-dependent (depth-wise) convolution layer with a standard convolution layer of the same kernel size ($25 \times 25 \times 3$). The other uses the proposed depth-wise convolution layer, but with random kernel initialization, which validates the benefit of our fully trainable kernel. Both kernels are *class II* representations.

The standard convolution layer achieves significantly lower accuracy on clean data and both benchmarks. Re-combining color channels provides the pre-filtering with more capacity and the ability to better overfit the training data, *i.e.* it provides less generalization. Yet, it yields better results than the *class I* content preserving kernel model.

The random initialization model tests initialization of the kernel and achieves comparable results to the fully trainable case for the corrupted data while suffering some loss in peak performance on clean data.

To test the impact of the trainability of our model, we also evaluate two non-trainable blur filters, one rotational-symmetric Gauss blur filter and a directional (horizontal coma) blur filter obtained from

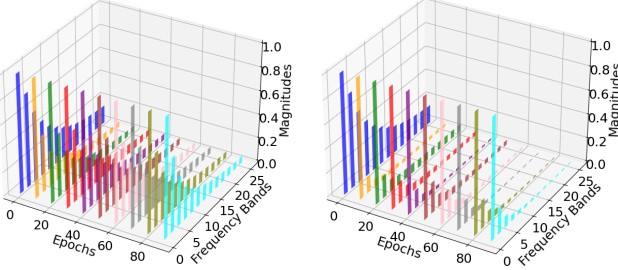

(a) Without Regularization  (b) *L*1 Regularization

Figure 6: Evolution of spectra of (a) Fully Trainable and (b) *L*1 Prior kernels. The bar height indicates the average of the absolute value of the Fourier coefficients in different frequency bands (DC component in the front). Each epoch is normalized separately.

OpticsBench (Müller et al., 2023). These have both lowpass characteristics and remove high frequency content. From the results in Table 5 the two blur filters have an in-domain accuracy comparable to the fully trainable model. Only the Gaussian blur performs better than the baseline on the two corruption benchmarks, while the directional blur model performs worse. Both perform substantially worse than the fully trained filter.

**Projection-type Kernels (Class III):** The *class III* kernel (*L*1 prior) in Table 5, tends to reduce the frequencies in the trainable layer via a Lasso regression on the frequency domain of the kernel. This can also be visualized in Fig. 6 (b) where the kernel learns to discard higher frequencies. From Fig. 6 (a), it is evident that the optimization reduces the Fourier coefficients of higher frequencies of freely learned *class II* kernels as well. To check whether numerically small frequency coefficients contribute to the networks' outputs, we

use our trained models and remove the low magnitude frequencies without retraining. With this approach, we were able to show that even after removing over 80 % of the frequency information (not necessarily high-frequencies), the model's performance is stable. The results over multiple removal intervals with our models can be examined in Fig. 7.

With the insights of Fig. 7 and the increments in performance while training with an $L1$ prior on the frequency domain of the trainable kernel, we conclude that low values in the fully trainable *class II* kernels are numerically zero, *i.e.* they effectively implement projection-type kernels.

### 4.3 Comparison with Augmentation and Joint Trainable Large Kernel and Augmentation

Data augmentation is the de-facto standard for robustifying deep learning models. It comes at the cost of a) an increased training data set and therefore enhanced training times, and b) a model for the expected corruptions must be known or guessed. Since our proposed technique does not rely on these prerequisites, we study how the two approaches compare and whether their combination can yield further benefits.

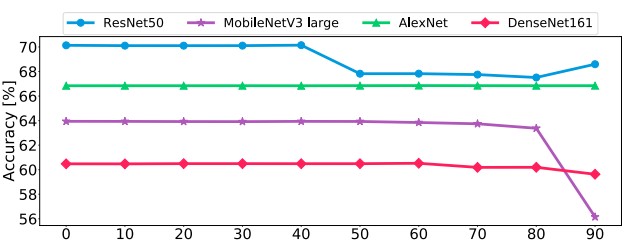

Figure 7: Thresholding the trained convolutional preprocessing kernels in the frequency domain only marginally decreases the accuracy on Optics-Bench (Müller et al., 2023).

We trained multiple models with Aug-Mix (Hendrycks et al., 2020), a generic data augmentation method, with default settings and compare ResNet50 models trained without AugMix (Base), with the same model trained with our proposed input layer, but without Augmix (Trainable), and a model with AugMix in combination with our proposed input layer (Augmix & Trainable). The trained models were evaluated on clean data, OpticsBench (Müller et al., 2023) corrupted data and Common Corruptions (Hendrycks & Dietterich, 2019). The result of these experiments is shown in Table 6.

| Dataset | Version | CD | OB | CC |
|---|---|---|---|---|
| ImageNette | Base | **\*0.800** | \*0.592 | \*0.487 |
| | Trainable (ours) | \*0.775 | \*0.685 | \*0.565 |
| | AugMix | 0.781 | 0.561 | 0.512 |
| | AugMix & Trainable (ours) | 0.795 | **0.774** | **0.639** |
| ImageNet-100 | Base | \*0.801 | \*0.536 | \*0.406 |
| | Trainable (ours) | \*0.797 | \*0.558 | \*0.437 |
| | AugMix | 0.809 | 0.639 | 0.518 |
| | AugMix & Trainable (ours) | **0.814** | **0.663** | **0.533** |

Table 6: Results from AugMix (Hendrycks et al., 2020) training experiments on ImageNette (Howard, 2023) and ImageNet-100 (Tian et al., 2020) datasets. CD= Clean Data, OB = OpticsBench (Müller et al., 2023), CC = Common corruptions (Hendrycks & Dietterich, 2019). * = multiple seeds.

As AugMix works with similar corruption methods, as in Common Corruptions, the results are improved by only using AugMix on both datasets. When trained on ImageNette, using our proposed input layer (Trainable), the improvement outperforms a pure AugMix setting, while on ImageNet-100, the converse is true. However, with either dataset, using AugMix in combination with our proposed input layer (AugMix & Trainable), the performance in both corrupted datasets increases significantly over every other combination. On Imagenet-100 the combination of AugMix and our input layer even outperforms the baseline on clean data.

Table 7 presents a comprehensive comparison of various robustness methods applied to ImageNet-1k using ResNet50. Next to the accuracy on different datasets - Clean Data (CD), OpticsBench (OB), and Common Corruptions (CC) - we compare the costs per epoch (CpE). Our proposed input layer model demonstrates a notable improvement in OB and CC (0.509 and 0.425, respectively) with only a slight increase in CpE. When combined with DeepAugment (DA), our method (*DA & fully trainable*) further enhances performance on OB and CC (0.641 and 0.542, respectively) while maintaining a competitive CpE of 5,083 seconds. Furthermore, this showcases, the ability to combine our proposed input layer with state-of-the-art robustness methods, such as DeepAugment Hendrycks et al. (2021).

# 5 Discussion

Our main findings can be summarized as follows: a simple convolutional pre-processing layer can significantly improve the robustness against unseen corruptions even when trained only on clean data without dedicated augmentation schemes. An analysis of the learned kernels and experiments with different classes of kernels that were designed to explore different levels of preservation of signal content show that projection-type kernels lead to the most robust results in the majority of corruptions while not significantly reducing peak performance and performance in the case of difficult corruptions (weather and color).

| Version | CD | OB | CC | CpE [s] |
|---|---|---|---|---|
| Base | 0.781 | 0.482 | 0.393 | 1,540 |
| Fully trainable (ours) | 0.774 | 0.509 | 0.425 | 1,580 |
| AugMix (Hendrycks et al., 2020) | 0.773 | 0.633 | 0.511 | 11,338 |
| DeepAugment (Hendrycks et al., 2020) | 0.769 | 0.637 | 0.529 | 4,971 |
| DA & Fully trainable (ours) | 0.776 | 0.641 | 0.542 | 5,083 |
| NoisyMix (Erichson et al., 2024) | 0.769 | 0.607 | 0.532 | 7,988 |
| SIN_IN (Geirhos et al., 2018) | 0.750 | 0.537 | 0.457 | 3,059 |
| DAD (Zhou et al., 2023) | 0.802 | 0.502 | 0.495 | 10,352 |

Table 7: Comparisons of different robustness methods on ImageNet-1k with ResNet50. CD=Clean Data, OB = OpticsBench, CC=Common Corruptions, CpE = Costs per epoch in seconds, DA = DeepAugment. The results on OB and CC are averaged across severity and corruption.

This indicates that a removal of signal content can aid the robustness of classification networks against unknown corruptions with the associated benefits of 1) not having to model expected corruptions for an augmentation-style training and 2) a computationally favorable implementation: only $\approx 2000$ additional coefficients are needed and training can be performed on a smaller dataset as compared to an augmentation approach. Besides this, several methods exist, which increase robustness by removing signal content: dropping high frequency wavelet coefficients (Li et al., 2021) allows for robust high-level features. Discarding high frequency content also helps in gaining robustness to common corruptions Grabinski et al. (2022a); Zhang (2019). However, our prepended layer does add negligible extra-costs without having to transform to any co-domain such as a Wavelet or Fourier basis.

We observe the usual trade-off of peak-performance vs. robustness. Our experiments indicate that enforcing sparsity of the frequency content of the proposed convolutional pre-processing layer is an alternative way of achieving robust classification results, which is in line with previous findings (Yin et al., 2019). While this trade-off also indicates that the signal content responsible for successful classification and possible corruptions do not occupy entirely disparate linear subspaces, it appears as if the signal content responsible for successful classification is essentially a linear subspace of the data. Our proposed convolutional pre-processing layer can therefore be interpreted as being an approximation of the responsible linear subspace. Yet, forcing the projection with the sparsity prior on frequencies yields a shift in the trade-off from high peak-performance to higher robustness. In contrast, an unconstrained non-overcomplete layer can learn to represent the essential content without any prior while better preserving the performance on clean data. A discussion of the relationship to sparse coding is given in the supplementary material.

The complementary experiment of designing signal-content preserving kernels yielded no appreciable performance improvement over the baseline, where the baseline, being an identity transformation, can be interpreted as signal-content preserving as well. This is an indication that the null-space of the high-performance kernels carries information that can lead to over-fitting with an associated decrease in robustness of the classification model.

Our proposed layer could, in principle learn a complete representation of the input images (an identity mapping) without increasing the loss in clean accuracy. The remaining question is why the layer actually learns the subspace projection that allows to focus on more essential information and leads to better generalization. One reason could be that the sparse classes, that are the output of classification, can propagate towards the input layer, paired with the spatial inductive bias of the convolution operation itself, which is trained on image data that is heavily correlated, *i.e.* neighboring pixels tend to be similar. The layer might therefore be biased towards the global image structures and learn to represent fine details only where they are needed to perform well on the training data (*e.g.* the essential high frequency details), which yields the observed benefits. It would be interesting to study why this is not occurring in the standard initial layers of the network, without additional regularization (Yin et al., 2019). We hypothesize that this is due to the over-completeness of the representations of most early layers, making it quite likely for a model to represent relevant content as well as noise. In contrast, our simple dimension-preserving layer can at most preserve the input data, and

needs gradient signal from the model loss to learn to do so. It therefore learns to predominantly represent the signal that is *needed* for the task at hand, *i.e.* the *essential*.

## 6 Conclusion & Future Work

We describe a novel, very simple robustifying scheme for classification networks that has the attractive features of being light-weight, both in training and in inference mode, and not requiring knowledge on the corruption model. Furthermore, we showed that the model is compatible with image augmentation. We therefore believe that this simple technique has a large application potential. However, the current paper is only a first step into its analysis. The convolution proposed and studied in this paper is a space-invariant linear transformation, which appears like a sensible choice for classification problems. An open question is whether other problems like semantic segmentation, tracking, etc. can benefit from similar strategies. In this context, it is further unclear whether space-invariance is a desired property or whether more general linear transformations could be beneficial, *e.g.* in yielding closer approximations to the relevant subspace of the signal content. A connected question is how space-variant processing by the follow-up network affects the initially space-invariant processing by our proposed layer.

### Acknowledgments

Syed Muhammad Kazim, Ivo Ihrke and Margret Keuper acknowledge funding from the DFG research unit DFG-FOR 5336 "Learning to Sense".

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
