## A Appendix Overview

In this supplementary material, we provide additional information and details to the experimental results in the main paper. In particular, we provide

- More information and analysis on the employed datasets in section B.

- More details in the experiment setup in section C.

- Additional experimental results in section D, including

  - more details on the aggregated results given in the main paper (D),
  - additionally results on state-of-the-art classification models (D1),
  - an ablation on kernel sizes (D2)
  - an ablation on selected, static kernels (D3)
  - an ablation on thresholded kernel performance over thresholds (D4),
  - an ablation on $L1$ prior regularization strengths (D5),
  - additional details on the results on ImageNet-100 and ImageNet -1k (D6)
  - an evaluation of the robustness against adversarial attacks (D7)
  - experiments on using pretrained models (D8).

- An analysis of the optimized kernels in section E.

- An extended discussion of our findings in section F.

## B Datasets

To train the different models, subsets of the ImageNet (Russakovsky et al., 2015) dataset are used. To have a widespread experiment set, the ImageNette (Howard, 2023) dataset with ten classes and the ImageNet-100 (Tian et al., 2020) with 100 classes from ImageNet are used.

Furthermore, to evaluate the robustness of these models, the subsets are corrupted via different corruptions from OpticsBench (Müller et al., 2023) and Common Corruptions (Hendrycks & Dietterich, 2019).

### B.1 Clean Datasets

ImageNette provides 9,469 training and 3,925 validation images on ten diverse ImageNet classes. We use the ImageNette2 version in all experiments, publicly available from Howard (2023). This allows extensive model training experiments to be carried out with limited computing resources. In addition, some of the experiments were carried out on ImageNet-100, consisting of 100 randomly selected classes from ImageNet-1k with a total of 128k training and 5k validation images. Each class has approximately 1,300 training and 50 validation images. As there are no extra test labels, we test on the validation images. The validation images are kept from training. Finally, two experiments are conducted on ImageNet-1k, which consists of 1,000 classes with a total of 1,281k training and 50k validation images.

### B.2 Common Corruptions

The diverse common corruptions (Hendrycks & Dietterich, 2019) consist of digital, weather, noise and blur corruptions aimed at benchmarking DNNs in safety-critical applications. We use 16 different algorithmically generated corruptions, organized into five different severities. Besides the 15 benchmark corruptions we also include the saturate corruption from Hendrycks & Dietterich (2019) to test for more color corruptions. The individual corruptions are shown in Fig. 8 for severity 4 out of 5. Following Hendrycks & Dietterich (2019) images are saved using light JPEG compression, which means that other corruptions are also lightly corrupted by the JPEG compression.

### B.3 OpticsBench

The recently published OpticsBench (Müller et al., 2023) provides additional blur corruptions obtained from optics, which are obtained by convolving images with a given 3D kernel. The blur kernels (x,y,color) are color dependent and have diverse shapes. They are intended as base classes for primary aberrations and complement the blur corruptions from Common Corruptions (Hendrycks & Dietterich, 2019). Fig. 8 shows the different OpticsBench blur corruptions in the upper left along with the corresponding blur kernel in linear scaling. Coma and astigmatism are shown in one orientation. As the authors use the defocus blur from Hendrycks & Dietterich (2019) as a baseline, we also visualize the corresponding defocus blur kernel in Fig. 8.

### B.4 Grouping Corruption Types

Combining the two corrupted datasets, we evaluated all models on 24 different corruptions, with each consisting of five severities. The high number of corruptions provides a broad and stable view on the robustness of our models against such corruptions. However, to visualize the results of our evaluation, we grouped corruptions in five super-categories: noise, blur, compression, weather, and color.

The five super-categories with the corresponding corruptions can be examined in Fig. 8. As the first super-category, we combined all noise generating corruptions from the Common Corruptions into *noise*. Furthermore, we grouped all blur corruptions from the Common Corruptions (Hendrycks & Dietterich, 2019) with all corruptions from OpticsBench (Müller et al., 2023) into the super-category *blur*, as they all mimic different blur types. Similar to Hendrycks & Dietterich (2019), we combine all weather corruptions into the super-category *weather*. Following the same theme, less drastic image transformations such as brightness, contrast and saturate are grouped into *color*. The last super-category combines image transformations and corruptions into the super-category *compression*.

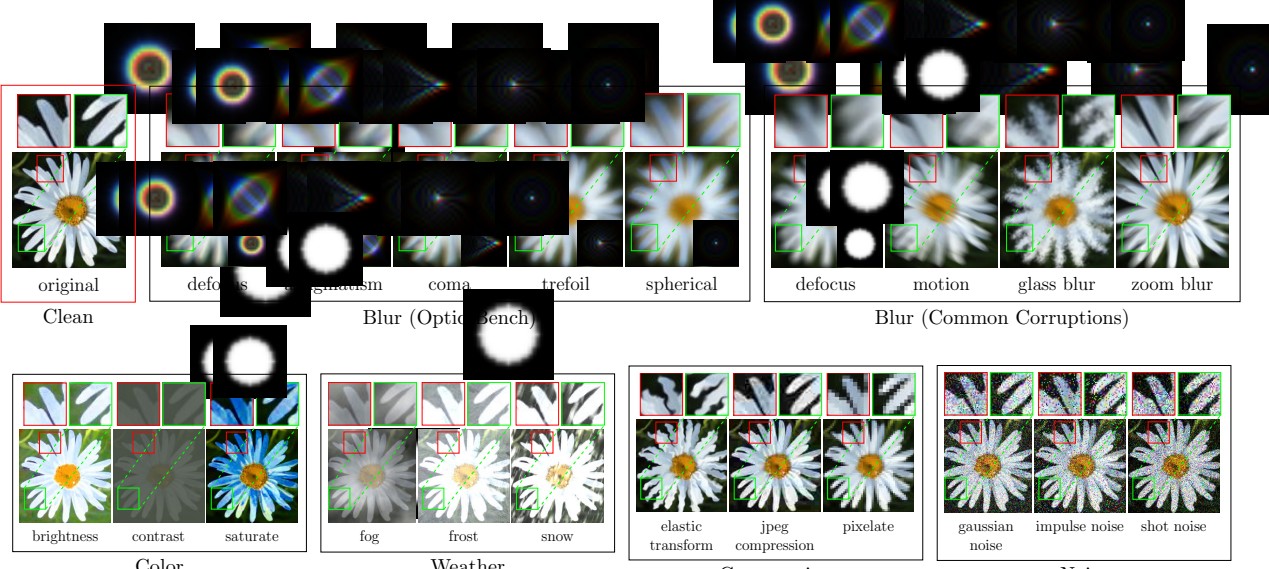

Figure 8: Overview of used corruptions applied to an ImageNet sample. The first row shows the unmodified image (upper left) and various blur corruptions from Müller et al. (2023) (left box) and Hendrycks & Dietterich (2019) (right box). The second row shows the remaining common corruptions from Hendrycks & Dietterich (2019) grouped to color, weather, compression and noise (from left to right). All shown corrupted images represent severity 4 out of 5.

Each of the evaluated corruptions and super-categories have five severities. In addition to the spatial domain visualization in Fig. 8, the super-categories are visualized in the frequency domain, in Fig. 9 to 13. Each of these figures is divided into six columns, in which the first five columns represents one of the super-categories.

For every super-category, the average Fourier transformation magnitude over 100 images is displayed in the top row. The bottom row corresponds to the absolute difference of corrupted average Fourier transformation magnitude against the clean data average Fourier transformation magnitude. The last column represents the average frequency magnitude of 100 clean data images.

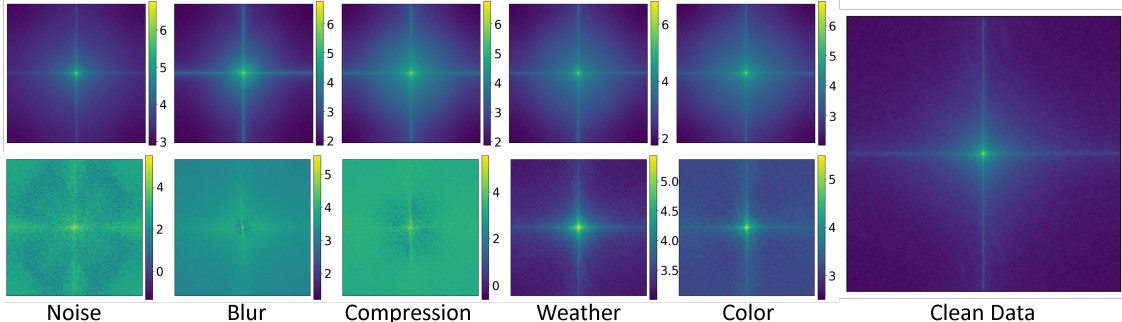

Figure 9: Corruption Severity 1 - Average Fourier transformation magnitude over 100 ImageNet images for each super-category (column 1–5) and the clean data (column 6). Top: The average Fourier transformation magnitude. Bottom: The difference between the corrupted average Fourier transformation magnitude and the clean average Fourier transformation magnitude.

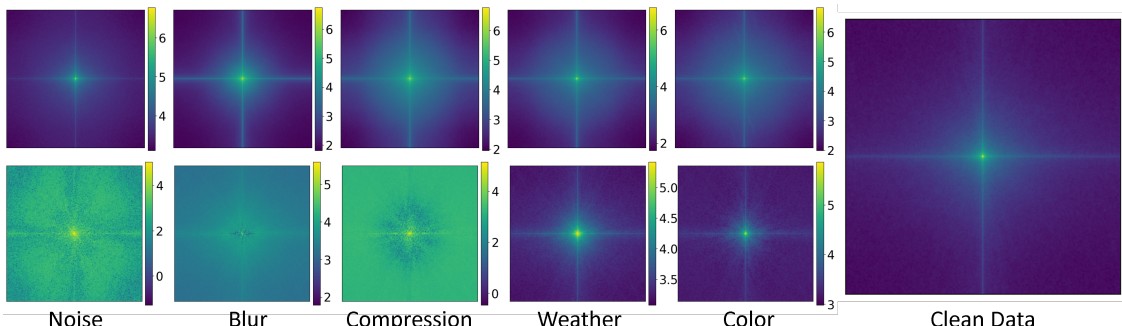

Figure 10: Corruption Severity 2 - Average Fourier transformation magnitude over 100 ImageNet images for each super-category (column 1-5) and the clean data (column 6). Top: The average Fourier transformation magnitude. Bottom: The difference between the corrupted average Fourier transformation magnitude and the clean average Fourier transformation magnitude.

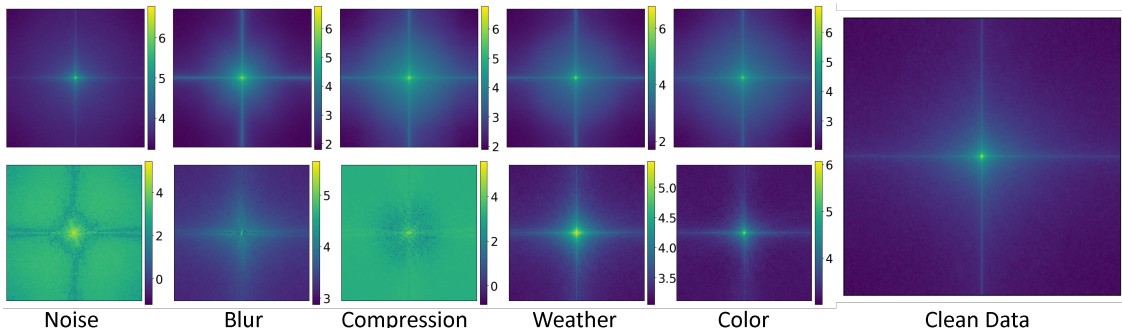

Figure 11: Corruption Severity 3 - Average Fourier transformation magnitude over 100 ImageNet images for each super-category (column 1-5) and the clean data (column 6). Top: The average Fourier transformation magnitude. Bottom: The difference between the corrupted average Fourier transformation magnitude and the clean average Fourier transformation magnitude.

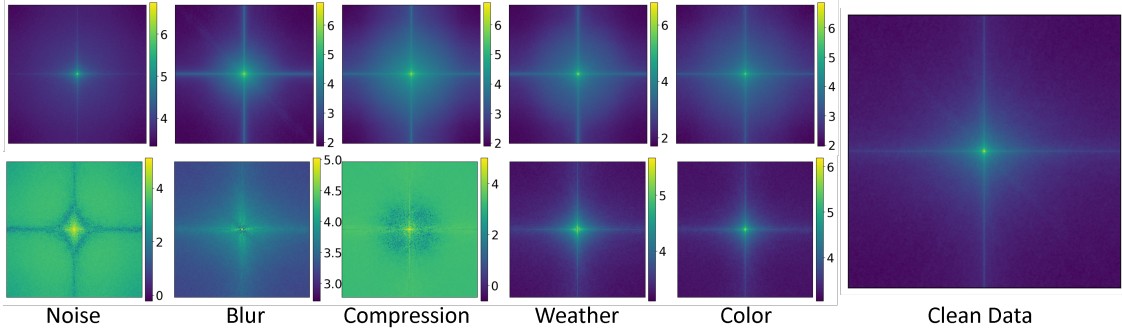

Figure 12: Corruption Severity 4 - Average Fourier transformation magnitude over 100 ImageNet images for each super-category (column 1-5) and the clean data (column 6). Top: The average Fourier transformation magnitude. Bottom: The difference between the corrupted average Fourier transformation magnitude and the clean average Fourier transformation magnitude.

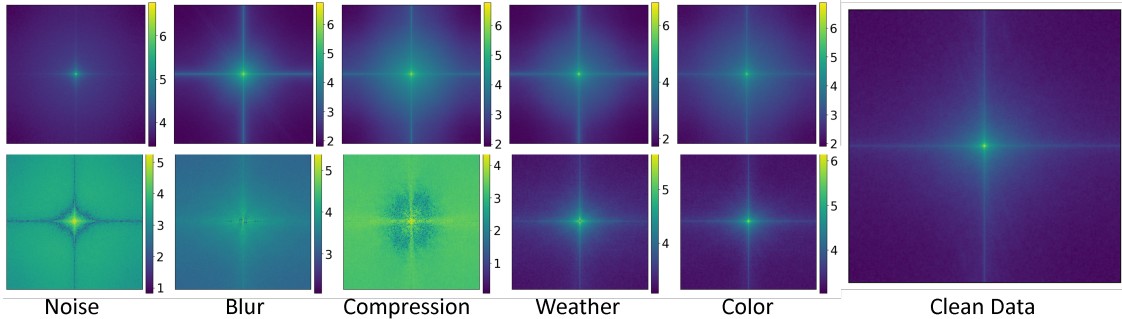

Figure 13: Corruption Severity 5 - Average Fourier transformation magnitude over 100 ImageNet images for each super-category (column 1-5) and the clean data (column 6). Top: The average Fourier transformation magnitude. Bottom: The difference between the corrupted average Fourier transformation magnitude and the clean average Fourier transformation magnitude.

## C  Experiment Setup

All models are trained from scratch on clean data, without corruption specific data augmentation. However, common non-corruption specific data augmentations, such as random horizontal flip and random crop are used during all model trainings. To guarantee a comparability between all models from the same architecture type, we only added our proposed input layer to the corresponding models. Therefore, we did not change any hyperparameters in-between the training of the same architecture type. The hyperparameters for the training are used from maintainers & contributors (2016). The XSEResNext50 model is a specialized ImageNette model, which is on the ImageNette leaderboard Howard (2023). The ViT version in use is the base model, with patch size of 16. The Swin Transformer v2 (tiny) and the larger version Swin Transformer v2 (base) have a window size of 8.

A quick overview over some important model information can be examined in Table 8.

| Model | Version | #Parameters | #Epochs | Training Recipe |
|---|---|---|---|---|
| ResNet50 | Base | 23,528,522 | 100 | (maintainers & contributors, 2016) |
| ResNet50 | Trainable | 23,530,397 | 100 | (maintainers & contributors, 2016) |
| EfficientNet b0 | Base | 4,020,358 | 200 | (maintainers & contributors, 2016) |
| EfficientNet b0 | Trainable | 4,022,233 | 200 | (maintainers & contributors, 2016) |
| MobileNet | Base | 4,214,842 | 150 | (maintainers & contributors, 2016) |
| MobileNet | Trainable | 4,216,717 | 150 | (maintainers & contributors, 2016) |
| DenseNet161 | Base | 26,494,090 | 90 | (maintainers & contributors, 2016) |
| DenseNet161 | Trainable | 26,495,965 | 90 | (maintainers & contributors, 2016) |
| AlexNet | Base | 57,044,810 | 100 | (maintainers & contributors, 2016) |
| AlexNet | Trainable | 57,046,685 | 100 | (maintainers & contributors, 2016) |
| XSEResNext50 | Base | 25,550,618 | 200 | (Howard, 2023) |
| XSEResNext50 | Trainable | 25,552,493 | 200 | (Howard, 2023) |
| ConvNeXt | Base | 49,462,378 | 600 | (maintainers & contributors, 2016) |
| ConvNeXt | Trainable | 49,464,253 | 600 | (maintainers & contributors, 2016) |
| Swin v2 (tiny) | Base | 27,585,844 | 300 | (Liu et al., 2022b) |
| Swin v2 (tiny) | Trainable | 27,587,719 | 300 | (Liu et al., 2022b) |
| Swin v2 (base) | Base | 88,728,418 | 300 | (Liu et al., 2022b) |
| Swin v2 (base) | Trainable | 88,730,293 | 300 | (Liu et al., 2022b) |
| ViT (base) | Base | 85,806,346 | 300 | (maintainers & contributors, 2016) |
| ViT (base) | Trainable | 85,808,221 | 300 | (maintainers & contributors, 2016) |

Table 8: A quick overview over the used model architectures and the corresponding number of parameters and trained epochs per training run.

## D  Additional Experimental Results

To have a more holistic view on the performance of models with our proposed pre-pend layer, multiple additional experiment are conducted. This section contains multiple detailed evaluation results from models trained on the ImageNette (Howard, 2023) dataset. The next subsection (Sec. D.1) demonstrates, that the proposed input layer also improves the performance of transformer-based models on corrupted images. The subsequent subsection (Sec. D.2) indicates the reasoning for the proposed kernel size. Section D.3 presents more experiments conducted with a static filter (class II). In Sec. D.4 and D.5, more experiments can be examined on class III kernels. More ImageNet-100 training results are introduced in Sec. D.6. Additional to the corruption robustness, Sec. D.7 indicates a higher robustness against adversarial attack by adding our proposed trainable pre-processing layer to existing models.

In Fig. 14 the same pattern, as in Fig. 1, can be inspected. The convolutional preprocessing (ours) increases the performance of the AlexNet model on all corruptions and severities of the OpticsBench (Müller et al., 2023) dataset.

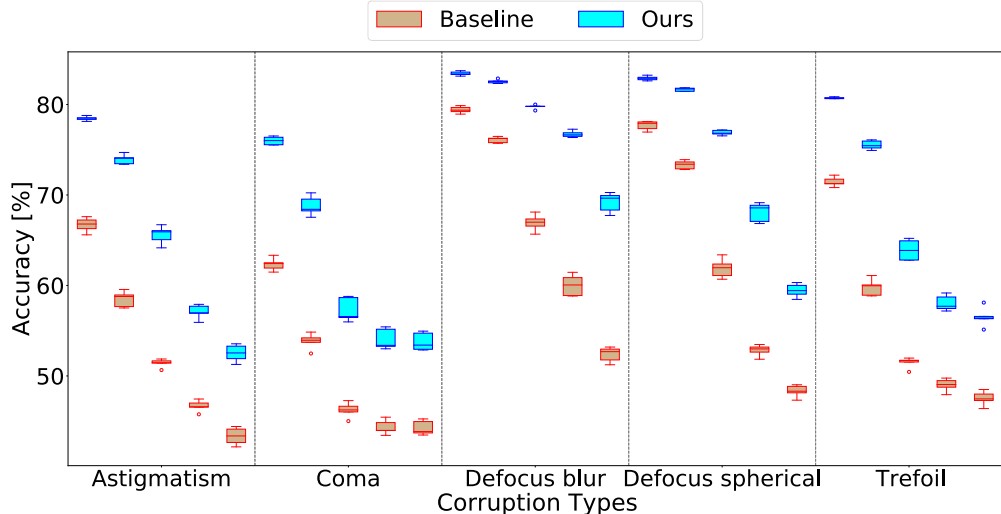

Figure 14: Convolutional preprocessing (ours) can increase the robustness of classification networks against unknown corruptions *without data augmentation*. AlexNet improved with a trainable preprocessing filter evaluated on ImageNette (Howard, 2023) blur and corruptions from OpticsBench (Müller et al., 2023). For each corruption type, five levels of severity are shown from left to right. The variation, visualized via the box plots, results from five different seeds per model.

An increase of accuracy over the most corruptions from Common Corruptions (Hendrycks & Dietterich, 2019) and OpticsBench (Müller et al., 2023), especially on higher severities are displayed in Fig. 16 to 22.

To extend the corruptions which we evaluate on, we also included the 3D Common Corruptions by Kar et al. (2022). The results on these additional corruptions can be examined in Table 9. On all three datasets (ImageNette, ImageNet-100, and ImageNet-1k), our proposed layer is able to outperform the baseline significantly. This is also true for different model architectures.

| Dataset | Model | Verson | CC3D |
|---|---|---|---|
| ImageNette | ResNet50 | Base | 0.509 |
| | ResNet50 | Fully Trainable | **0.665** |
| | AlexNet | Base | 0.655 |
| | AlexNet | Fully Trainable | **0.733** |
| ImageNet-100 | ResNet50 | Base | 0.423 |
| | ResNet50 | Fully Trainable | **0.470** |
| | ConvNeXt | Base | 0.337 |
| | ConvNeXt | Fully Trainable | **0.403** |
| ImageNet-1k | MobileNet | Base | 0.255 |
| | MobileNet | Fully Trainable | **0.271** |
| | Swin v2 (base) | Base | 0.330 |
| | Swin v2 (base) | Fully Trainable | **0.375** |

Table 9: 3D Common Corruptions on ImagenNette (Howard, 2023), ImageNet-100 (Tian et al., 2020), and Imagenet-1k (Russakovsky et al., 2015). 3DCC = 3D Common Corruptions

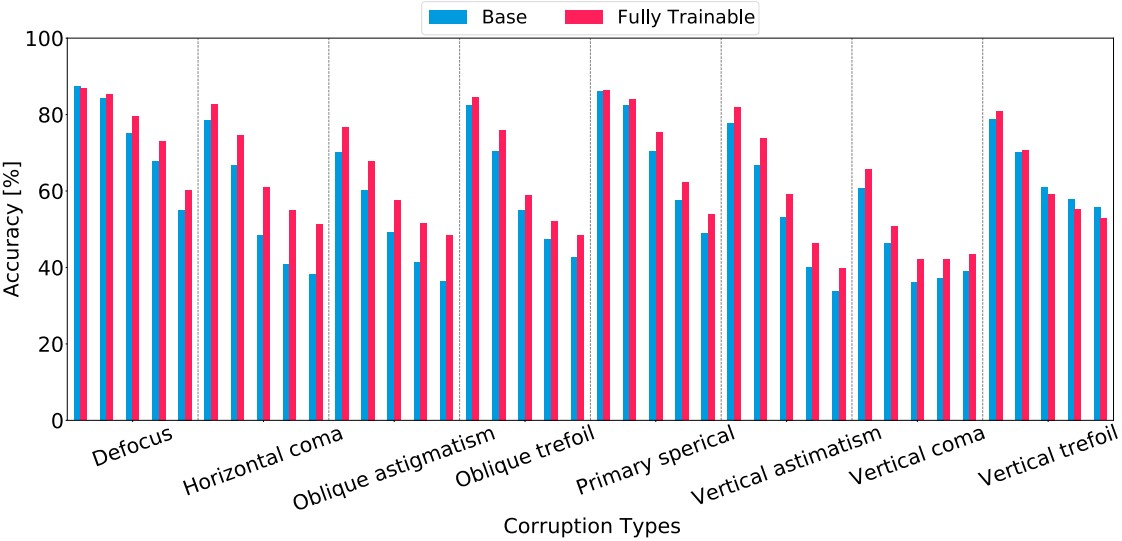

Figure 15: MobileNet v3 large - ImageNette OpticsBench. A comparison of the MobileNet v3 large (Base) and a MobileNet v3 large with our proposed layer (Fully Trainable), on ImageNette OpticsBench.For each corruption type, five levels of severity (Severity 1 to 5) are shown from left to right in each corruption column.

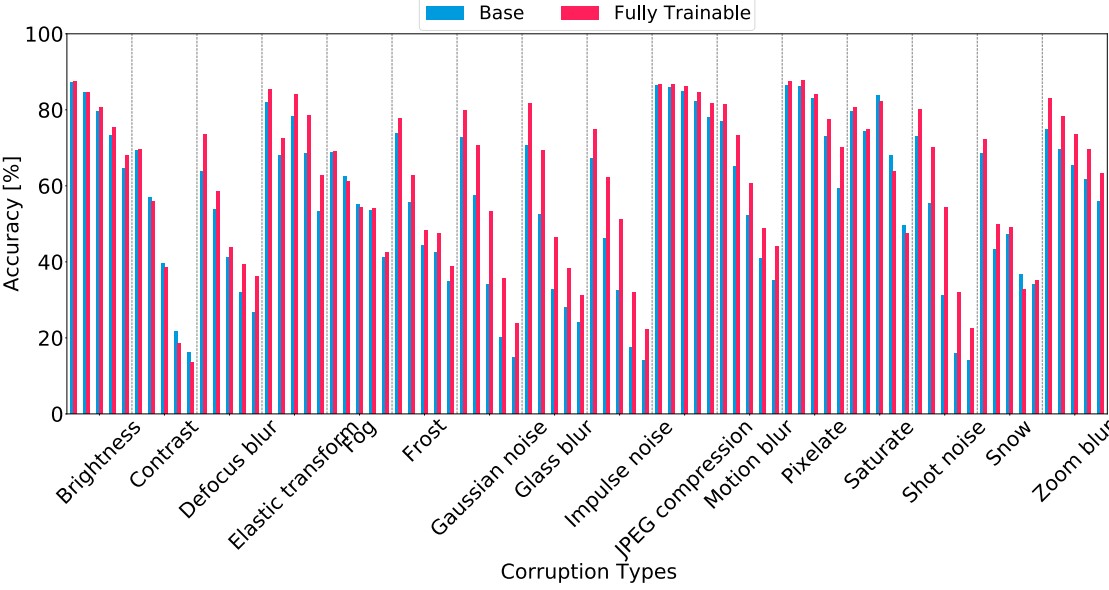

Figure 16: MobileNet v3 large - ImageNette Common Corruptions. A comparison of the MobileNet v3 large (Base) and a MobileNet v3 large with our proposed layer (Fully Trainable), on ImageNette Common Corruptions. For each corruption type, five levels of severity (Severity 1 to 5) are shown from left to right in each corruption column.

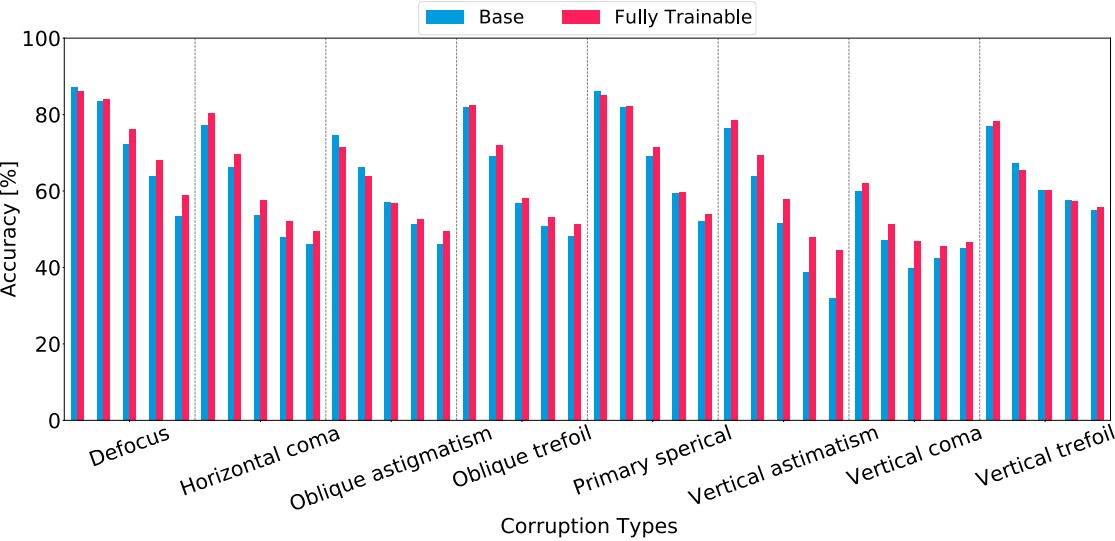

Figure 17: EfficientNet b0 - ImageNette OpticsBench. A comparison of the EfficientNet b0 (Base) and a EfficientNet b0 with our proposed layer (Fully Trainable), on ImageNette OpticsBench. For each corruption type, five levels of severity (Severity 1 to 5) are shown from left to right in each corruption column.

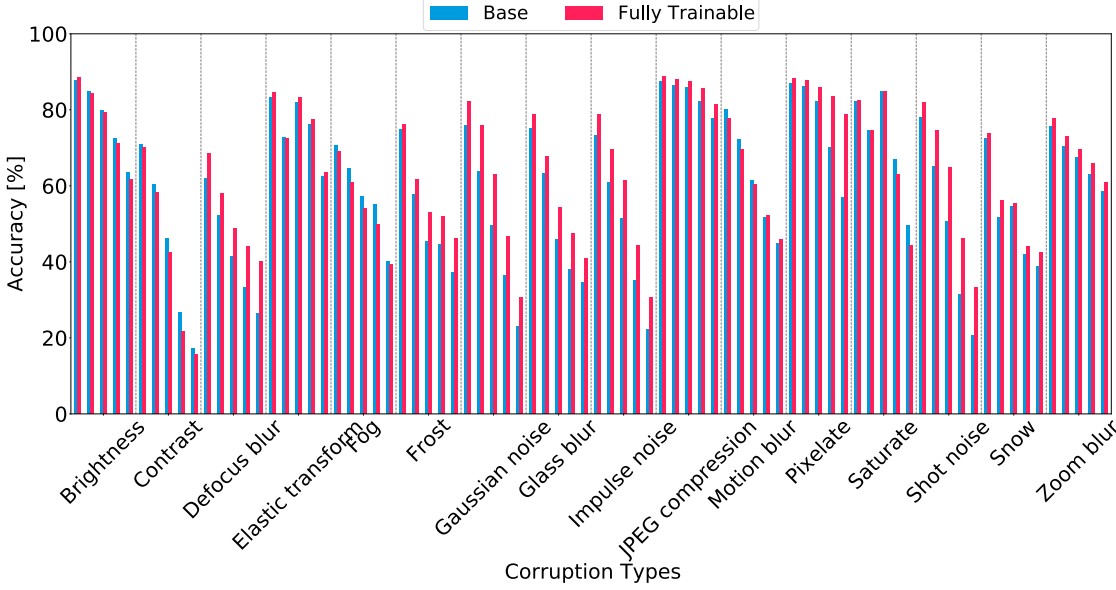

Figure 18: EfficientNet b0 - ImageNette Common Corruptions. A comparison of the EfficientNet b0 (Base) and a EfficientNet b0 with our proposed layer (Fully Trainable), on ImageNette Common Corruptions. For each corruption type, five levels of severity (Severity 1 to 5) are shown from left to right in each corruption column.

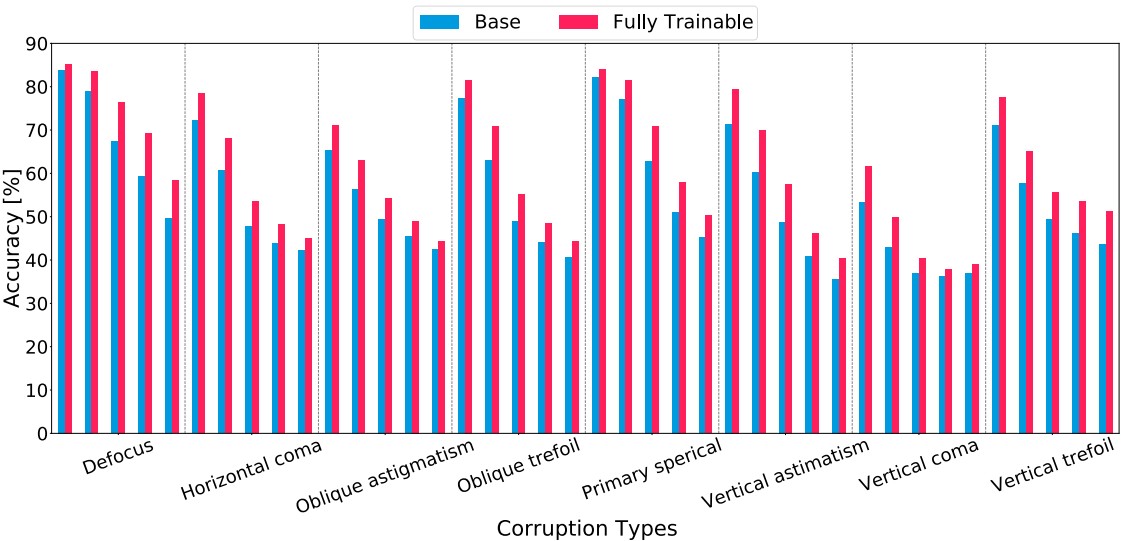

Figure 19: DenseNet161 - ImageNette OpticsBench. A comparison of the DenseNet161 (Base) and a DenseNet161 with our proposed layer (Fully Trainable), on ImageNette OpticsBench. For each corruption type, five levels of severity (Severity 1 to 5) are shown from left to right in each corruption column.

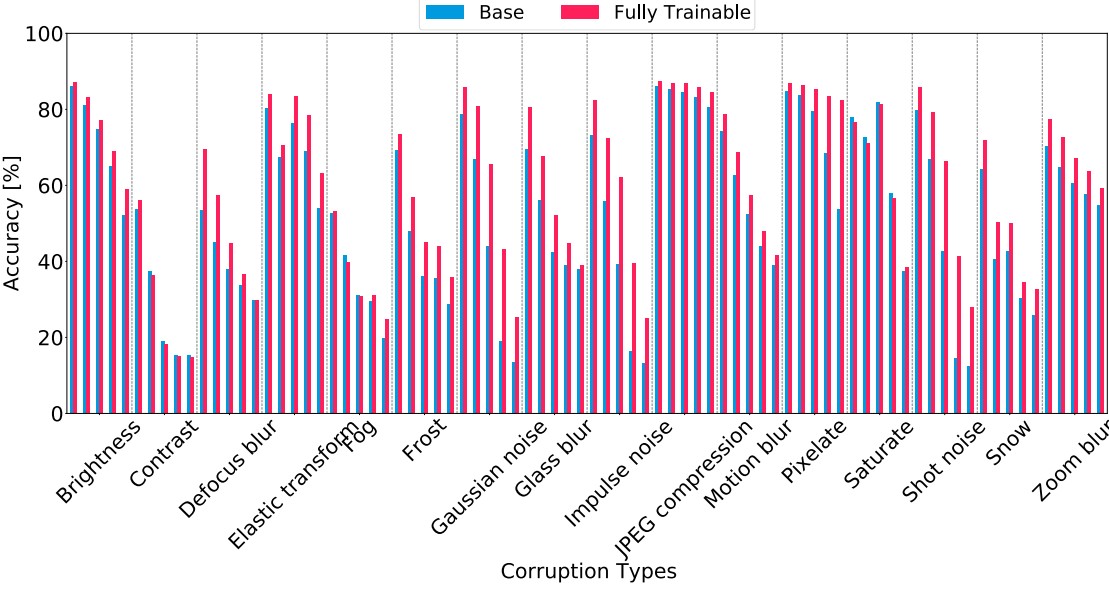

Figure 20: DenseNet161 - ImageNette Common Corruptions. A comparison of the DenseNet161 (Base) and a DenseNet161 with our proposed layer (Fully Trainable), on ImageNette Common Corruptions. For each corruption type, five levels of severity (Severity 1 to 5) are shown from left to right in each corruption column.

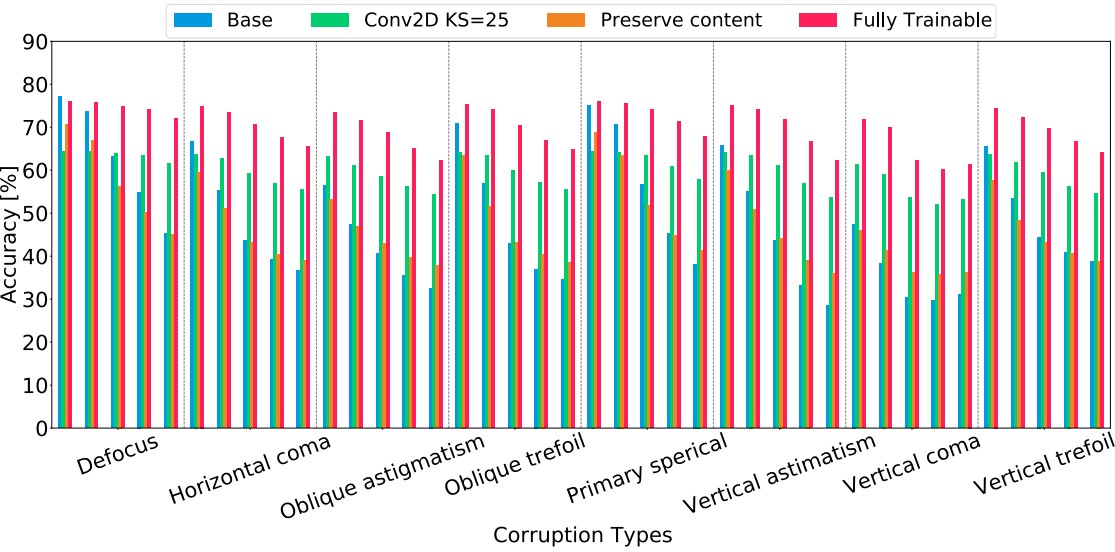

Figure 21: ResNet50 - ImageNette OpticsBench. A comparison of the ResNet50 (Base), the ResNet50 with an additional prior 2D convolutional layer (Conv2D KS=25), a ResNet50 with an additional trainable class I layer (Preserve Content), and a ResNet50 with our proposed layer (Fully Trainable), on ImageNette OpticsBench. For each corruption type, five levels of severity (Severity 1 to 5) are shown from left to right in each corruption column.

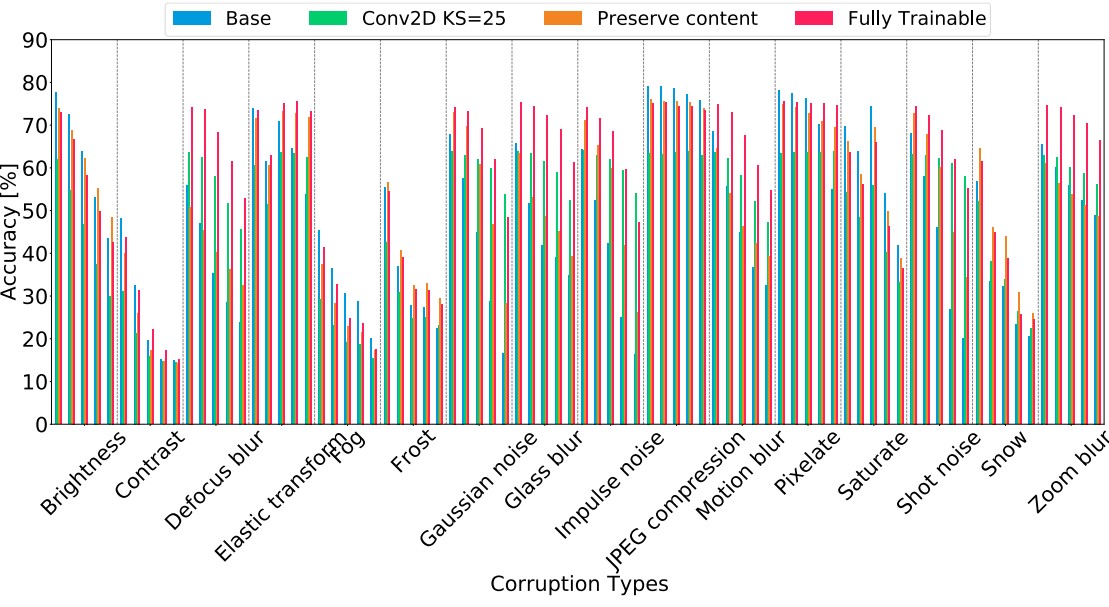

Figure 22: ResNet50 - ImageNette Common Corruptions. A comparison of the ResNet50 (Base), the ResNet50 with an additional prior 2D convolutional layer (Conv2D KS=25), a ResNet50 with an additional trainable class I layer (Preserve Content), and a ResNet50 with our proposed layer (Fully Trainable), on ImageNette Common Corruptions. For each corruption type, five levels of severity (Severity 1 to 5) are shown from left to right in each corruption column.

### D.1  Transformer-based Model Evaluation

In this section of the appendix, we show, that our proposed input layer not only performs well in combination with convolutional models, it also increases the performance of transformer-based models. Therefore, we trained two transformer-based models and compared them to a state-of-the-art convolutional-based model. Fig. 23 highlights that our proposed input layer increases the accuracy of the Vision Transformer base (Dosovitskiy et al., 2021) and the Swin Transformer v2 tiny (Liu et al., 2022b) on corrupted ImageNette data. In comparison to the convolutional-based ConvNeXt small (Liu et al., 2022d) model, both transformer-based models benefit in similar ranges from the added input layer. Furthermore, Fig. 24 to Fig. 30 present more detailed evaluation results of these models on corrpted ImagenNette data.

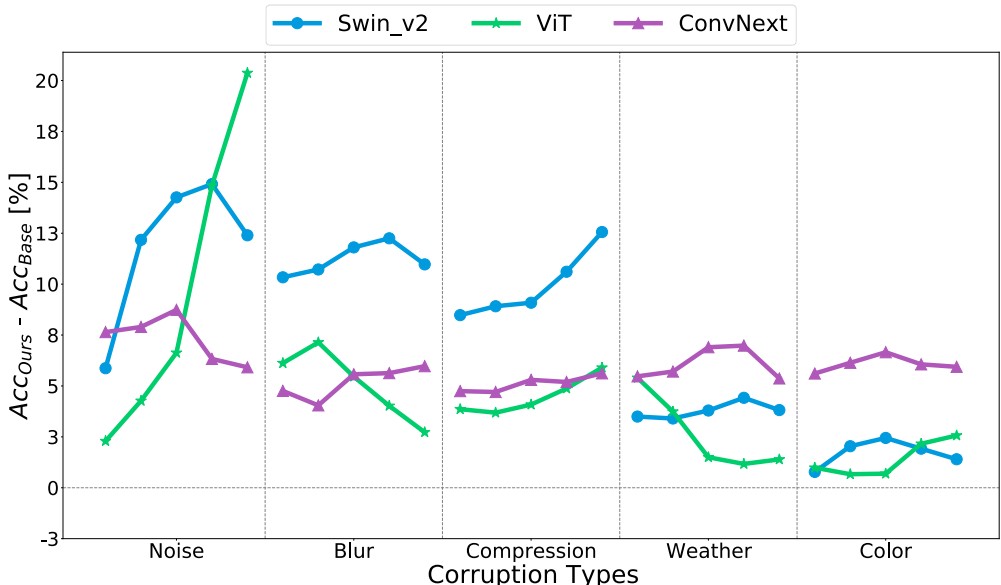

Figure 23: Relative accuracy improvements on ImageNette (Howard, 2023). A fully trainable input layer (class II) can increase the robustness of classification networks against unknown corruptions *without data augmentation.* We evaluate different state-of-the-art DNNs on blur and noise corruptions from Optics-Bench (Müller et al., 2023) and Common Corruptions (Hendrycks & Dietterich, 2019). For each corruption type, five levels of severity are shown from left to right.

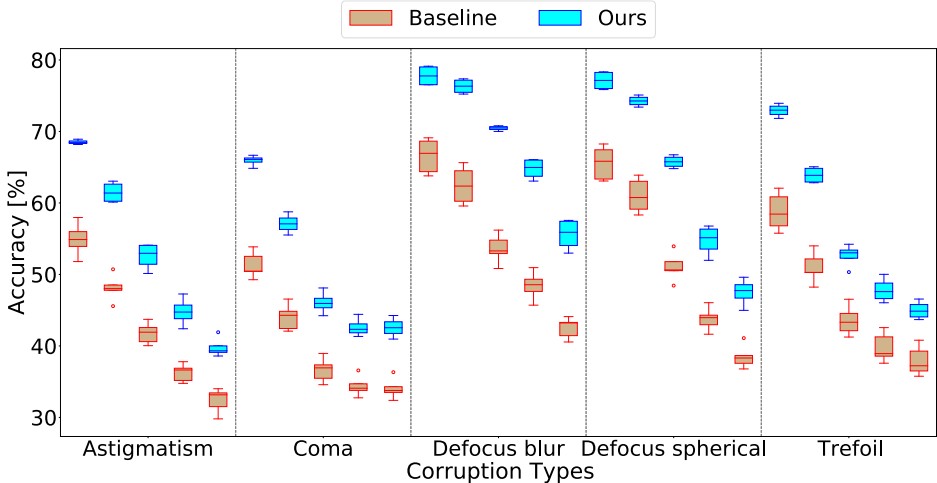

Figure 24: Convolutional preprocessing (ours) can increase the robustness of classification networks against unknown corruptions *without data augmentation*. Swin Transformer v2 tiny (Liu et al., 2022b) improved with a trainable preprocessing filter evaluated on ImageNette (Howard, 2023) blur and corruptions from OpticsBench (Müller et al., 2023). For each corruption type, five levels of severity are shown from left to right. The variation, visualized via the box plots, results from five different seeds per model.

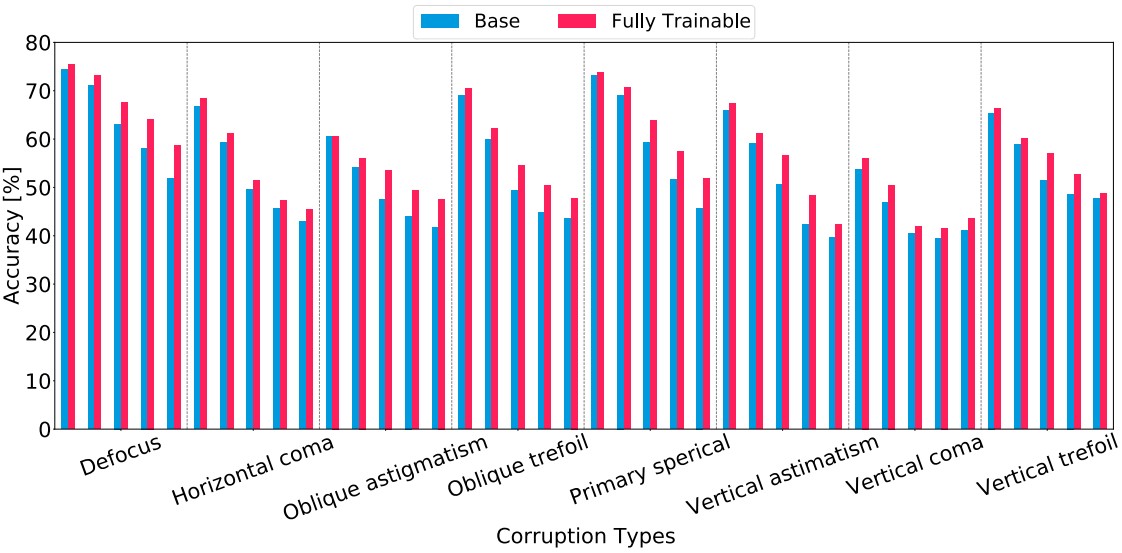

Figure 25: ConvNeXt small - ImageNette OpticsBench. A comparison of the ConvNeXt (Base) and a ConvNeXt with our proposed layer (Fully Trainable), on ImageNette OpticsBench. For each corruption type, five levels of severity (Severity 1 to 5) are shown from left to right in each corruption column.

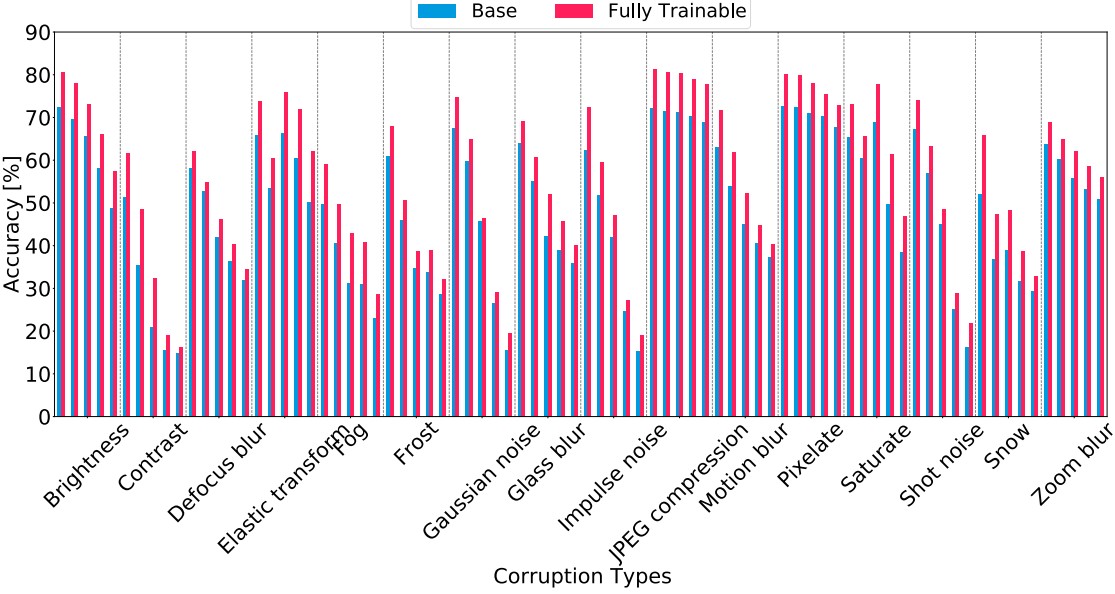

Figure 26: ConvNeXt small - ImageNette Common Corruptions. A comparison of the ConvNeXt (Base) and a ConvNeXt with our proposed layer (Fully Trainable), on ImageNette Common Corruptions. For each corruption type, five levels of severity (Severity 1 to 5) are shown from left to right in each corruption column.

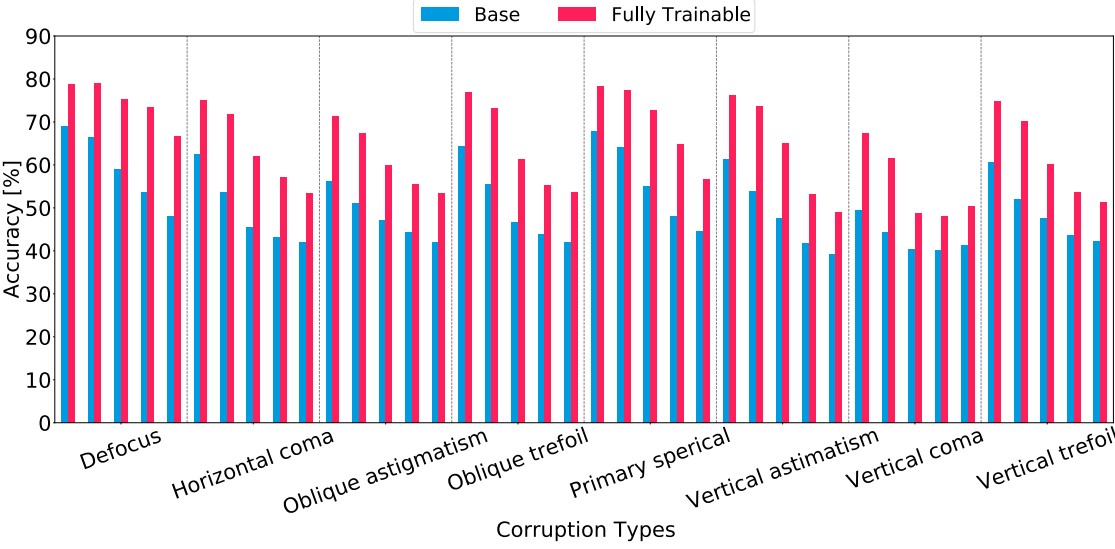

Figure 27: Vision Transformer base - ImageNette OpticsBench. A comparison of the Vision Transformer (Base) and a Vision Transformer with our proposed layer (Fully Trainable), on ImageNette OpticsBench. For each corruption type, five levels of severity (Severity 1 to 5) are shown from left to right in each corruption column.

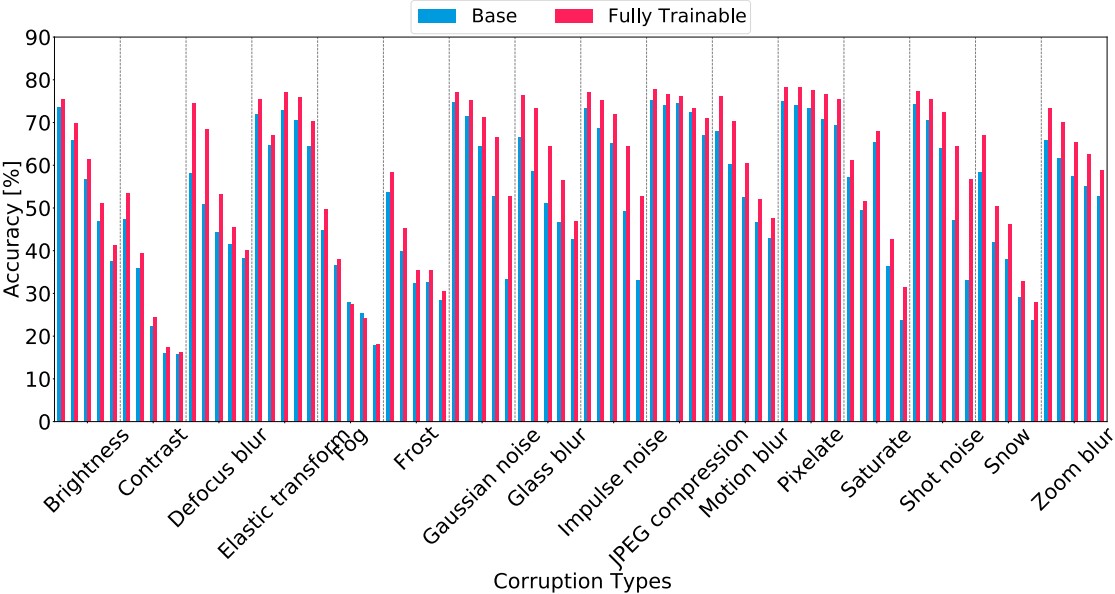

Figure 28: Vision Transformer base - ImageNette Common Corruptions. A comparison of the Vision Transformer (Base) and a Vision Transformer with our proposed layer (Fully Trainable), on ImageNette Common Corruptions. For each corruption type, five levels of severity (Severity 1 to 5) are shown from left to right in each corruption column.

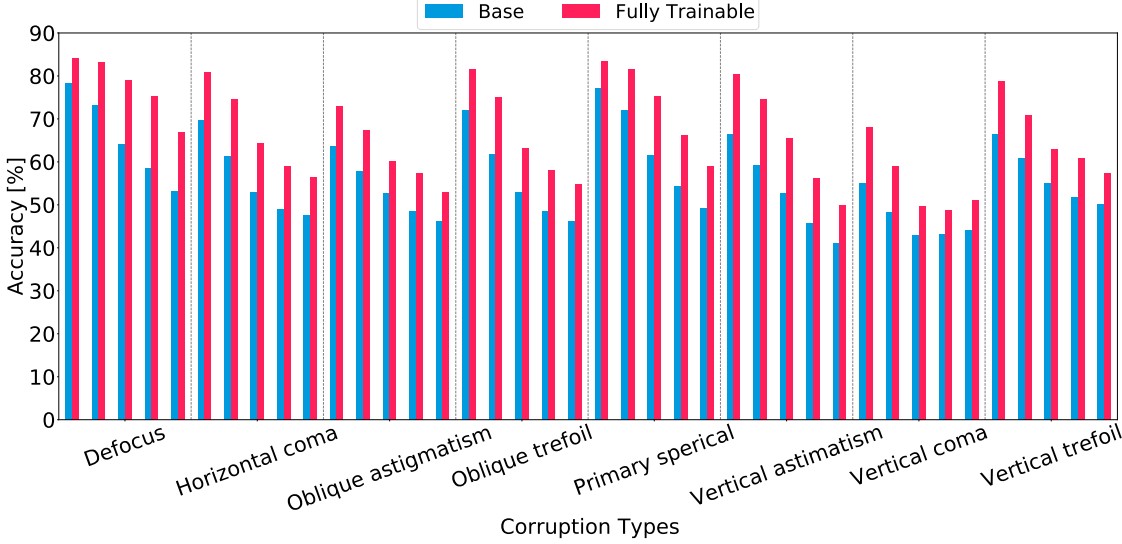

Figure 29: Swin Transformer v2 tiny - ImageNette OpticsBench. A comparison of the Swin Transformer v2 (Base) and a Swin Transformer v2 with our proposed layer (Fully Trainable), on ImageNette OpticsBench. For each corruption type, five levels of severity (Severity 1 to 5) are shown from left to right in each corruption column.

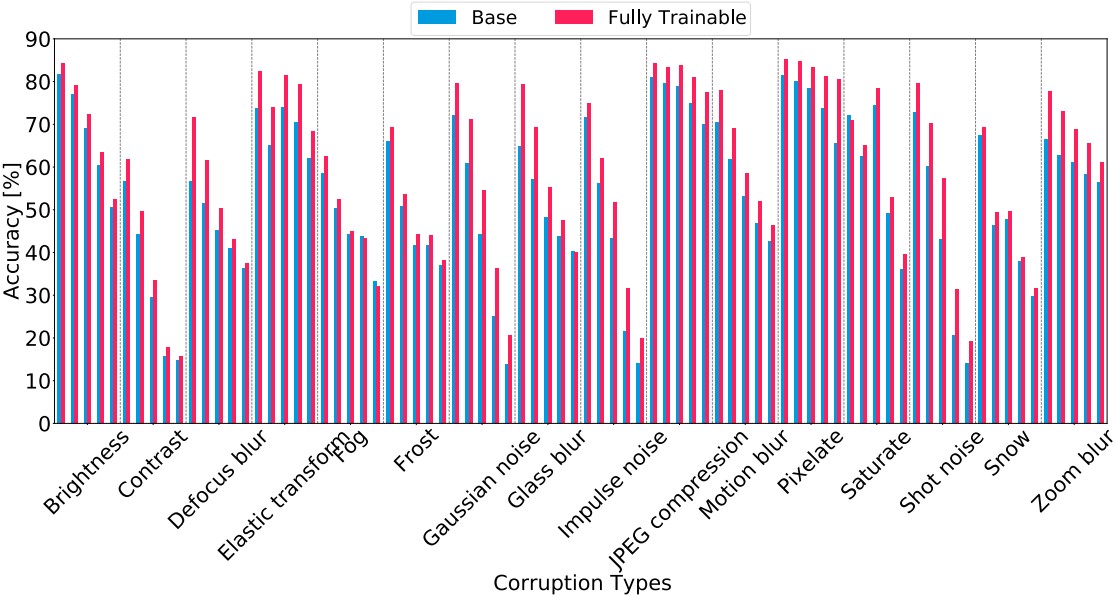

Figure 30: Swin Transformer v2 tiny - ImageNette Common Corruptions. A comparison of the Swin Transformer v2 (Base) and a Swin Transformer v2 with our proposed layer (Fully Trainable), on ImageNette Common Corruptions. For each corruption type, five levels of severity (Severity 1 to 5) are shown from left to right in each corruption column.

## D.2    Kernel Sizes

To explore a suitable kernel size for our proposed layer, we conducted six experiments with slightly increasing kernel sizes. The results over the clean data, the Common Corruption corrupted data, and the OpticsBench corrupted data, is displayed in Table 10. On the clean data, the baseline, without any prior layer to the mode, outperforms all kernel sizes. Furthermore, the models with a prior trainable layer and a small kernel size ($7 \times 7$ and $15 \times 15$) achieve a higher accuracy on the clean data, than models with a large kernel size, such as $31 \times 31$ and $35 \times 35$. However, on the corrupted datasets (OpticsBench and Common Corruptions), the models with a larger kernel size outperform models with smaller kernel sizes and the baseline significantly. The best performance on both corrupted datasets and a just slightly worse performance on the clean dataset is achieved by the model with the kernel size of $25 \times 25$. Therefore, we used this kernel size for all other experiments, if not stated differently.

| Kernel size | CD | OB | CC |
|:---:|:---:|:---:|:---:|
| Base | **\*0.800** | \*0.592 | \*0.487 |
| 7 | 0.782 | 0.616 | 0.526 |
| 15 | 0.790 | 0.575 | 0.532 |
| 21 | 0.765 | 0.610 | 0.530 |
| 25 | \*0.775 | **\*0.685** | **\*0.565** |
| 31 | 0.707 | 0.681 | 0.556 |
| 35 | 0.713 | 0.678 | 0.549 |

Table 10: Results on ImageNette for ResNet50 and different kernel size for the proposed trainable convolutional preprocessing layer. CD = Clean Data, OB = OpticsBench (Müller et al., 2023), CC = Common corruptions (Hendrycks & Dietterich, 2019). * = average from multiple seeds. The results on the two corruption benchmarks are averaged across severity and corruption

## D.3    Static Kernels (Class II)

To investigate the extent to which kernel training improves prediction stability, we freeze different convolutional layer initializations. These static filters are then compared to both the baseline and the fully trainable filters. We investigate two blur filters, one rotational-symmetric Gauss blur filter and a directional (horizontal coma) blur filter obtained from OpticsBench (Müller et al., 2023). These have both lowpass characteristic and remove high frequency content. The color distortion filter aims to split color-specific information by translation. The results on clean data and the different corruption benchmarks on ImageNette are shown in Table 11. It is noticeable that the two blur filters have an in-domain accuracy comparable to the fully trainable or the baseline model, while the color distortion kernel model is significantly lower. Overall, only the Gaussian blur and color distortion models perform better than the baseline on the two corruption benchmarks. The directional blur model performs worst on corruptions, but second best on clean data.

| Kernel type | class | CD | OB | CC |
|---|---|---|---|---|
| None (Base) | - | **\*0.800** | \*0.592 | \*0.487 |
| Preserve content | I | 0.754 | 0.476 | 0.513 |
| Preserve content large | I | 0.645 | 0.478 | 0.440 |
| Fully trainable | II | \*0.775 | \*0.685 | \*0.565 |
| L1 prior | III | 0.712 | **0.699** | **0.567** |
| Random initialization | II | 0.711 | 0.673 | 0.550 |
| Con2D KS=25 | II | 0.655 | 0.601 | 0.501 |
| Directional blur filter | II | 0.768 | 0.437 | 0.345 |
| Gauss blur filter | II | 0.764 | 0.668 | 0.541 |
| Color distortion | II | 0.677 | 0.660 | 0.533 |

Table 11: Results on ImageNette for ResNet50 and different input layer large kernel types. CD = Clean Data, OB = OpticsBench (Müller et al., 2023), CC = Common corruptions (Hendrycks & Dietterich, 2019). * = average from multiple seeds. The results on the two corruption benchmarks are averaged across severity and corruption.

### D.4 Detailed plots of Thresholded Kernel Performance

Fig. 7 shows, the nearly constant performance on the OpticsBench dataset, while reducing the frequencies, with small coefficients, in the trainable kernel. To show this effect in a little more detail, Fig. 31 presents the evaluation results of only one training run. To illustrate the model performance of class III kernels, we trained on MobileNet v3 large model with our proposed convolutional preprocessing layer and no restrictions, such as $L1$ prior. Subsequent to the training, we evaluated the model multiple times with different shares of removed frequencies of the proposed kernel layer. In Fig. 31 the evaluation runs with a lower frequency removal share (0% & 90%) performs better for low severities, but decreases in a faster manner, than evaluation runs with a high frequency removal share (95% & 99%).

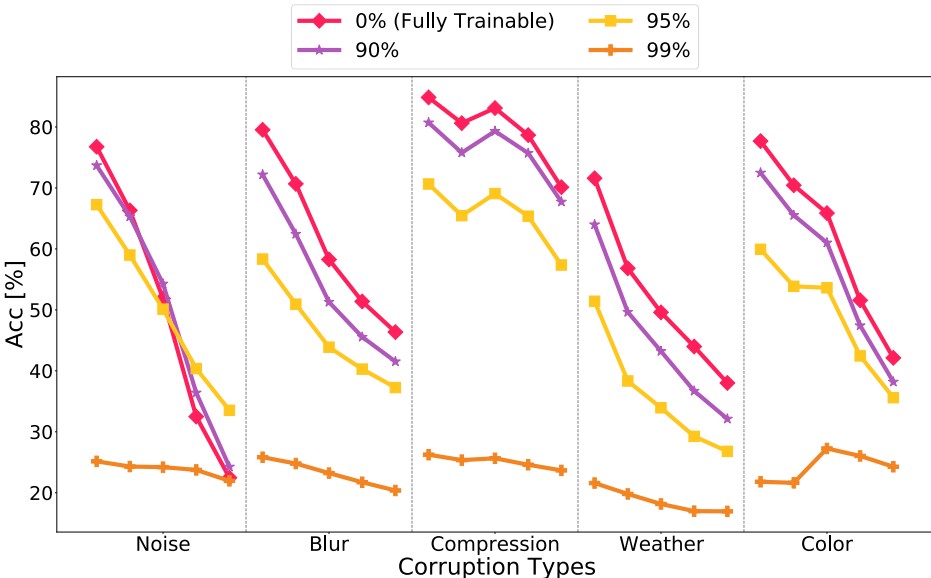

Figure 31: Comparison of frequency thresholded kernels on OpticsBench (Müller et al., 2023) and Common Corruptions (Hendrycks & Dietterich, 2019). All evaluation results from this diagram are from the same MobileNet v3 large trainings run, with different thresholding intervals.

### D.5  $L1$ **Prior different Regularization Strengths**

Sec. D.4 compares already trained class II kernels, which are adjusted to class III kernels, by removing frequencies from the proposed layer kernel. In this section, we further investigate the different class III kernels. Therefore, multiple ResNet50 models with our proposed fully trainable convolutional layer are trained with an $L1$ prior on the frequency coefficients on the trainable kernel of our proposed layer. Each training run uses a different $\lambda$ to shift the loss more on the frequency coefficients on the trainable kernel. Subsequently, these models are evaluated on the OpticsBench (Müller et al., 2023) and Common Corruptions (Hendrycks & Dietterich, 2019), which results in Fig. 32. This experiment shows similar results as in Fig. 31, as the results with a lower lambda are decreasing over higher severities faster. Especially, for noise and blur corruptions, this decreasing over severities effect is more significant. However, it seems like models, which are trained and optimized to also minimize the frequency coefficients on the trainable kernel, tend to have a higher overall accuracy for small $\lambda$s ($\lambda = 0.00001$ to $\lambda = 0.005$), while models with larger $\lambda$s ($\lambda = 0.001$ to $\lambda = 0.01$), seem to perform more constant over all severities of corruptions.

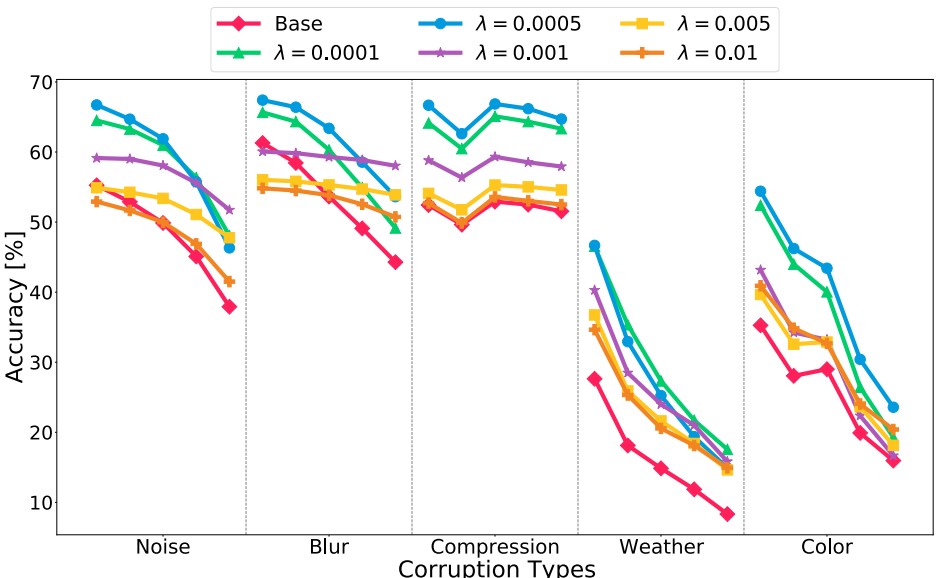

Figure 32: Comparison of models, which are trained with an $L1$ prior. Each model has an ResNet50 architecture, with our additional proposed fully trainable convolutional layer. All these models are trained on ImageNette (Howard, 2023) and evaluated on corruptions from OpticsBench (Müller et al., 2023) and Common Corruptions (Hendrycks & Dietterich, 2019).

### D.6  **ImageNet-100 and ImageNet-1k**

Additionally, to all the experiments on ImageNette (Howard, 2023), we also tested the scalability of the proposed convolutional prior layer. Therefore, Fig. 33 to 42 show comparisons of five different convolutional-based models (ResNet, AlexNet, EfficientNet, MobileNet, and ConvNeXt) on ImageNet-100 (Tian et al., 2020). To evaluate our proposed input layer on two transformer-based Models (Vision Transformer and Swin Transformer v2), Fig. 43 to 46 show the comparisons on ImageNet-100. In each figure, we compare a model with our proposed fully trainable convolutional input layer against the baseline. The figures show the performance on the ImageNet-100 corruptions of OpticsBench (Müller et al., 2023) and Common Corruptions (Hendrycks & Dietterich, 2019). All the comparisons on ImageNet-100 show, that our proposed model outperforms the baseline on the evaluated corruptions. However, the performance gain is slightly lower than on ImageNette.

While training on the whole ImageNet-1k dataset (Russakovsky et al., 2015), the performance differences between the baseline and our proposed convolutional-based model is even lower than on ImageNet-100. Only

for severity 4 and 5 our model outperforms the baseline. This effect is visualized in Fig. 47 and 48. However, the performance on clean data is also just slightly in favor of the baseline, which is stated in Table 4. This does not hold for the transformer-based model (Swin Transformer v2), which outperforms the baseline on all severities, which can be examined in Fig. 49.

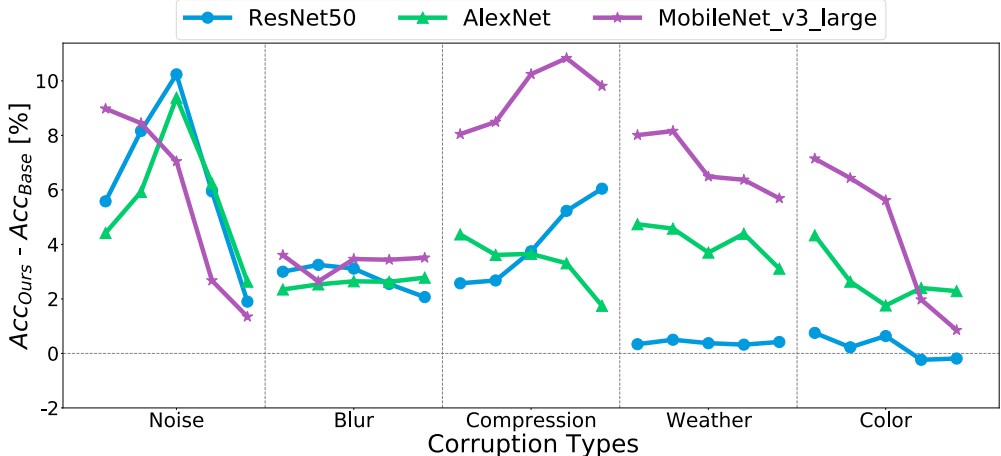

Figure 33: Improvement on ImageNet-100 OpticsBench and Common Corruptions. For each corruption, five levels of severity are shown from left to right. For readability, we summarize different corruption types and take the average.

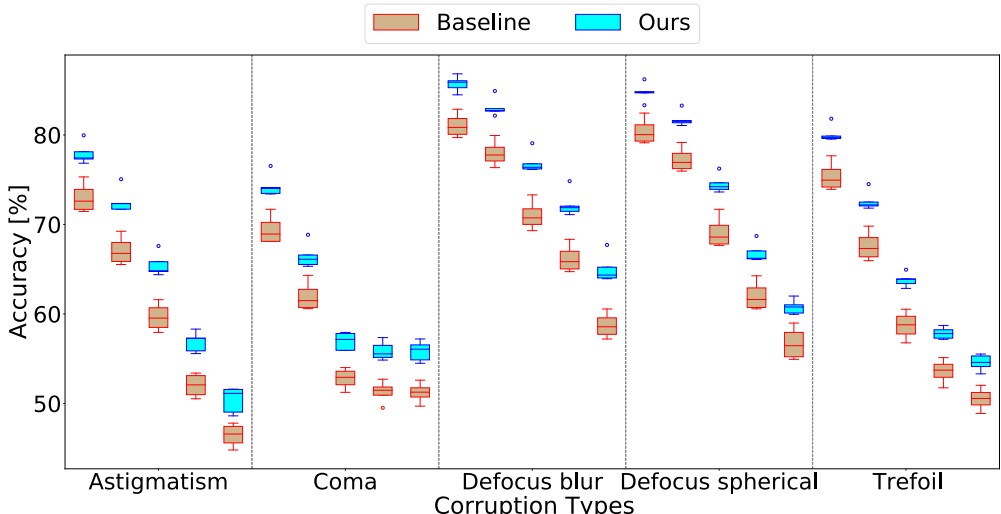

Figure 34: Convolutional preprocessing (ours) can increase the robustness of classification networks against unknown corruptions *without data augmentation*. ResNet50 improved with a trainable preprocessing filter evaluated on ImageNet-100 (Tian et al., 2020) blur and corruptions from OpticsBench (Müller et al., 2023). For each corruption type, five levels of severity are shown from left to right. The variation, visualized via the box plots, results from five different seeds per model.

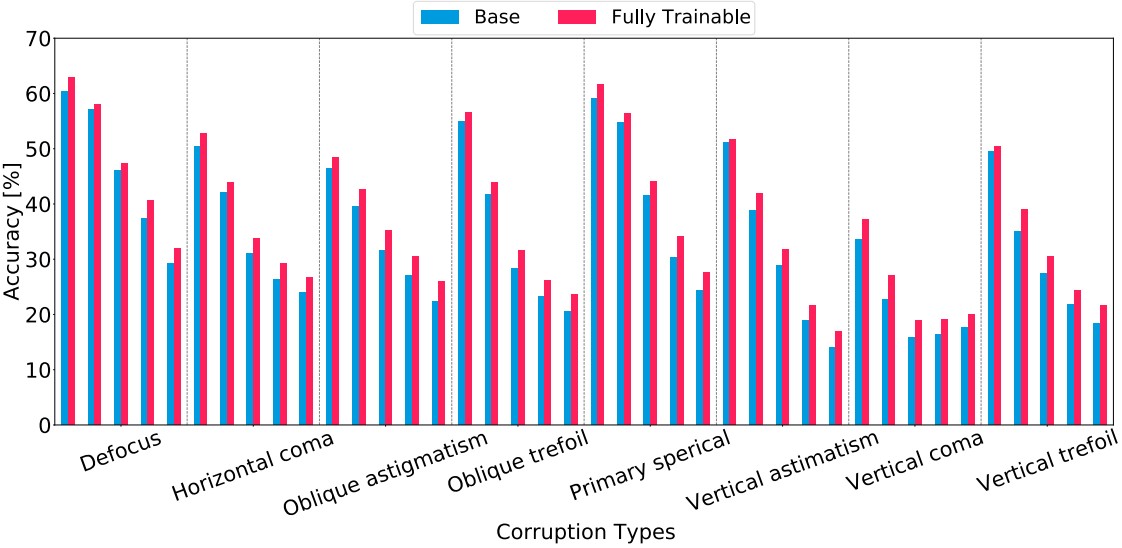

Figure 35: AlexNet - ImageNet-100 OpticsBench. A comparison of the AlexNet (Base) and a AlexNet with our proposed layer (Fully Trainable), on ImageNet-100 OpticsBench. For each corruption type, five levels of severity (Severity 1 to 5) are shown from left to right in each corruption column.

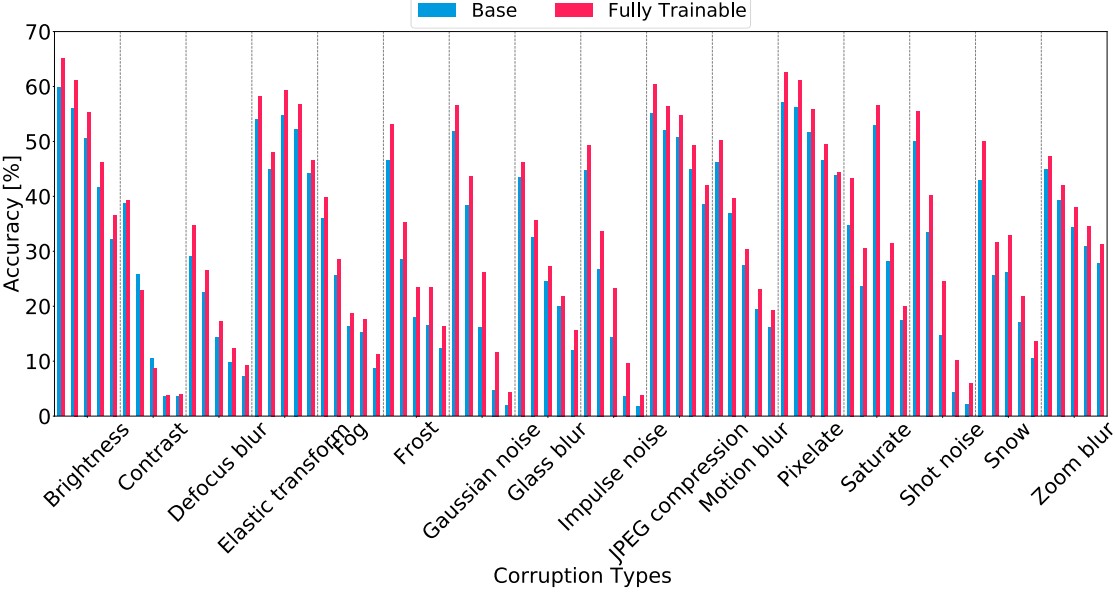

Figure 36: AlexNet - ImageNet-100 Common Corruptions. A comparison of the AlexNet (Base) and a AlexNet with our proposed layer (Fully Trainable), on ImageNet-100 Common Corruptions. For each corruption type, five levels of severity (Severity 1 to 5) are shown from left to right in each corruption column.

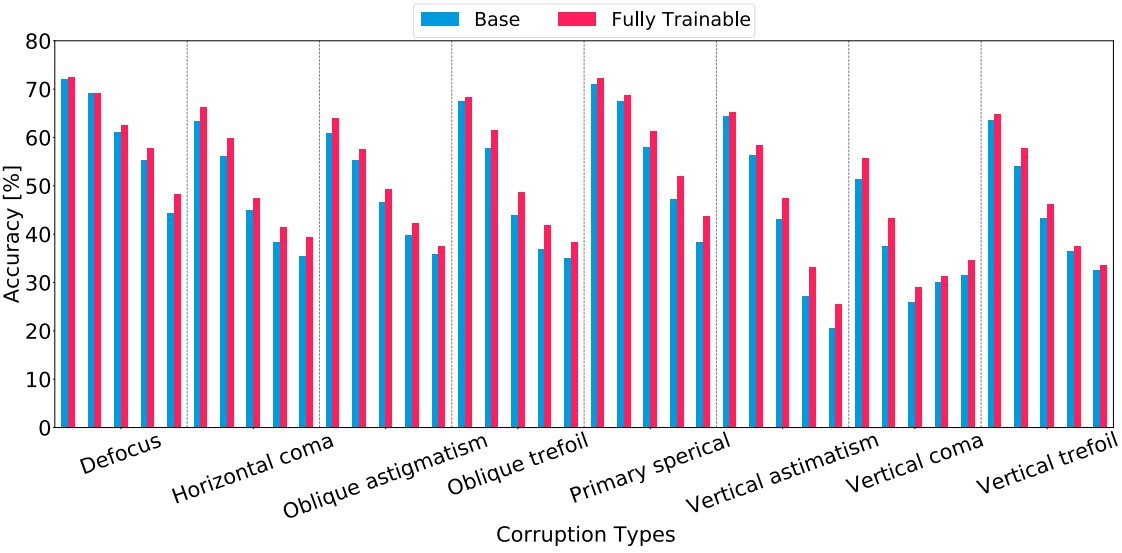

Figure 37: EfficientNet b0 - ImageNet-100 OpticsBench. A comparison of the EfficientNet b0 (Base) and a EfficientNet b0 with our proposed layer (Fully Trainable), on ImageNet-100 OpticsBench. For each corruption type, five levels of severity (Severity 1 to 5) are shown from left to right in each corruption column.

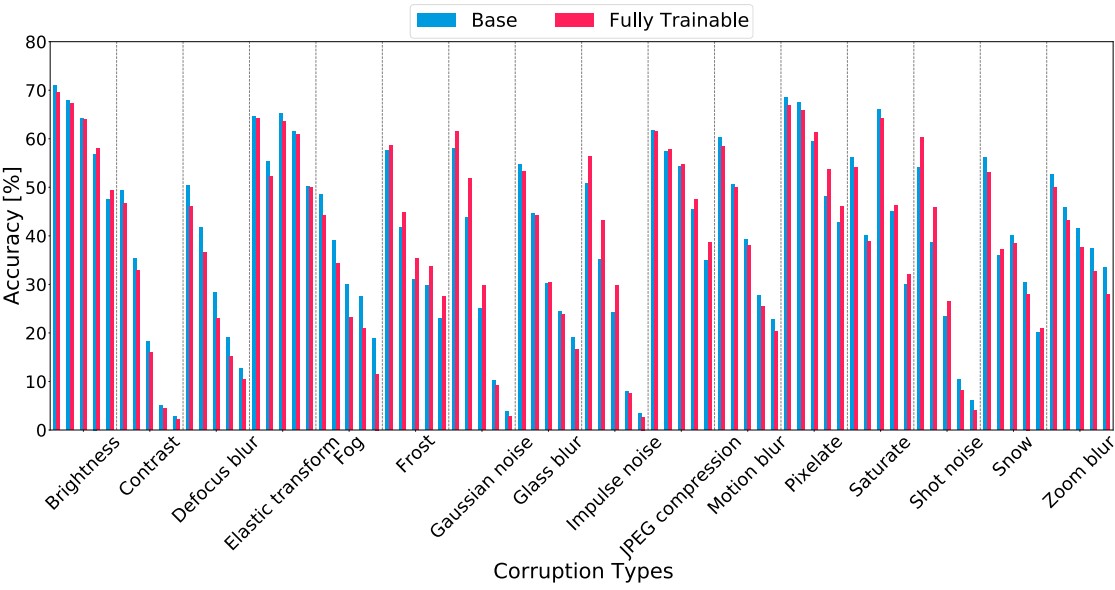

Figure 38: EfficientNet b0 - ImageNet-100 Common Corruptions. A comparison of the EfficientNet b0 (Base) and a EfficientNet b0 with our proposed layer (Fully Trainable), on ImageNet-100 Common Corruptions. For each corruption type, five levels of severity (Severity 1 to 5) are shown from left to right in each corruption column.

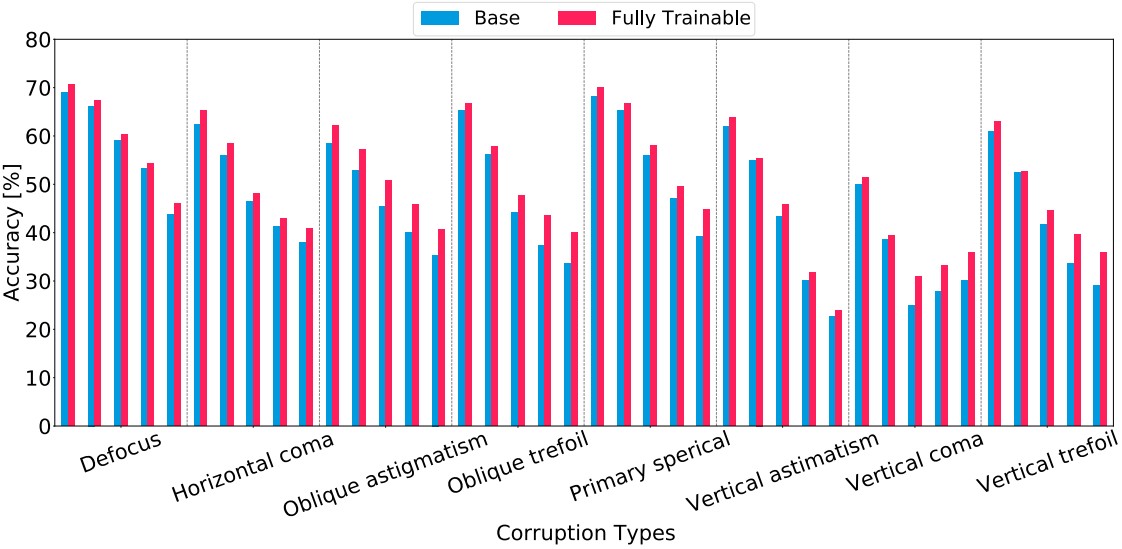

Figure 39: MobileNet v3 large - ImageNet-100 OpticsBench. A comparison of the MobileNet v3 large (Base) and a MobileNet v3 large with our proposed layer (Fully Trainable), on ImageNet-100 OpticsBench. For each corruption type, five levels of severity (Severity 1 to 5) are shown from left to right in each corruption column.

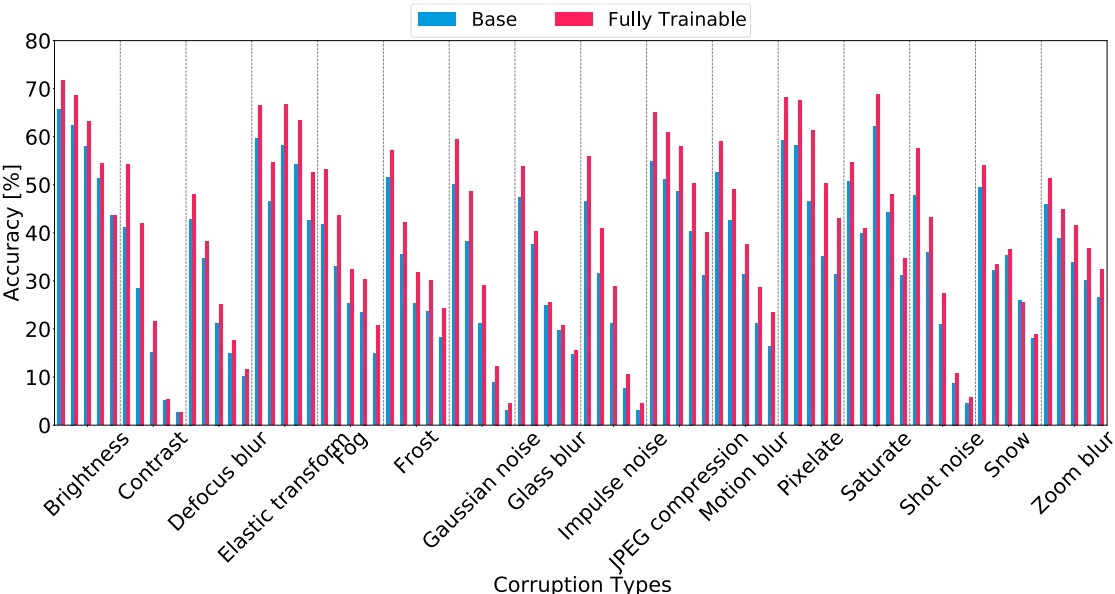

Figure 40: MobileNet v3 large - ImageNet-100 Common Corruptions. A comparison of the MobileNet v3 large (Base) and a MobileNet v3 large with our proposed layer (Fully Trainable), on ImageNet-100 Common Corruptions. For each corruption type, five levels of severity (Severity 1 to 5) are shown from left to right in each corruption column.

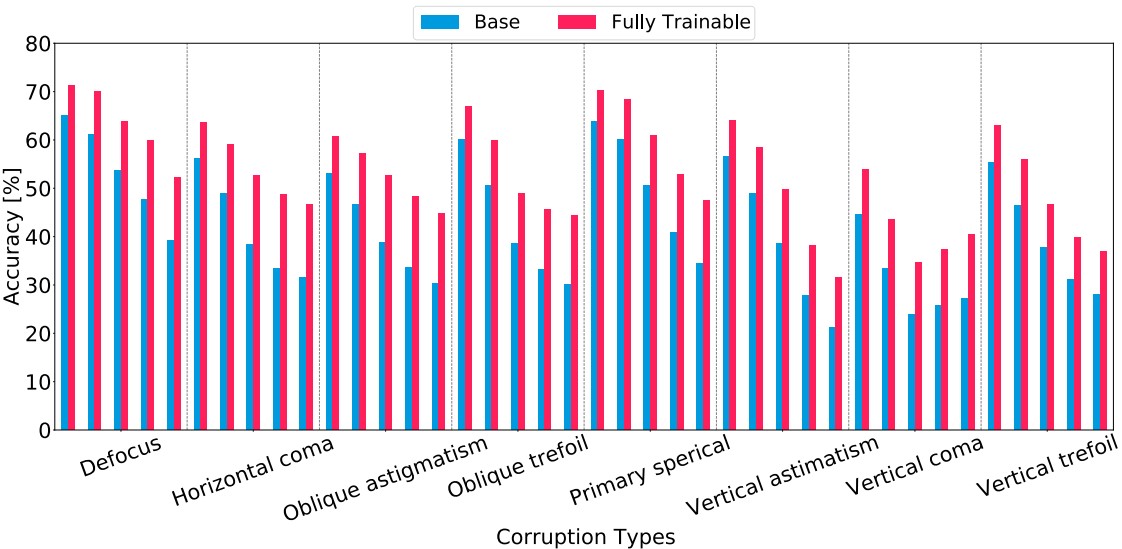

Figure 41: ConvNeXt small - ImageNet-100 OpticsBench. A comparison of the ConvNeXt (Base) and a ConvNeXt with our proposed layer (Fully Trainable), on ImageNet-100 OpticsBench.For each corruption type, five levels of severity (Severity 1 to 5) are shown from left to right in each corruption column.

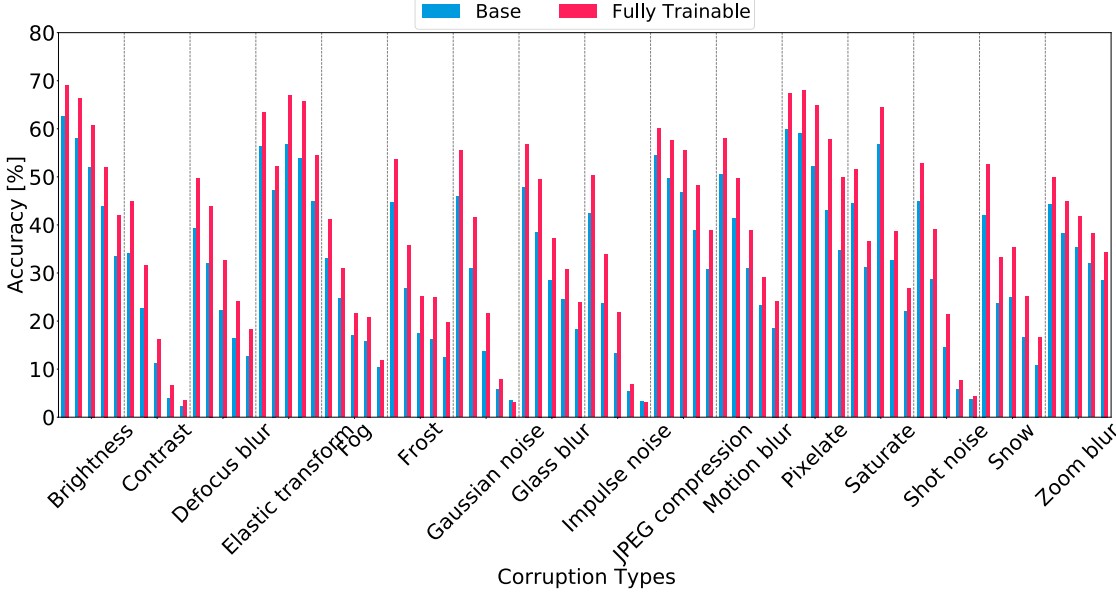

Figure 42: ConvNeXt small - ImageNet-100 Common Corruptions. A comparison of the ConvNeXt (Base) and a ConvNeXt with our proposed layer (Fully Trainable), on ImageNet-100 Common Corruptions.For each corruption type, five levels of severity (Severity 1 to 5) are shown from left to right in each corruption column.

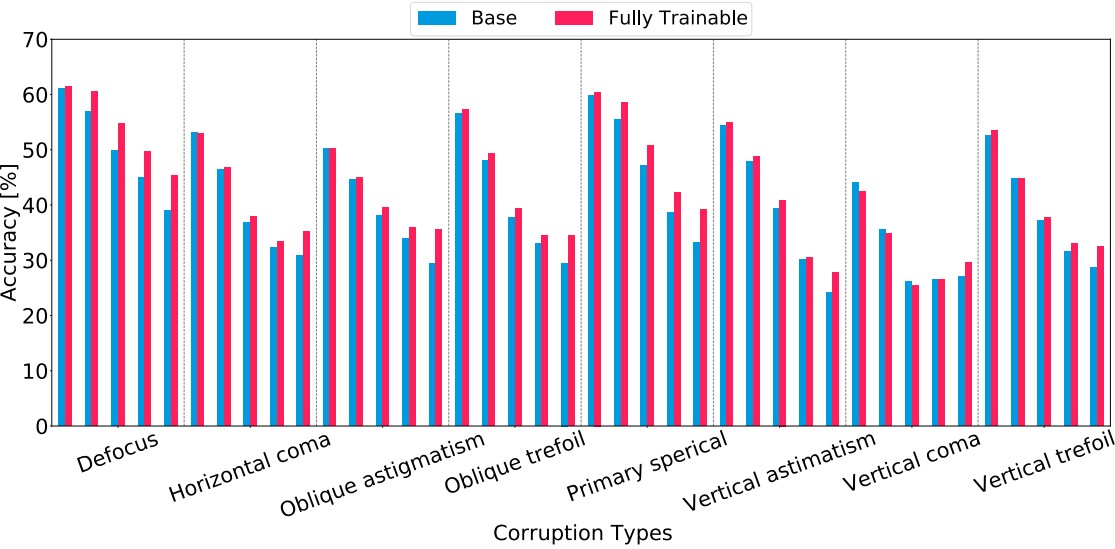

Figure 43: Vision Transformer base - ImageNet-100 OpticsBench. A comparison of the Vision Transformer (Base) and a Vision Transformer with our proposed layer (Fully Trainable), on ImageNet-100 Optics-Bench.For each corruption type, five levels of severity (Severity 1 to 5) are shown from left to right in each corruption column.

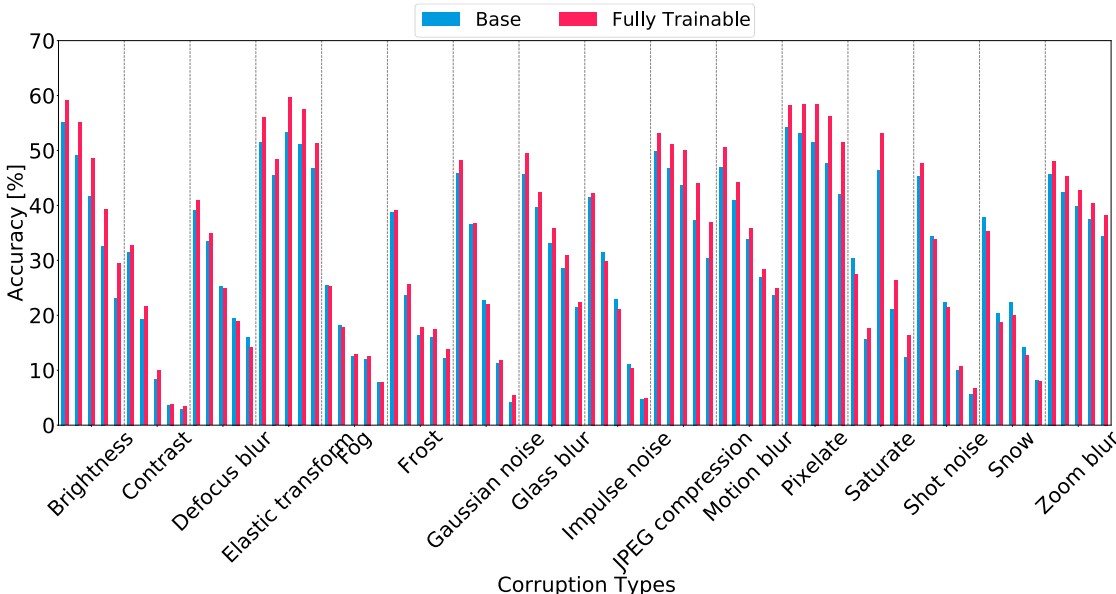

Figure 44: Vision Transformer base - ImageNet-100 Common Corruptions. A comparison of the Vision Transformer (Base) and a Vision Transformer with our proposed layer (Fully Trainable), on ImageNet-100 Common Corruptions. For each corruption type, five levels of severity (Severity 1 to 5) are shown from left to right in each corruption column.

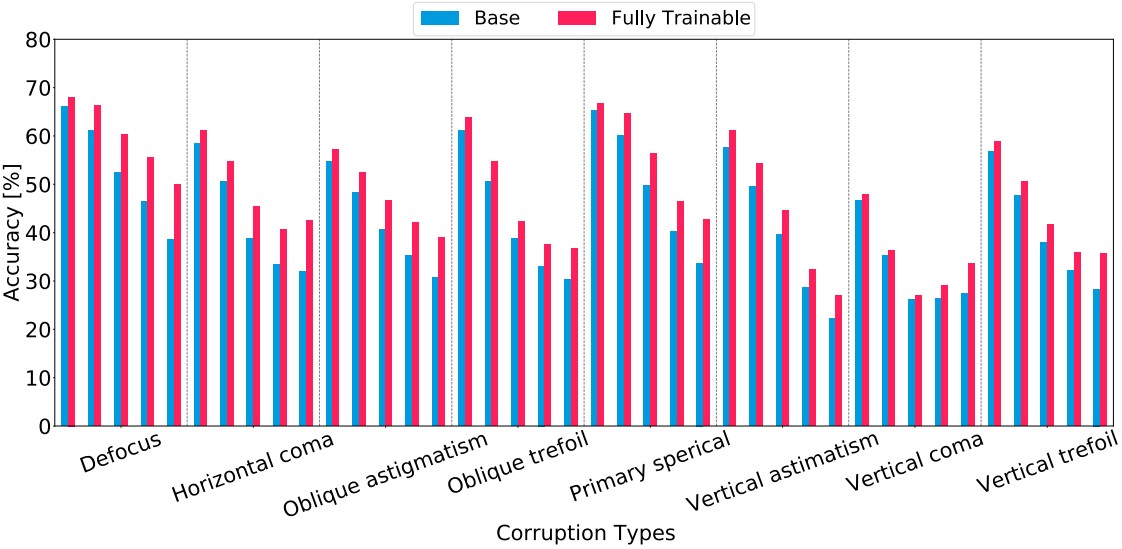

Figure 45: Swin Transformer v2 tiny - ImageNet-100 OpticsBench. A comparison of the Swin Transformer v2 (Base) and a Swin Transformer v2 with our proposed layer (Fully Trainable), on ImageNet-100 OpticsBench. For each corruption type, five levels of severity (Severity 1 to 5) are shown from left to right in each corruption column.

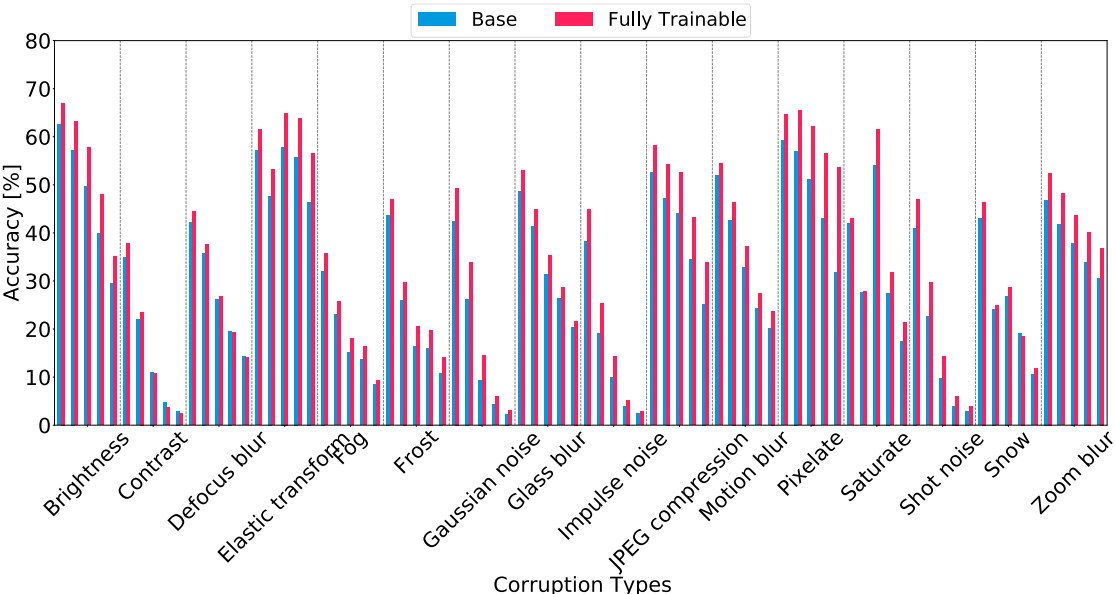

Figure 46: Swin Transformer v2 tiny - ImageNet-100 Common Corruptions. A comparison of the Swin Transformer v2 (Base) and a Swin Transformer v2 with our proposed layer (Fully Trainable), on ImageNet-100 Common Corruptions. For each corruption type, five levels of severity (Severity 1 to 5) are shown from left to right in each corruption column.

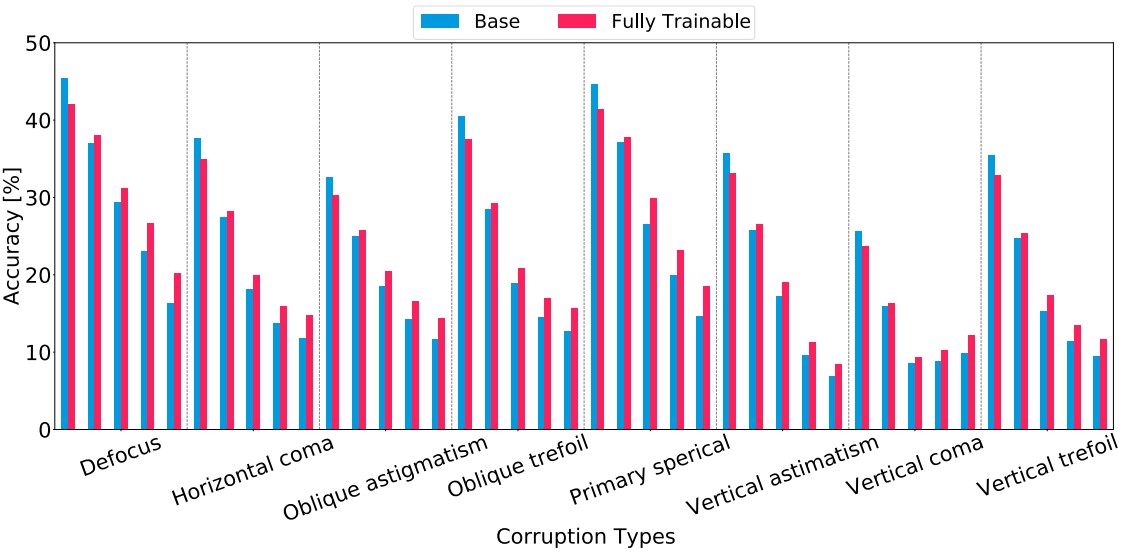

Figure 47: ResNet50 - ImageNet-1k OpticsBench. A comparison of the ResNet50 (Base) and a ResNet50 with our proposed layer (Fully Trainable), on ImageNet-1k OpticsBench. For each corruption type, five levels of severity (Severity 1 to 5) are shown from left to right in each corruption column.

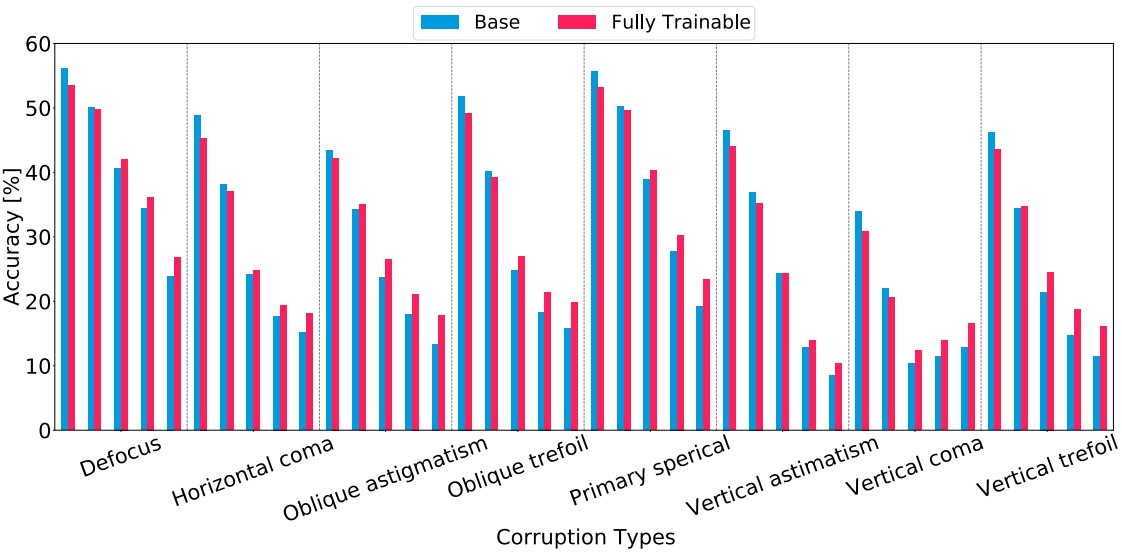

Figure 48: MobileNet v3 large - ImageNet-1k OpticsBench. A comparison of the MobileNet v3 large (Base) and a MobileNet v3 large with our proposed layer (Fully Trainable), on ImageNet-1k OpticsBench. For each corruption type, five levels of severity (Severity 1 to 5) are shown from left to right in each corruption column.

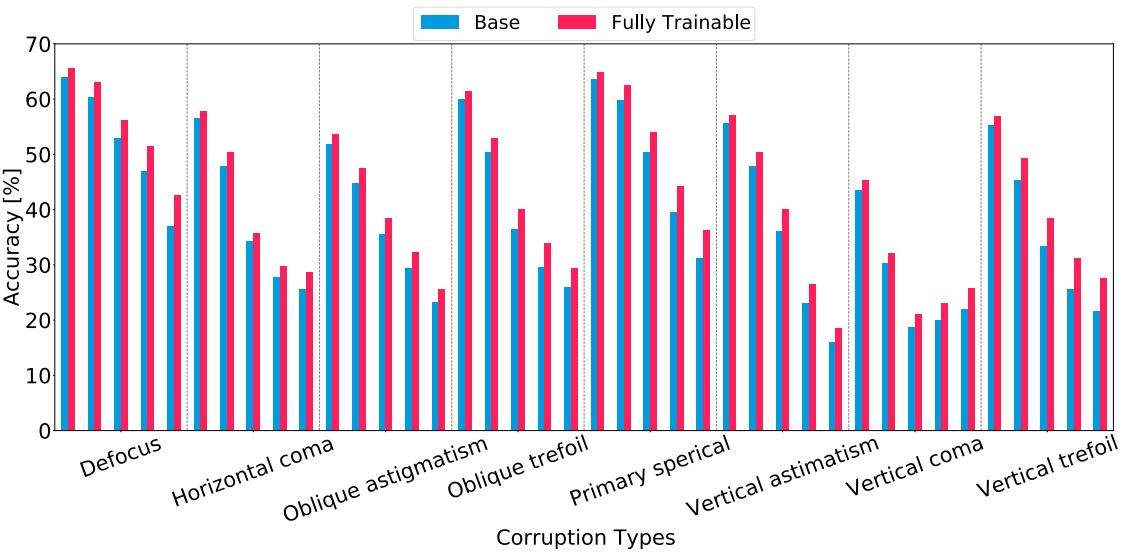

Figure 49: Swin Transformer v2 tiny - ImageNet-1k OpticsBench. A comparison of the Swin Transformer v2 (Base) and a Swin Transformer v2 with our proposed layer (Fully Trainable), on ImageNet-1k OpticsBench. For each corruption type, five levels of severity (Severity 1 to 5) are shown from left to right in each corruption column.

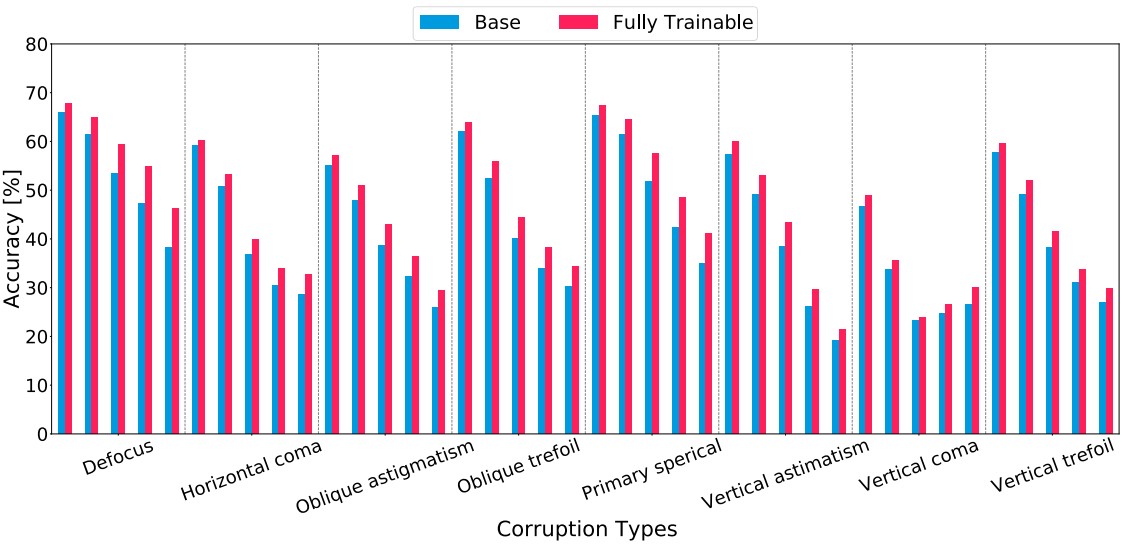

Figure 50: Swin Transformer v2 base - ImageNet-1k OpticsBench. A comparison of the Swin Transformer v2 (Base) and a Swin Transformer v2 with our proposed layer (Fully Trainable), on ImageNet-1k OpticsBench. For each corruption type, five levels of severity (Severity 1 to 5) are shown from left to right in each corruption column.

## D.7 Adversarial Training and Robustness

We further investigate the performance of the proposed input layer on white-box adversarial attacks. Therefore, we trained a ResNet50 with and without our proposed trainable convolutional input layer. The models are trained with the fast gradient sign method (FGSM) (Goodfellow et al., 2015) and an $\epsilon$ of $\frac{2}{255}$ and $\frac{6}{255}$ and evaluated on clean data and under the FGSM adversarial attack, with two different $\epsilon$ of $\frac{2}{255}$ and $\frac{6}{255}$.

| Model | Version | CD | FGSM $\epsilon = \frac{2}{255}$ | FGSM $\epsilon = \frac{6}{255}$ |
|---|---|---|---|---|
| ResNet50 | Adv. Baseline | 0.665 | 0.567 | 0.385 |
| ResNet50 | Adv. Trainable | **0.776** | **0.667** | **0.389** |
| ResNet50 | Adv. Baseline | 0.534 | 0.483 | 0.760 |
| ResNet50 | Adv. Trainable | **0.581** | **0.527** | **0.776** |

Table 12: Adversarial Results on ImageNette (Howard, 2023) data. CD= Clean Data, FGSM = FGSM attack (Goodfellow et al., 2015) with two different $\epsilon$ of $\frac{2}{255}$ and $\frac{6}{255}$. The two models in the top rows are trained with an $\epsilon$ of $\frac{2}{255}$, while the two model on the bottom are trained with an $\epsilon$ of $\frac{6}{255}$.

In Table 12 the results of these experiments, indicate the robustness of the models with our preprocessing layer: they are outperforming the baseline on all evaluated data. While both models have similar accuracy under FGSM attacks with $\epsilon$ of $\frac{2}{255}$, our model achieves better results on clean data (+11.1% & +4.7%) and with a lower $\epsilon$ value of $\frac{2}{255}$ (+10.0% & +4.4%). These increases are possible while only adding one layer with less than 2k of parameters.

To further extend the experiment on robustness against adversarial attacks, we trained models without adversarial training and compared, whether our proposed input layer is able to increase the robustness even without specific adversarial training. Therefore, we used AutoAttack (Croce & Hein, 2020) with its 4 adversarial attacks and reported SQUARE (Andriushchenko et al., 2019) additionally as a black-box attack. Table 13 displays the results of this experiment. On both architectures is our model able to outperform the baseline.

| Model | Version | AA | SQUARE |
|---|---|---|---|
| ResNet50 | Base | 0.199 | 0.156 |
| ResNet50 | Fully Trainable | **0.229** | **0.193** |
| AlexNet | Base | 0.446 | 0.518 |
| AlexNet | Fully Trainable | **0.468** | **0.657** |

Table 13: Adversarial Results on AutoAttack (Croce & Hein, 2020) with an epsilon of 4/255 and SQUARE (Andriushchenko et al., 2019). AA= AutoAttack, SQUARE on ImagenNette (Howard, 2023).

## D.8 Prepended Input Layer on Pretrained Models

To have a deeper understanding of the scope of our proposed input layer, we tested whether the entire model needs to be trained or only our prepended input layer can be trained. Therefore, we used pretrained models and prepended our proposed input layer, subsequently, we trained the models while freezing all layers except our proposed input layer. This resulted in a model, which was on par with the baseline on clean data, but performed slightly worse on the OpticsBench dataset. However, as table 14 shows, that the model with the input layer slightly outperforms the baseline on the Common Corruptions dataset.

To summarize these experiments, our proposed input layer should be trained with the model, as the feature extraction layer is tuned on the specific input to these layers. As the input layer transforms the input to the subsequent layers, they are not able to extract the corresponding features in the same way anymore. Thus, the performance boost of the prepended input layer on corrupted data is no longer present.

| Dataset | Model | Version | CD | OB | CC |
|---------|-------|---------|-----|-----|-----|
| ImageNette | ResNet50 | Pretrained Base | **0.987** | **0.870** | 0.807 |
| | ResNet50 | Prepended Input Layer | 0.986 | 0.851 | **0.813** |
| Imagenet-100 | ResNet50 | Pretrained Base | **0.868** | **0.615** | 0.467 |
| | ResNet50 | Prepended Input Layer | 0.866 | 0.607 | **0.470** |

Table 14: Results from prepending our input layer to pretrained ResNet50 models on Results on ImageNette (Howard, 2023) and ImageNet-100 (Tian et al., 2020) datasets.

# E    Analysis of Optimized Kernels

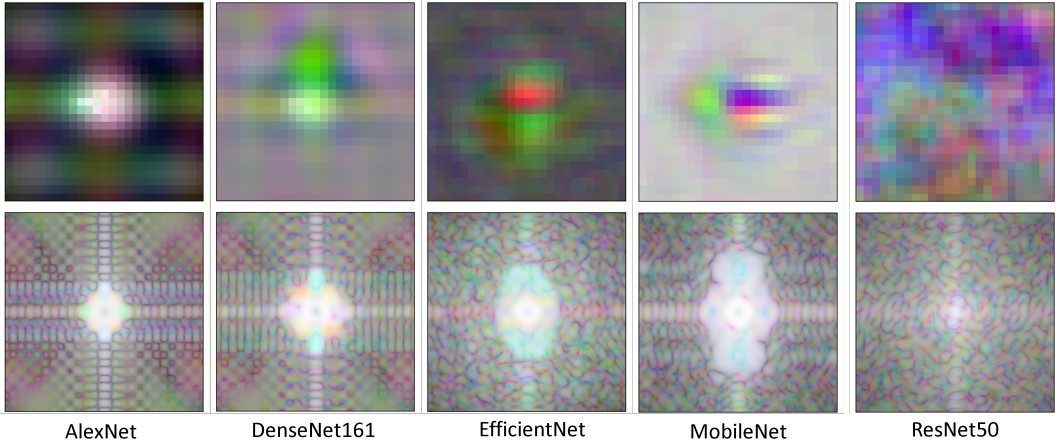

AlexNet          DenseNet161          EfficientNet          MobileNet          ResNet50

Figure 51: Optimized kernels in the spatial domain (top) and their image-sized frequency magnitudes in $\log_{10}$ scale (bottom) trained on ImageNette.

In this section, we perform an in-depth analysis of the pre-pended kernels. Fig. 51 visualizes optimized class II kernels and their corresponding frequency magnitudes. It can be observed that the optimized kernels for different neural networks are different. In the following, we quantitatively assess the similarities of these kernels, compare the evolution of their spectra, and discuss their condition numbers in finer detail.

## E.1    Spectrum Correlation

To ensure that our results reflect performance gains due to the proposed method and not because of fortuitous initialization, we experiment with multiple seeds. The kernel is, without change, initialized with a specific coma filter, and the weights of the remaining network are generated pseudo-randomly. We find that the performance of our networks is unaffected. A note-worthy discovery, however, is that despite different initializations, the optimized kernels' spectra are found to be very similar, given the neural network architecture and the input dataset remain the same. Fig. 52 numerically assesses the structural similarity of the spectra of multiple kernels post-optimization. One reason for the observed similarity could be that the proposed kernel learns to map the input to an information-dense subspace most suited to aid the following network in its classification task. Given that the input dataset and the architecture remain invariant, the information-dense subspace should not change as well. Another observation is that given the same input dataset, changing the network changes the mapping learned by the kernel. This suggests that different neural networks predominantly make use of different information content of the unadulterated dataset.

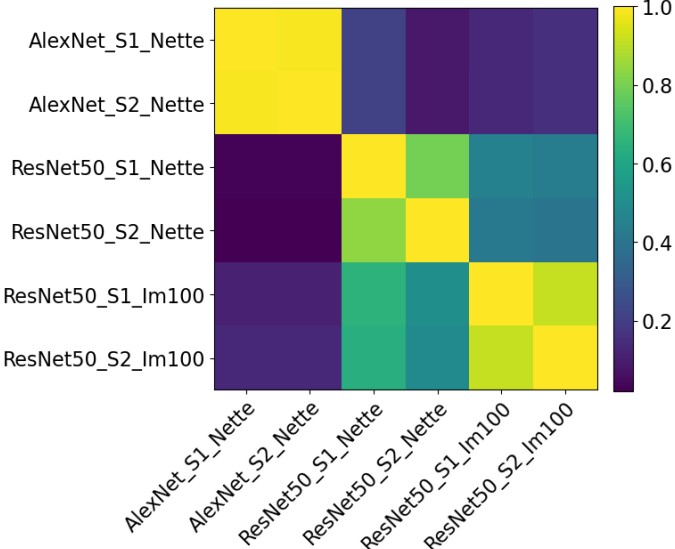

Figure 52: Structural similarity of the magnitudes of the spectra of different kernels. S1 and S2 refer to different seed values, and Im100 and Nette refer to ImageNet-100 and ImageNette, respectively.

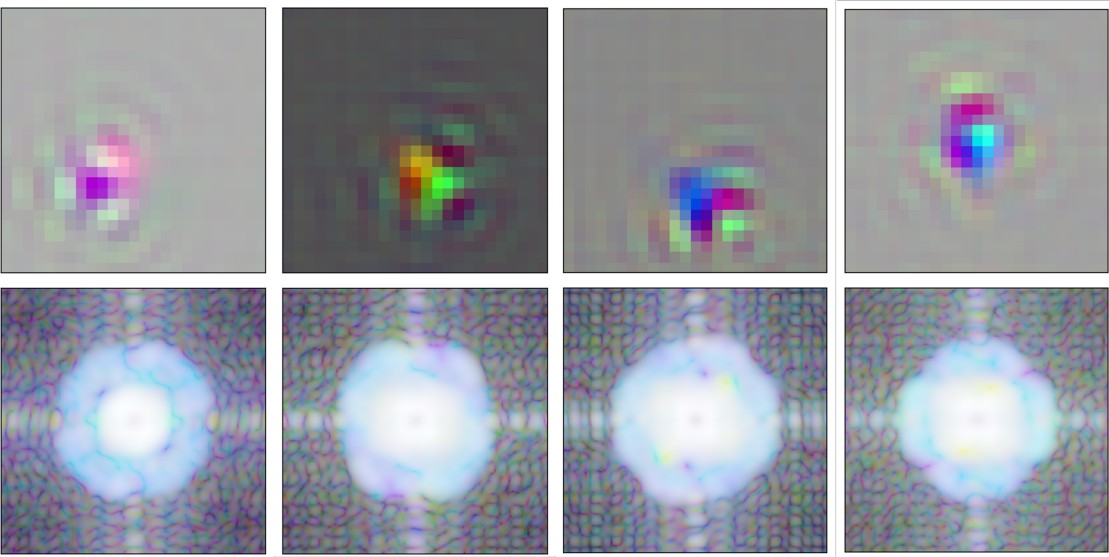

Figure 53: Optimized kernels in the spatial domain (top) and their image-sized frequency magnitudes in $\log_{10}$ scale (bottom) pre-pended to ResNet50 initialized with different seeds and trained on ImageNet-100.

## E.2 Spectrum Evolution

We initialize our kernels with a specific coma filter, which can be observed in Fig. 54 (a) and (d). Moreover, these kernels do not mix channel information. We can observe the evolution of the kernels in the spatial and frequency domains in Fig. 54, when the networks are trained on ImageNet-100. From the figure, it is also apparent that the kernels learn different projections for each color channel.

Fig. 55 compares the evolution of class II and class III kernels. The frequency band axis represents the dimension of a square window, with its center being the center of the frequency-shifted spectrum. Each bar indicates the average magnitude of the frequencies in this window, which we refer to as a frequency band. Then, the first bar in each epoch is the magnitude of the DC component of the spectrum and the last is

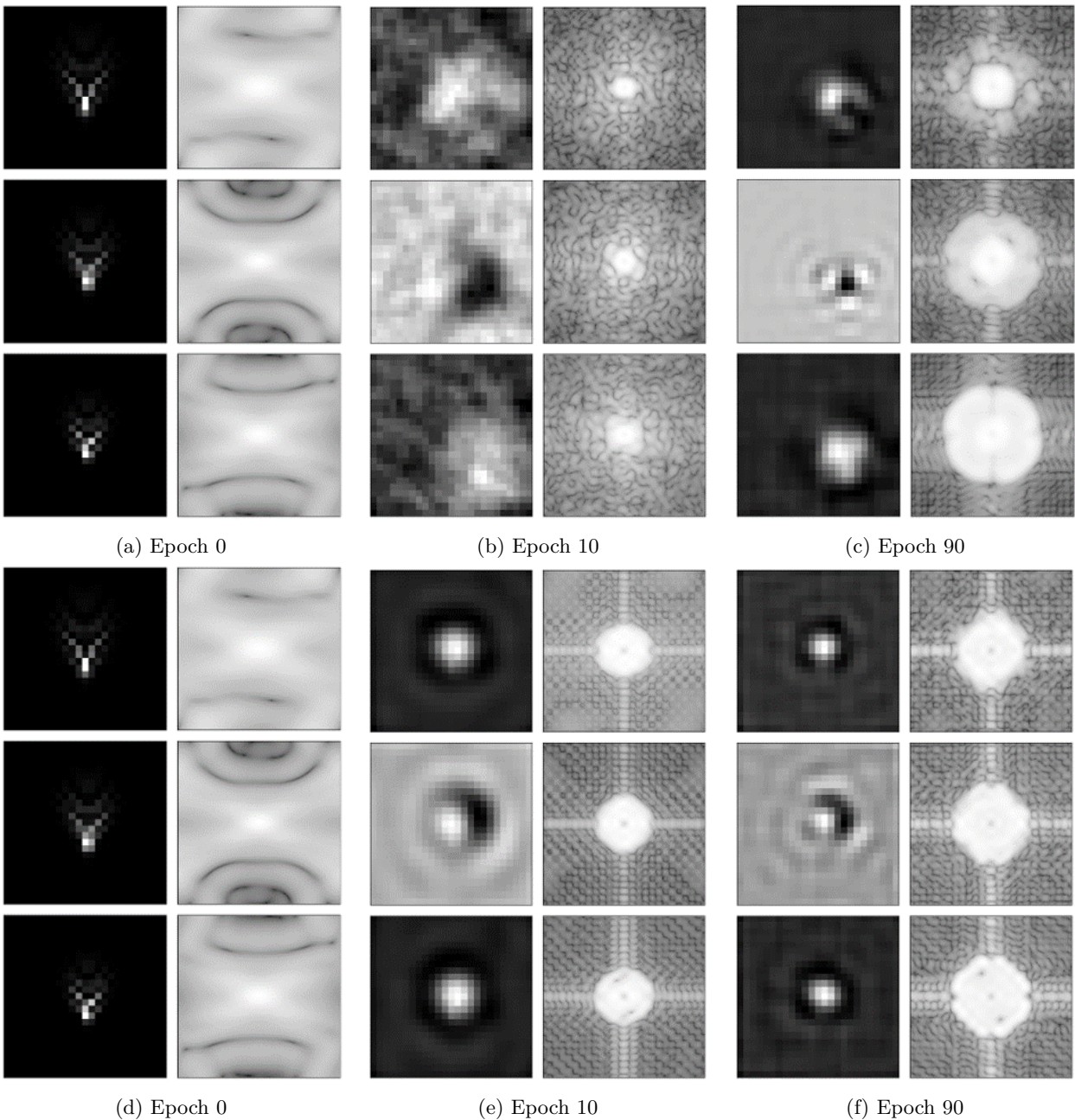

Figure 54: Evolution of kernels in the spatial domain (left) and their frequency magnitudes in $\log_{10}$ scale (right). Kernels in (a-c) and (d-f) were pre-pended to ResNet50 and AlexNet, respectively. Each row corresponds to a color channel.

the mean of the magnitudes of the entire spectrum. From Fig. 55, we can observe that the spread of the bars depends on the presence of the $L1$ prior. The $L1$ prior reduces the higher frequency components of the spectrum. Barring the change in prior, the evolution of the spectrum is similar.

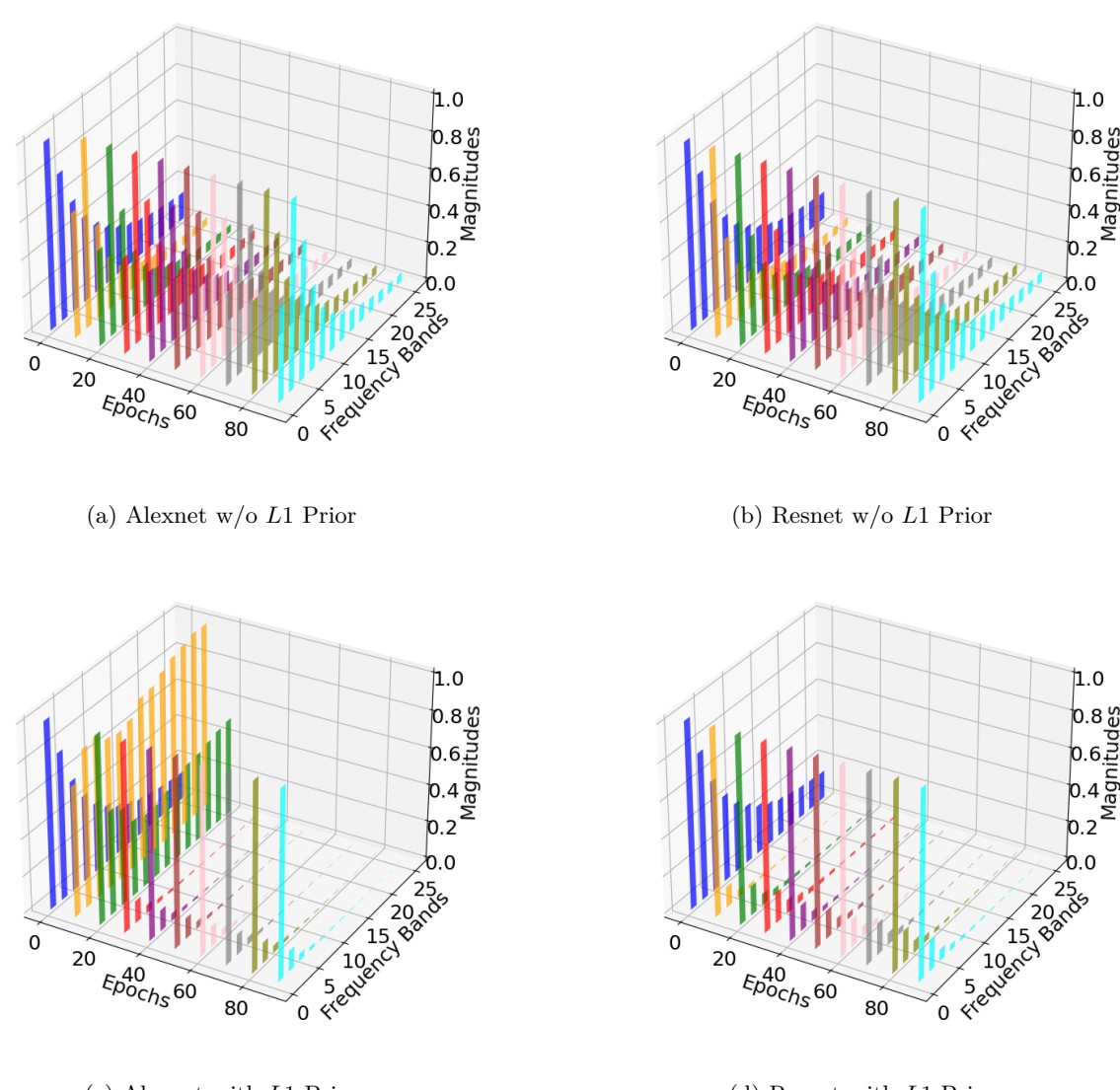

(a) Alexnet w/o $L1$ Prior

(b) Resnet w/o $L1$ Prior

(c) Alexnet with $L1$ Prior

(d) Resnet with $L1$ Prior

Figure 55: Evolution of spectra of (a-b) Fully Trainable and (c-d) $L1$ Prior kernels. The bar height indicates the average of the absolute value of the Fourier coefficients in different frequency bands (DC component in the front). Each epoch is normalized separately.

### E.3 Condition Numbers

We use the condition number (CN) as one of the two defining characteristics of our classes of kernels. One significant feature of this description is that the convolution kernel, $\cdot * g$, and the corresponding deconvolution kernel, $\cdot * g^{-1}$, share the same CN. This is particularly important for class I kernels because it suggests that the neural network following $\cdot * g$ can perfectly restore the input without being stymied by noise amplification, which distinguishes class I kernels from the rest. In this section, we will derive the equation for CN provided in Sec. 3 and further inspect the CNs of class II kernels.

**Derivation of the Condition Number:** For an arbitrary invertible matrix $A$, CN is defined as:

$$CN(A) = ||A|| \cdot ||A^{-1}|| = ||A^{-1}|| \cdot ||A|| = CN(A^{-1}),$$

where $||.||$ is the norm of a matrix. The sensitivity of $A$ to noise or the noise amplification possible by the transformation is equal to that of its inverse. Moreover, $||A|| = \sigma_{\max}(A)$ and $||A^{-1}|| = 1/\sigma_{\min}(A)$, where $\sigma_{\max}$ and $\sigma_{\min}$ are the maximum and minimum singular values of the matrix, respectively. In our case, the transformation $A$ is a circulant matrix describing the convolution operation. The eigenvectors of $A$ form an orthonormal Fourier basis. Let each basis be a column of $Q$, and let the corresponding eigenvalues $\lambda_i$, the Fourier coefficients, form a diagonal matrix, $\Lambda$. $A$ can be decomposed as

$$A = Q\Lambda Q^T,$$

as it is a normal matrix. Then using the relation $\sigma_i = \sqrt{\lambda_i(A^T A)}$,

$$\sigma_{\max}(A) = \sqrt{\lambda_{\max}(A^T A)}$$
$$= \sqrt{\lambda_{\max}\left((Q\Lambda Q^T)^T Q\Lambda Q^T\right)}$$
$$= \sqrt{\lambda_{\max}(Q\Lambda^2 Q^T)}$$
$$= |\lambda_{\max}(A)|.$$

Similarly, $\sigma_{\min}(A) = |\lambda_{\min}(A)|$. Therefore,

$$CN(A) = \frac{|\lambda_{\max}(A)|}{|\lambda_{\min}(A)|} = \frac{1/|\lambda_{\min}(A)|}{1/|\lambda_{\max}(A)|} = CN\left(A^{-1}\right).$$

**Class II Kernels:** The CN of class I kernels is by their definition equal to unity and the CNs of class III kernels are numerically very high (approaching infinity) due to the presence of zeros in the Fourier domain. However, class II kernels have high finite CNs, around $10^4$. We can observe these in Fig. 56. The range of CNs of the optimized kernels is within one order of magnitude despite being pre-pended to different neural networks and trained on different data sets.

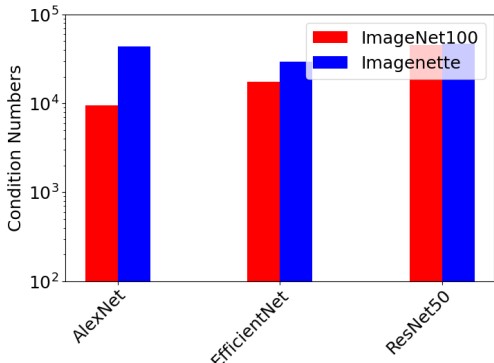

Figure 56: Condition numbers of post-optimization class II kernels pre-pended to different neural networks and trained on ImageNet-100 and ImageNette. Each bar is a mean of up to 5 runs.

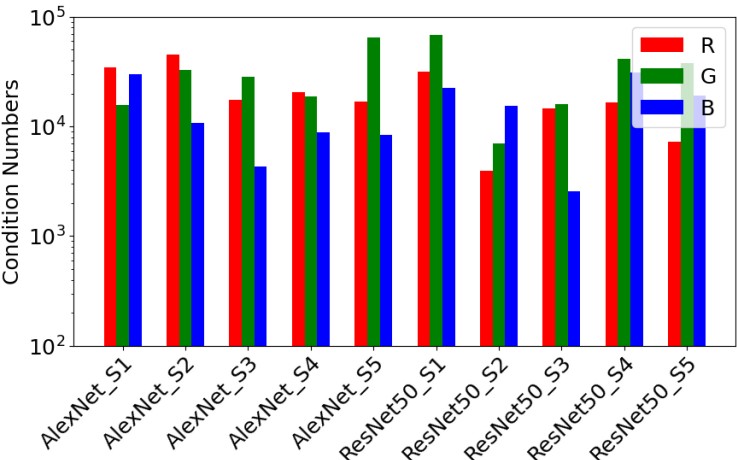

Figure 57: Condition numbers of post-optimization class II kernels pre-pended to networks specified in the xlabel and trained on ImageNette. S1-S5 refers to different seed values, and R, G, and B refer to the color channels.

# F  Extended Discussion

In the main paper, we only briefly visit two interesting discussion points, which we pick up here to extend the discussion.

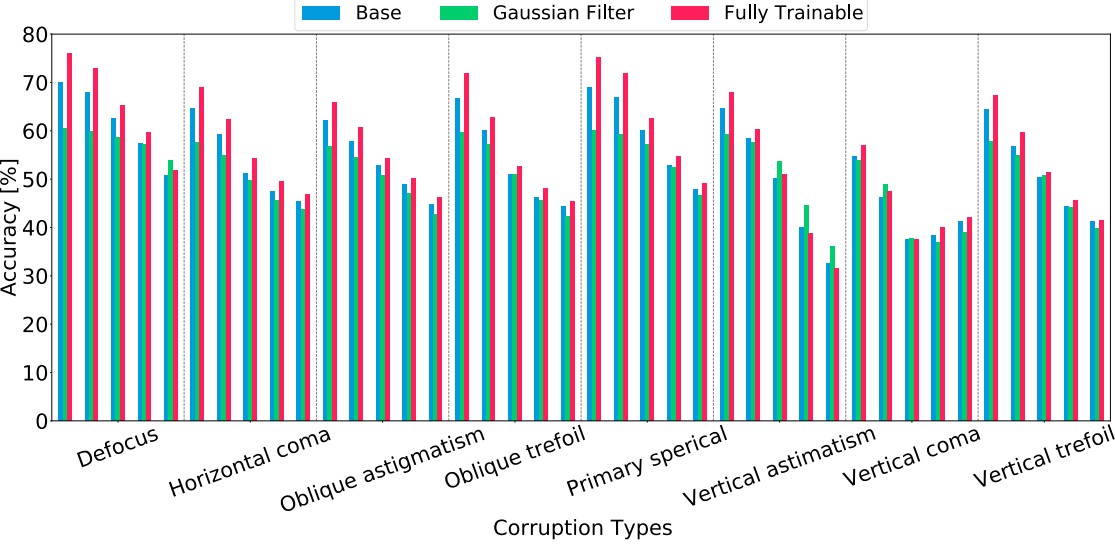

Figure 58: ResNet ImageNet100 OpticsBench. This evaluation demonstrates the benefit of our proposed fully trainable large kernel filter over a fixed Gaussian filter. While the performance gap on ImageNette is smaller, on ImageNet-100, the Gaussian pre-filtered images can not be reliably classified.

## F.1  Static Low Pass Filtering

The first point arises when considering Table 5, in particular the results for the non-trainable, static Gaussian kernel. This kernel defines a particular subspace projection, which one would expect to perform well under high-frequency noise. Yet, it also removes information in a fixed and pre-determined way which leads to a decay in the clean classification accuracy, which becomes more severe as the hardness of the considered

classification task increases. While in ImageNette, the static Gaussian kernel only performs one percent point below a fully learned filter of the same size of $25 \times 25$, the gap is severe when looking at ImageNet-100 in Figure 58. As the classes become more difficult to discriminate, the simple Gaussian blurring yield very unsatisfactory results whereas the proposed fully trainable filter increases model robustness across various corruptions.

## F.2 Sparse Coding

The second point is an extended discussion on the relationship of our approach to sparse coding. In the main paper, we started the discussion of why the standard initial layers of the usual networks, do not learn similar patterns as our proposed dimension-preserving large kernel convolution. These layers are known to learn an overcomplete representation of the data in a sparse coding sense, *i.e.* in every channel of every feature map, only very few features are "active". When considering for example a convolution layer with $3 \times 3$ kernels, it is also clear that the usual increase in feature map channels from 3 in the input to *e.g.* 64 likely leads to redundant information (a $3 \times 3$ filter can be represented by 9 basis vectors, *i.e.* more than 9 feature map channels imply redundancy, not only with respect to the input information but also in the amount of different (*i.e.* linearly independent) features that can be extracted). Consequently, such usual early layers are likely to represent sparse codes, representing every kind of data that is given in the input. Such behavior is likely to impact a model's robustness, as examplified in the below toy example:

Suppose the input data lives in an approximate subspace: the clean data forms (noisy) samples of this subspace since it is only approximate. If this approximate subspace is mapped to higher dimensions via a convolutional input layer with many redundant output channels, more degrees of freedom are available to fit the data points. The optimizer may therefore choose to use additional redundant dimensions to marginally increase the fit of the noisy data points. While this improves performance on the clean data, the extrapolation capabilities to unseen samples may decrease, giving rise to overfitting and by implication worse performance on corrupted data. Forcing the optimizer to stay in the original subspace and choosing a more constrained model therefore may aid extrapolation, *i.e.* generalization capabilities of the resulting model.

In our layer, such behavior is avoided, since we are mapping every channel of the input to exactly one channel in the output of our layer. If a particular piece of information from the input is to be preserved in the output of our layer, the large kernel with its $3 \times 25 \times 25$ weights has to be learned appropriately. As discussed in section 3 **Content Preserving Filters**, only a limited set of special filters can fully preserve the input content. Yet, for the kernel to be learned, it uses gradient signal that is provided from the classification task at hand. In particular, if an input feature is not contributing to the discrimination between given classes, there will be no gradient signal provided to the input that encourages to learn this particular feature, and since the representation learned is not over-complete, it will also not be learned by chance. A large kernel makes it particularly hard to learn local patterns by chance. It learns to predominantly represent signal that is *needed* for the task at hand, *i.e.* the *essential.*

One potential interpretation is to understand the trained layer as an information bottleneck, through which only such information is passed, that is explicitly needed for the classifier to perform well on the training set. However, removing the additional parts of the signal that are not helpful for classification makes the model more robust under input corruptions.