# OpenReview forum: "Learning the essential in less than 2k additional weights - a simple approach to improve image classification stability under corruptions"
_TMLR — Accepted by TMLR_

### Review · Reviewer_ig1t · 2024-03-11

**Summary Of Contributions:**

The paper introduces the use of an extra depth-wise convolutional layer, just after the first convolutional layer. The new component covers a larger neighborhood by the use of a large kernel (25x25) and introduces around 1.9k new parameters. The layers are trained on clean data with no use of augmentation strategies. Results are presented on smaller subsets of ImageNet, namely ImageNette and ImageNet-100.
The main contribution claimed is that the addition of such an extra layer increases the robustness across a set of well known classification architectures including: Resnet, Alexnet, Efficientnet, Mobilenet, Densenet, ConvNext, Vit and Swin v2.

**Audience:**

No

**Claims And Evidence:**

No

**Requested Changes:**

- fully describe the architectures ablated. What are the specific version of the models ablated? Which is the VIT,and Swin model adopted? Were the models trained from scratch or finetuned?
- fully describe the settings for baseline numbers shown. They look particularly low (https://github.com/fastai/imagenette)
- review the discussion on increase of robustness by the removal of signal content, under stronger training settings (larger dataset and following SOTA best practices).

**Strengths And Weaknesses:**

The Strengths of the paper are its simplicity, making it easy to be replicated; and the inclusion of ablations using different architectures, including convolutional based, transformers based and mixed models.

The weakness of the paper can be summarized as:
- the lack of comparison with other approaches
- evaluations covering  only tasks with smaller datasets (ImageNet with 9k and Imagenet-100 with 128k) and a small number of classes (10 or 100 respectively).
- the lack of evidence under stronger training regimes. In the supplementary results, training on the whole ImageNet-1k dataset the performance improvements shown are not as clear, or not presented as the results shown on the main text. Adding to that, results on individual distortions show that the benefit is not shared across severity levels not different distortions.

In summary, the results shown are valid under the low-data regime, and apparently, ablations cover small models only, but has not shown to be robust under larger/sota training regimes.

---

> ### Author Response · Authors · 2024-04-25
> **Answer - I**
>
> * (Q1) *Which is the VIT,and Swin model adopted?* (ig1t/C1) *fully describe the architectures ablated. What are the specific version of the models ablated?*
>     * The Vision Transformer (ViT), which we used is the base version with a patch size of 16. We used two Swin Transformer v2, the smaller version (tiny) and the larger version Swin Transformer v2 (base). Both Swin Transformer v2 are used with a window size of 8. More information for all models are added to the supplementary of the paper in Section C (Experiment Setup).
>
> * (C4) *fully describe the settings for baseline numbers shown. They look particularly low (https://github.com/fastai/imagenette)*
>
>     * As most of the models are designed for larger datasets, such as ImageNet-1k and ImageNet-21k, the used models do not outperform specially ImageNette tuned models (Details are now extended in Section C for clarity). However, to also validate the same effect on these models, we used a model from the ImageNette leaderboard and trained it with and without our input layer. We have added this experiment to Table 2 in Section 4 (Experimental Evaluation). The evaluation results from the baseline and our model, on clean data, are nearly the same as the one in the leaderboard from ImageNette. However, the corruption robustness of our model is improved on OpticsBench by 2.5\% and by 5.2\% for Common Corruptions. The following table shows these results:
>
> | Model | Version | Clean Data | OpticsBench | Common Corruptions |
> | ----- | ------- | ---------- | ----------- | ------------------ |
> | XSEResNext50  |Base|**0.936**|0.652|0.607|
> | XSEResNext50  (ours)| Trainable| 0.933|**0.677**|**0.659**|
>
> * (W1) *the lack of comparison with other approaches*
>
>     * To improve the comparison to previous approaches, we trained models with AugMix from Hendrycks et al. and compared them with our light-weight approach. First, we compare our approach to AugMix and then combine both methods. This can be examined in the first table below. The results of this experiment can now be examined in Section 4.3 (Comparison with Augmentation and Joint Trainable Large Kernel and Augmentation) in the paper in more detail. Furthermore, we compared our performance on ImageNet-100 corrupted data (2D common corruptions & OpticsBench) to the SotA robustness methods.
>
> | Dataset | Model | Version |Clean Data|OpticsBench|Common Corruptions|
> | -------- | -------- | -------- | --- | --- | --- |
> | ImageNette|ResNet50|Base|***0.800**|0.592|*0.487|
> | ImageNette |ResNet50 (ours)|Trainable|*0.775|*0.685|*0.565|
> |ImageNette|ResNet50|AugMix|0.781|0.561|0.512|
> |ImageNette|ResNet50 (ours)|AugMix \& Trainable|0.795|**0.774**|**0.639**|
> |ImageNet-100|ResNet50|Base|*0.801|*0.536|*0.406|
> |ImageNet-100|ResNet50 (ours)|Trainable|*0.797|*0.558|*0.437|
> |ImageNet-100|ResNet50|AugMix|0.809|0.639|0.518|
> |ImageNet-100| ResNet50 (ours)| AugMix \& Trainable  |**0.814**|**0.663**|**0.533**|
>
> |ResNet50 ImageNet-100 version | Clean Data  | OpticsBench  | 2D Common Corruptions  | Costs per epoch [s] |
> | ---------- | --- | --- | --- | ----|
> |Base|0.801|0.536|0.406|174|
> |Prepended Input Layer (ours) |0.797|0.558|0.437|181|
> |AugMix (Hendrycks et al.,2020)  |0.809|0.639|0.518|441|
> |Prepended Input Layer + AugMix (Hendrycks et al.,2020)  (ours)|0.814|0.663|0.533|455
> |DeepAugment (Hendrycks et al., 2021) |0.804 | 0.686 |0.591 | 503
> |NoisyMix (Erichson et al., 2024) |0.801| 0.646 | 0.581 |855|
> |SIN_IN (Geirhos et al., 2018) |0.769| 0.582 | 0.506 | 356
> | DAD (Zhou et al., 2023) | 0.785 | 0.594 | 0.523 | 1056|

---

> ### Author Response · Authors · 2024-04-25
> **Answer - II**
>
> * (W2) *evaluations covering only tasks with smaller datasets (ImageNet with 9k and Imagenet-100 with 128k) and a small number of classes (10 or 100 respectively).*
>     * To highlight the results on a larger dataset, such as ImageNet-1k, we add this part into the main paper (in Section 4.1). We have reported the performance of our method on ImageNet-1k in Table 4. To further examine on these results, we trained an additional model - Swin v2 (base). The following table shows, that our models are able to outperform the baseline even on 3D Common Corruptions. The subsequently table shows
>
> | Model          | Version         | Clean Data | OpticsBench | Common Corruptions |3D Common Corrutions |
> | -------------- | --------------- | ---------- | ----------- | ------------------ | --- |
> | Swin v2 (base) | Base            | **0.783**  | 0.423       | 0.320              |0.330|
> | Swin v2 (base) (ours)| Fully trainable | **0.783**  | **0.459**   | **0.355**          |**0.375**|
>
> * (W4) *Adding to that, results on individual distortions show that the benefit is not shared across severity levels not different distortions.*
>     * Indeed, the benefit varies across severity levels. Yet, only in rare cases does the base model perform better than our proposed model. This trend is captured Figs. 1 and 4, and the general benefit is apparent in averaged numbers in Tables 2, 3 and 4. As seen in Fig. 4, for transformer-based models, the proposed models are more robust across **all** severities.
>
> * (C3) *Were the models trained from scratch or finetuned?*
>     * All models (base and with prepended kernel) have been trained from scratch unless otherwise stated. Clarification has been added in Sec. 4, paragraph 2, line 1.
>
> * (W5) *In summary, the results shown are valid under the low-data regime, (...).*
> (C5) *review the discussion on increase of robustness by the removal of signal content, under stronger training settings (larger dataset and following SOTA best practices).*
> (W3) *the lack of evidence under stronger training regimes. (...).*
>
>     * To address the aforementioned, we train larger models (Swin v2 (base)) on larger datasets (ImageNet-1k), and models like XSEResNext50 which have been trained using SOTA practices, and report their performances in **Tables 2 (ImageNette), 3 (ImageNet-100) and 4 (ImageNet-1k)** with an explanation in Sec. 4.1 of the main paper. The discussion in Sec. 5 holds true for the new experiments as well.
>     * We further summarize our findings here and review the robustness of our method when 1) SOTA practices (used to train leaderboard models) are applied to ImageNette, and 2) SOTA models are trained on larger datasets, and assess if the advantages showcased on simpler models carry over to the state of the art.
> 1. Here we present the effects SOTA practices can have on a model's performance when the model is especially trained on a particular small dataset. As can be seen that XSEResNext50, a leaderboard model on ImageNette (**added in Table 2**), leverages the benefit of the prepended layer in a similar fashion as other models. **This suggests that the latest training regimes do not undercut the advantage of the proposed layer**. An excerpt of Table 2 is given below:
> | Model | Version | Clean Data | OpticsBench | Common Corruptions |
> | ----- | ------- | ---------- | ----------- | ------------------ |
> | XSEResNext50  |Base|**0.936**|0.652|0.607|
> | XSEResNext50  (ours)| Trainable    | 0.933       |**0.677**|**0.659**|
> | ConvNext      |Base          |**0.824**|0.516|0.489|
> | ConvNext (ours)| Trainable    | 0.796       |**0.565**|**0.539**|
>
> 2. For this discussion, we will draw on **newly added results in Table 3 and Table 4**. Firstly, we compare the performance of Swin v2 (tiny) on ImageNet-100 and ImageNet-1k. Swin v2 (tiny) with the prepended kernel, **trained on ImageNet-100, improves the robustness by 4.3% and 5.6%** when evaluated on OpticsBench and Common Corruptions, respectively, over their base versions. We observe a similar advantage, but to a smaller advantage, when the same model is **trained on ImageNet-1k with improvements by 3.1% and 3.4%** on OpticsBench and Common Corruptions, respectively, over their base versions.
> With the Swin v2 (base) **trained on ImageNet-1k, the larger model improves the absolute robustness on a whole, but the addition of our prepended layer still offers about the same percentage improvement in robustness**.
> | Dataset | Model | Version | Clean Data |  OpticsBench | Common Corruptions |
> | -------- | -------- | -------- | -------- | --- | --- |
> | ImageNet-100 |Swin v2 (tiny)|Base|**0.779**|0.433|0.323|
> | ImageNet-100 |Swin v2 (tiny) (ours)|Fully trainable|0.774|**0.476**|**0.379**|
> | ImageNet-1k |Swin v2 (tiny)|Base|**0.778**|0.395|0.292|
> | ImageNet-1k |Swin v2 (tiny) (ours)|Fully trainable|0.771|**0.426**|**0.326**|
> |ImageNet-1k |Swin v2 (base)|Base|**0.783**|0.423|0.320|
> |ImageNet-1k |Swin v2 (base) (ours)|Fully trainable|**0.783**|**0.459**|**0.355**|

---

### Review · Reviewer_8XnL · 2024-03-28

**Summary Of Contributions:**

The authors propose a simple convolutional layer that is prepended to any DNN architecture to improve corruption robustness in image classification tasks. It is shown that having a dimension preserving layer with large kernel size is crucial.

**Audience:**

Yes

**Broader Impact Concerns:**

No concerns.

**Claims And Evidence:**

Yes

**Requested Changes:**

See above.

**Strengths And Weaknesses:**

Strengths:
- Sizable improvements over baseline on standard datasets such as ImageNet-C.
- Thorough evaluation and ablation of different kernel variants, including interesting insights.
- Lightweight and applicable post-hoc without data augmentation.
- Paper nicely written and easy to follow.

Weaknesses and questions:
- My understanding is that models are all trained from scratch. Did the authors try fine-tuning by just training the initial new layer? I.e., can this be done on pre-trained models. If not, what changes in the other layers to improve robustness?
- It would be interesting to see evaluations across a wider range of robustness problems such as other ImageNet variants or easy adversarial examples. [1], for example, has a good mix of ImageNet variants to evaluate on. For adversarial examples, simple one-step or black-box attacks would be interesting. Proper adversarial examples will still succeed 100% of the time (is my assumption), but would be interesting if this at least gets a bit more difficult, especially as there are adversarial training results in the appendix (btw, these could also be interesting for the main paper, but might require additional baselines to be interesting and for adversarially trained models, the attacks are too weak, I would expect AutoAttack as evaluation).

[1] https://openaccess.thecvf.com/content/ICCV2023/papers/Guo_Robustifying_Token_Attention_for_Vision_Transformers_ICCV_2023_paper.pdf

- I am also missing some more sensible baseline. I appreciate that the method does not require additional data augmentation. But it would still be valuable to compare to methods like AugMix [2].

[2] https://arxiv.org/abs/1912.02781

- With more baselines, I am curious why the authors did not explore the effect of the proposed layer on top of data augmentation based methods.
- Experiments with transformer architectures could be in the main paper.
- The method seems to be worse only on Contrast on ImageNet-C (see appendix), any reasoning behind this?

Conclusion:
I generally like the paper but am a bit hesitant because it is unclear whether the approach scales on top of existing data augmentation strategies or other methods.

---

> ### Author Response · Authors · 2024-04-25
> **Answer Review - l**
>
> * (8XnL/Q1) *The method seems to be worse only on Contrast on ImageNet-C (see appendix), any reasoning behind this?*
>     * We thank the reviewer for the discussion. The method is observed to not be as effective against contrast (and more generally, color) corruptions. We postulate that the reduction in the benefit is because the prepended kernel applies a channel-wise convolution on the input, thereby precluding color mixing. The color corruptions are however caused by color mixing so the proposed method finds it harder to mitigate their effects.
>
> * (8XnL/C1) *My understanding is that models are all trained from scratch. Did the authors try fine-tuning by just training the initial new layer? I.e., can this be done on pre-trained models. If not, what changes in the other layers to improve robustness?*
>     * All models (base and with prepnded kernel) have been trained from scratch unless otherwise stated. Clarifications have been added in Sec. 4, 2nd paragraph, 1st line. We experimented with pretrained models, by adding our proposed input layer and only train this input layer. However, the results show, that the model needs to be trained from scratch to ensure the demonstrated performance boost. As the pretrained models are trained without such an input layer, they expect the input data not to be transformed, so that the prepended layers would tend to learn identity mappings. Due to the low number of parameters and therefore the low extra training time, we would recommend the train the models from scratch to make use of the increased accuracy on corrupted data.
>
> * (8XnL/C5) *I am also missing some more sensible baseline. I appreciate that the method does not require additional data augmentation. But it would still be valuable to compare to methods like AugMix [2]. https://arxiv.org/abs/1912.02781*
> (8XnL) *I generally like the paper but am a bit hesitant because it is unclear whether the approach scales on top of existing data augmentation strategies or other methods.*
> (8XnL/C6) *With more baselines, I am curious why the authors did not explore the effect of the proposed layer on top of data augmentation based methods.*
>     * Thank you for the suggestion. We have run these experiments and they are reported in the revised main paper, Sec. 4.3. This indicates, that a combination of AugMix and our proposed layer, is beneficial for the corrupted image accuracy. These experiments are displayed in the following table for convenience:
>
> | Dataset | Model | Version |Clean Data|OpticsBench|Common Corruptions|
> | -------- | -------- | -------- | --- | --- | --- |
> | ImageNette|ResNet50|Base|***0.800**|0.592|*0.487|
> | ImageNette |ResNet50 (ours)|Trainable|*0.775|*0.685|*0.565|
> |ImageNette|ResNet50|AugMix|0.781|0.561|0.512|
> |ImageNette|ResNet50 (ours)|AugMix \& Trainable|0.795|**0.774**|**0.639**|
> |ImageNet-100|ResNet50|Base|*0.801|*0.536|*0.406|
> |ImageNet-100|ResNet50 (ours)|Trainable|*0.797|*0.558|*0.437|
> |ImageNet-100|ResNet50|AugMix|0.809|0.639|0.518|
> |ImageNet-100| ResNet50 (ours)| AugMix \& Trainable     |**0.814**|**0.663**|**0.533**|

---

> ### Author Response · Authors · 2024-04-25
> **Answer Review - ll**
>
> * (8XnL/C2) *It would be interesting to see evaluations across a wider range of robustness problems such as other ImageNet variants or easy adversarial examples. [1], for example, has a good mix of ImageNet variants to evaluate on.*
>
>     * Thank you for pointing us to the paper "Robustifying Token Attention for Vision Transformers" and the corresponding datasets. They are using ImageNet-C, i.e. the ImageNet variant that we refer to as Common Corruptions (Hendrycks et al. 2019) in their main paper.
>     * We further evaluated our models on the domain transfer dataset ImageNet-A. While ImageNet-A does not contain corrupted data but image data from different abstraction levels (e.g. oregami folded strawberries), no strong boost in performance is to be expected. However, we can confirm that our models perform on par with the respective baseline in this case, i.e. there is also no harm in using the proposed approach under such domain shifts.
>     * To extend our evaluation, we furthermore evaluated on 3D Common Corruptions (Kar et al., CVPR 2022), for which we also observe a significant performance boost. This can be examined in the three tables below and in Section D of the appendix.
>
>
> ImageNette:
> | Model    | Version         | CC3D      |
> | -------- | --------------- | --------- |
> | ResNet50 | Base            | 0.509     |
> | ResNet50 (ours)| Fully trainable | **0.665** |
> | AlexNet  | Base            |0.655|
> | AlexNet (ours)| Fully trainable  |**0.733**|
>
> ImageNet-100:
> | Model    | Version         | CC3D      |
> | -------- | --------------- | --------- |
> | ResNet50 | Base            | 0.423     |
> | ResNet50 (ours)| Fully trainable | **0.470** |
> | ConvNeXt  | Base            |0.337|
> | ConvNeXt (ours)| Fully trainable |**0.403**|
>
> ImageNet-1k:
> | Model    | Version         | CC3D      |
> | -------- | --------------- | --------- |
> | MobileNet | Base            | 0.255     |
> | MobileNet (ours)| Fully trainable | **0.271** |
> | Swin v2 (base)  | Base            |0.330|
> | Swin v2 (base) (ours)| Fully trainable |**0.375**|
>
>
>
> * (8XnL/C3) *For adversarial examples, simple one-step or black-box attacks would be interesting. Proper adversarial examples will still succeed 100% of the time (is my assumption), but would be interesting if this at least gets a bit more difficult, especially as there are adversarial training results in the appendix*
>     * Regarding results for FGSM (a one step attack), we had already provided some results in the appendix D7 (yet on adversarially trained models). Thereby, we intended to elaborate that there is potential defence against adverserial one-step attacks. To further experiment with adversarial attacks, we now added AutoAttack (Croce et al., 2020) as an evaluation for our proposed method. Therefore, we attacked the baseline and our models (without adversarial training) with the same adversarial attacks on ImageNette via AutoAttack with an epsilon of 4/255. Under black-box attack (SQUARE, Andriushchenko et al., 2020), our models are also able to achieve higher accuracy than the respective baselines. As the following table suggest, our approach allows models to be more robust against all of the four AutoAttack averserial attacks.
>
> | Model    | Version                | AutoAttack | SQUARE*   |
> | -------- | ---------------------- | ---------- | --------- |
> | ResNet50 | Base                   | 0.199      | 0.156     |
> | ResNet50 (ours)| Fully Trainable | **0.229**  | **0.193** |
> | AlexNet         |Base|0.446|0.518|
> |AlexNet(ours)|Fully Trainable|**0.468**|**0.657**|
>
> We have added these results to Section D.7 in the appendix. If the reviewer prefers, we can also add them to the main paper. Please let us know your opinion.

---

> ### Author Response · Authors · 2024-04-25
> **Answer Review - lll**
>
> * (8XnL/C7) *Experiments with transformer architectures could be in the main paper.*
>     * We thank the reviewer for the comment. We refer the reviewer to Tables 2, 3 and 4 in the main paper, where we have added the performances of ViT, Swin v2 and Swin v2 (base).
>     * By adding another larger state-of-the-art model (Swin v2 base) to our experiments, we want to highlight that our proposed layer not only works for low-parameter convolutional models, but also is able to improve the Swin Transformer v2 (base) on corrupted data. Therefore, we trained the model as proposed in the original paper on ImageNet-1k and once again with our input layer. The results of the subsequent evaluation are stated in our main paper in Section 4.1 (Trainable Large Kernels can Improve Prediction Stability) in table 4. Furthermore, we added detailed results in the appendix in Fig. 50. The performance boost with our proposed layer is even higher on the larger Swin Transformer v2 than on the smaller Swin Transformer v2 (tiny). Additionally, the model with our input layer is not lacking in clean data accuracy. The following table states the evaluation results of the model:
>
> | Model | Version | Clean Data |  OpticsBench | Common Corruptions |
> | -------- | -------- | -------- | --- | --- |
> |Swin v2 (base)|Base|**0.783**|0.423|0.320|
> |Swin v2 (base) (ours)|Fully trainable|**0.783**|**0.459**|**0.355**|

---

### Review · Reviewer_5xbk · 2024-04-11

**Summary Of Contributions:**

The paper introduces a simple modification to image classification models—a large kernel convolution layer added before existing layers—to significantly improve robustness against various corruptions without needing extra data augmentation, demonstrating notable accuracy increases on ImageNet-100 and ImageNette datasets.

**Audience:**

Yes

**Broader Impact Concerns:**

none noted.

**Claims And Evidence:**

Yes

**Requested Changes:**

1. please offer the comparisions to those methods dedicated to vision robustness, at least for corruptions of images, preferrably with comparisons of computational load, so that there is a better chance this proposed method will stand out. I understand these new methods require huge computational load to compare, but please try to include a comprehensive list of new methods by leveraging reported numbers. This one is the latest one I can find that can give the authors some ideas of the latest method for discussion in their table of comparison [1].

2. not sure why the ImageNet-1k result is put into appendix instead of the main paper, and also why only Table 2 reports newer methods like ViT and SWIN, but not Table 3, or Figure 4?

3. the result table in appendix is hard to parse, e.g., five groups of results for each corruptions in figures of Section D.6, what is the first group of each corruption? Then, it seems the proposed method does not outperform base methods in the first block for ImageNet-1k, is this CD, clean data?

4. Does this new kernel method involve the tuning of the base model together with the new kernel? There is no guarantee that the kernel can be directly plugged to pretrained weights, right? In this case, the proposed method also have a non-negligible computational load, maybe roughly the same with some regularization based methods, although probably still faster than augmentation-based methods.

[1] Zhou, Andy, et al. "Distilling Out-of-Distribution Robustness from Vision-Language Foundation Models." Advances in Neural Information Processing Systems 36 (2024).

**Strengths And Weaknesses:**

Strengths:
+ the method is very interesting, adding a simple and efficient component that can improve the performances.

+ the empirical improvement is notable, over the two datasets reported.

+ there are discussions of comparisons of kernels to use.

Weakness:
- the biggest weakness would be the empirical scope. Machine learning robustness has been studied extensively with tons of methods, tested on much bigger scope than the one in this paper, but the empirical comparison has ignored all of them.
    - a potential reason for the authors to ignore them is that this new method is considered efficient, but I would recommend the authors to include them in the table, with a comparison of training time together with accuracy.

-  the method is effectively a preprocessing of images, which is instead of preprocessing, is designed for specialized preprocessing with learnable parameters. The kernels are preprocessing of images is well known to the community already, thus, we need more discussions of this method vs. simply preprocessing of the images (although the authors already mentioned data augmentation, but there seems not enough empirical comparisons)

---

> ### Author Response · Authors · 2024-04-25
> **Answer I**
>
> * (W1) *the biggest weakness would be the empirical scope (...).*
> *a potential reason for the authors to ignore them is that this new method is considered efficient, but I would recommend the authors to include them in the table, with a comparison of training time together with accuracy.*
> *please offer the comparisons to those methods dedicated to vision robustness, at least for corruptions of images, preferrably with comparisons of computational load,(...). This one is the latest one I can find that can give the authors some ideas of the latest method for discussion in their table of comparison.[1]*
>     * Thanks for pointing us to the "Distilling Out-of-Distribution Robustness from Vision-Language Foundation Models"-paper. To increase the scope and the comparison of the paper we compared our performance on ImageNet-100 corrupted data (2D common corruptions & OpticsBench) to the SotA robustness methods. In the following table, we add a comparison of accuracy on ImageNet (2D common corruptions & OpticsBench) and training time for different data augmentation methods (AugMix, DeepAugment, NoisyMix, joint training on Stylized ImageNet + ImageNet) from the RobustBench leaderboard. We agree that a comparison on ImageNet1k should be added to the main paper, which we could not finish in time due to limited computational resources and short timeframe. We also added an extensive review of these methods to the related work section. If you agree, we will add a table alike the one below computed on ImageNet-1k to the main paper as soon as the numbers are available to us (within the next few days).
>
> |ResNet50 ImageNet-100 version | Clean Data  | OpticsBench  | 2D Common Corruptions  | Costs per epoch [s] |
> | ---------- | --- | --- | --- | ----|
> |Base|0.801|0.536|0.406|174|
> |Prepended Input Layer (ours) |0.797|0.558|0.437|181|
> |AugMix (Hendrycks et al.,2020)  |0.809|0.639|0.518|441|
> |Prepended Input Layer + AugMix (Hendrycks et al.,2020)  (ours)|0.814|0.663|0.533|455
> |DeepAugment (Hendrycks et al., 2021) |0.804 | 0.686 |0.591 | 503
> |NoisyMix (Erichson et al., 2024) |0.801| 0.646 | 0.581 |855|
> |SIN_IN (Geirhos et al., 2018) |0.769| 0.582 | 0.506 | 356
> | DAD (Zhou et al., 2023) | 0.785 | 0.594 | 0.523 | 1056|
>
>
> * (W2) *(...) we **need more discussions of this method vs. simply preprocessing** of the images (although the authors already mentioned data augmentation, but there seems not enough empirical comparisons)*
>     * To show that our model has the possibility to be used in combination with other prepocessing methods, we trained a model with AugMix from Hendrycks et al. and our input layer. The combination of both methods with a comparison with the baseline can be found in the following table. Furthermore, the results of this experiment can be examined in Section D.7 (Data Augmentation Comparison) in more detail and has been added to the main paper in Sec. 4.3.
>
> | Dataset | Model | Version |Clean Data|OpticsBench|Common Corruptions|
> | -------- | -------- | -------- | --- | --- | --- |
> | ImageNette|ResNet50|Base|***0.800**|0.592|*0.487|
> | ImageNette |ResNet50 (ours)|Trainable|*0.775|*0.685|*0.565|
> |ImageNette|ResNet50|AugMix|0.781|0.561|0.512|
> |ImageNette|ResNet50 (ours)|AugMix \& Trainable|0.795|**0.774**|**0.639**|
> |ImageNet-100|ResNet50|Base|*0.801|*0.536|*0.406|
> |ImageNet-100|ResNet50 (ours)|Trainable|*0.797|*0.558|*0.437|
> |ImageNet-100|ResNet50|AugMix|0.809|0.639|0.518|
> |ImageNet-100| ResNet50 (ours)| AugMix \& Trainable     |**0.814**|**0.663**|**0.533**|
>
>  * (Q1) *Does this new kernel method involve the tuning of the base model together with the new kernel?*
>     * Unless otherwise stated, the prepended kernel is trained together with the model from scratch. We clarify this in Sec. 4, paragraph 2, line 1.

---

> > ### Author Response · Authors · 2024-05-03
> > **Addional ImageNet-1k comparison results**
> >
> > As mentioned in the reply above, here are the ImageNet-1k comparison results regarding accuracy (2D common corruptions & OpticsBench) and training time for different data augmentation methods (AugMix, DeepAugment, NoisyMix, joint training on Stylized ImageNet + ImageNet) from the RobustBench leaderboard. We do not have a model, which combines a data augmentation method and our proposed input layer on ImageNet-1k yet (we just started the training), however we are optimistic that this combination can further increase the accuracy on corrupted data, as already examined in the prior table on ImageNet-100 with AugMix and our input layer.
> >
> > |ResNet50 ImageNet-1k version | Clean Data  | OpticsBench  | 2D Common Corruptions  | Costs per epoch [s] |
> > | ---------- | --- | --- | --- | ----|
> > |Base|0.781|0.482|0.393| 1,540|
> > |Prepended Input Layer (ours) |0.774|0.509|0.425| 1,580|
> > |AugMix (Hendrycks et al., 2020) | 0.773 | 0.633 |0.511 | 11,338|
> > |DeepAugment (Hendrycks et al., 2020) |0.769 | 0.637 |0.529 | 4,971
> > |NoisyMix (Erichson et al., 2024) |0.769| 0.607 | 0.532 |7,988|
> > |SIN_IN (Geirhos et al., 2018) |0.750| 0.537 | 0.457 | 3,059
> > | DAD (Zhou et al., 2023) | 0.802 | 0.502 | 0.495 | 10,352 |

---

> ### Author Response · Authors · 2024-04-25
> **Answer - II**
>
> * (Q2) *There is no guarantee that the kernel can be directly plugged to pretrained weights (...).*
>
>     * For all our experiments, we trained the models from scratch. To highlight this, we added a remark to Section 4 (Experimental Evaluation) and Section C (Experiment Setup) in the appendix. Since our number of extra parameters is low, this training from scratch is actually very affordable in practice for many models, which makes our approach highly relevant when training models on new data and domains. We now specify and discuss the benefits and costs of leveraging pre-trained models versus training from scratch in the related work section, paragraph "Image Corruptions and Data Augmentations".
>     * To test, whether our proposed channel-wise input layer, can be used in combination with pretrained models, we used pre-trained models from PyTorch and added our input layer to the model. Subsequently, the model was trained, while freezing all layers except of the input layer. Due to the importance of this question, we added this to these experiments to the paper in Section D.9 (Prepend Input Layer on Pretrained Models) in the appendix. Furthermore, the results can be found in the following table.
>     * In fact, there seems to be some benefit when it comes to robustness towards common corruptions. Yet, this evaluation shows that it is highly beneficial to train the entire model from scratch rather than using a pretrained model.
>
>     * the additional computational cost for the prepended layer is about 4% (measured on ImageNet-100 and ImageNet-1k for ResNet50).
>
> | Dataset | Model | Version | Clean Data  | OpticsBench  | Common Corruptions  |
> | ------- | ----- | ------- | --- | --- | --- |
> |ImageNette |ResNet50|Pretrained Base|**0.987**|**0.870**|0.807|
> |ImageNette |ResNet50 (ours)|Prepended Input Layer|0.986|0.851|**0.813**|
> |ImageNet-100|ResNet50|Pretrained Base|**0.868**|**0.615**|0.467|
> | ImageNet-100|ResNet50 (ours)|Prepended Input Layer|0.866|0.607 |**0.470**|
>
>
> * (C3) *not sure why the ImageNet-1k result is put into appendix instead of the main paper*,
>
>     * We have included the ImageNet-1k results in Section 4.1 (Trainable Large Kernels can Improve Prediction Stability) of the main paper. In the same section, we also added the transformer-based models to Table 3 and Figure 4. To highlight the experiment on the larger ImageNet-1k dataset more, we increased the scope by adding another model to the experiments (Swin Transformer v2 base). Therefore, we trained the model as proposed in the paper on ImageNet-1k and once again with our input layer. The results of the subsequent evaluation are stated in the main paper in Section 4.1 (Trainable Large Kernels can Improve Prediction Stability) in Table 5. Furthermore, we added the detailed results in the appendix in Fig. 50. The performance boost with our proposed layer on the larger Swin Transformer v2 is even higher than on the smaller Swin Transformer v2 (tiny). Additionally, the model with our input layer is not lacking in clean data accuracy. The following table states the evaluation results of the model:
>
> | Dataset | Model | Version | Clean Data |  OpticsBench | Common Corruptions |
> | -----------| -------- | -------- | -------- | --- | --- |
> |ImageNet-1k | Swin v2 (base)|Base|**0.783**|0.423|0.320|
> |ImageNet-1k | Swin v2 (base) (ours)|Fully trainable|**0.783**|**0.459**|**0.355**|
>
>  * The performance boost with our proposed layer on the larger Swin Transformer v2 is even higher than on the smaller Swin Transformer v2 (tiny). Additionally, the model with our input layer is not lacking in clean data accuracy. A detailed ImageNet-1k OpticsBench evaluation can be examined in **Figure 50**.
>
> * (C4) *and also why only Table 2 reports newer methods like ViT and SWIN, but not Table 3, or Figure 4?*
>
>     * Thank you for the suggestion. We added the transformer-based models to tables 3 and 4 in the main paper.
>
> * (C5) *The result table in appendix is hard to parse, (...)*
>     * The first bar for each corruption is the lowest (1) severity of the corresponding corruption. We extended the caption of all figures in Section D (Additional Experimental Results) in the appendix accordingly and are open to any suggestions.

---

### Author Response · Authors · 2024-04-24
**Revision Summary**

We would like to thank all reviewers for their valuable and positive feedback! We are in particular glad that they find that our "method is very interesting" - Reviewer 5xbk, with "Sizable improvements over baseline on standard datasets such as ImageNet-C" - Reviewer 8XnL, and "The Strengths of the paper are its simplicity, making it easy to be replicated" - Reviewer ig1t.
We are also glad for the feedback on improving our manuscript. We tried to incorporate the suggestions as permitted by our available (academic) compute resources and short timeframe. Only a few of the suggested evaluations are missing in the current revision (e.g. ImageNet-1k trained strong ResNet50 baseline), which we will add as soon as our computations finish.
An overview of our changes is listed in the "Changes Since Last Submission". The parts that have been changed in the manuscript are highlighted in blue and figures as well as tables that have been added are indicated as such by blue captions.

---

### Decision · Action_Editor_KxQe · 2024-05-21

**Recommendation:** Accept as is

**Comment:**

The submission meets the bar for acceptance on both the Claims and Evidence and Audience criteria. Reviewers also noted its writing quality and clarity (8XnL, ig1t).

**Audience:**

All reviewers agree that the submission is of interest to at least some individuals in TMLR's audience which they signaled in their official recommendation's "Audience" answer and through their review:

*  "method is very interesting" (5xbk)
* "interesting insights [from the evaluation and ablation of different kernel variants]" (8XnL)

**Claims And Evidence:**

Opinions were initially mixed. On the positive side:

* "empirical improvement is notable" (5xbk)
* "sizable improvements over baseline on standard datasets such as ImageNet-C (8XnL)
* "thorough evaluation and ablation of different kernel variants" (8XnL)
* "results shown are valid under the low-data regime" (ig1t)

On the negative side, all reviewers shared concerns over the scope of the empirical evidence presented:

* "[machine learning] robustness has been studied extensively with tons of methods, tested on much bigger scope than the one in this paper, but the empirical comparison has ignored all of them" (5xbk)
* "would be interesting to see evaluations across a wider range of robustness problems such as other ImageNet variants or easy adversarial examples" (8XnL)
* "missing some more sensible baseline" (8XnL)
* "lack of comparison with other approaches" (ig1t)

In their response the authors presented several new results to address these concerns to the reviewers' satisfaction (in Reviewer 5xbk's words, "[the] authors have taken a substantial amount of efforts to offer more results needed"), and as a result all reviewers agree that the claims are supported by accurate, convincing, and clear evidence, which they signaled in their official recommendation's "Claims And Evidence".